# Sinkhorn Natural Gradient for Generative Models

**Zebang Shen***   **Zhenfu Wang**[†]   **Alejandro Ribeiro***   **Hamed Hassani***

*Department of Electrical and Systems Engineering   [†]Department of Mathematics
University of Pennsylvania
{zebang@seas,zwang423@math,aribeiro@seas,hassani@seas}.upenn.edu

## Abstract

We consider the problem of minimizing a functional over a parametric family of probability measures, where the parameterization is characterized via a push-forward structure. An important application of this problem is in training generative adversarial networks. In this regard, we propose a novel Sinkhorn Natural Gradient (SiNG) algorithm which acts as a steepest descent method on the probability space endowed with the Sinkhorn divergence. We show that the Sinkhorn information matrix (SIM), a key component of SiNG, has an explicit expression and can be evaluated accurately in complexity that scales logarithmically with respect to the desired accuracy. This is in sharp contrast to existing natural gradient methods that can only be carried out approximately. Moreover, in practical applications when only Monte-Carlo type integration is available, we design an empirical estimator for SIM and provide the stability analysis. In our experiments, we quantitatively compare SiNG with state-of-the-art SGD-type solvers on generative tasks to demonstrate its efficiency and efficacy of our method.

## 1   Introduction

Consider the minimization of a functional $\mathcal{F}$ over a parameterized family probability measures $\{\alpha_\theta\}$:

$$\min_{\theta \in \Theta} \{F(\theta) := \mathcal{F}(\alpha_\theta)\}, \tag{1}$$

where $\Theta \subseteq \mathbb{R}^d$ is the feasible domain of the parameter $\theta$. We assume that the measures $\alpha_\theta$ are defined over a common ground set $\mathcal{X} \subseteq \mathbb{R}^q$ with the following structure: $\alpha_\theta = T_{\theta\sharp}\mu$, where $\mu$ is a fixed and known measure and $T_\theta$ is a push-forward mapping. More specifically, $\mu$ is a simple measure on a latent space $\mathcal{Z} \subseteq \mathbb{R}^{\bar{q}}$, such as the standard Gaussian measure $\mu = \mathcal{N}(\mathbf{0}_{\bar{q}}, \mathbf{I}_{\bar{q}})$, and the parameterized map $T_\theta : \mathcal{Z} \to \mathcal{X}$ transforms the measure $\mu$ to $\alpha_\theta$. This type of push-forward parameterization is commonly used in deep generative models, where $T_\theta$ represents a neural network parametrized by weights $\theta$ [Goodfellow et al., 2014, Salimans et al., 2018, Genevay et al., 2018]. Consequently, methods to efficiently and accurately solve problem (1) are of great importance in machine learning.

The de facto solvers for problem (1) are generic nonconvex optimizers such as Stochastic Gradient Descent (SGD) and its variants, Adam [Kingma and Ba, 2014], Amsgrad [Reddi et al., 2019], RMSProp [Hinton et al.], etc. These optimization algorithms directly work on the parameter space and are agnostic to the fact that $\alpha_\theta$'s are probability measures. Consequently, SGD type solvers suffer from the complex optimization landscape induced from the neural-network mappings $T_\theta$.

An alternative to SGD type methods is the natural gradient method, which is originally motivated from Information Geometry [Amari, 1998, Amari et al., 1987]. Instead of simply using the Euclidean structure of the parameter space $\Theta$ in the usual SGD, the natural gradient method endows the parameter space with a "natural" metric structure by pulling back a known metric on the probability space and then searches the steepest descent direction of $F(\theta)$ in the "curved" neighborhood of $\theta$. In particular, the natural gradient update is invariant to reparametrization. This allows natural gradient to

avoid the undesirable saddle point or local minima that are artificially created by the highly nonlinear maps $T_\theta$. The classical Fisher-Rao Natural Gradient (FNG) [Amari, 1998] as well as its many variants [Martens and Grosse, 2015, Thomas et al., 2016, Song et al., 2018] endows the probability space with the KL divergence and admits update direction in closed form. However, the update rules of these methods all require the evaluation of the score function of the variable measure. Leaving aside its existence, this quantity is in general difficult to compute for push-forward measures, which limits the application of FNG type methods in the generative models. Recently, Li and Montúfar [2018] propose to replace the KL divergence in FNG by the Wasserstein distance and propose the Wasserstein Natural Gradient (WNG) algorithm. WNG shares the merit of reparameterization invariance as FNG while avoiding the requirement of the score function. However, the Wasserstein information matrix (WIM) is very difficult to compute as it does not attain a closed form expression when the dimension $d$ of parameters is greater than 1, rendering WNG impractical.

Following the line of natural gradient, in this paper, we propose Sinkhorn Natural Gradient (SiNG), an algorithm that performs the steepest descent of the objective functional $\mathcal{F}$ on the probability space with the Sinkhorn divergence as the underlying metric. Unlike FNG, SiNG requires only to sample from the variable measure $\alpha_\theta$. Moreover, the Sinkhorn information matrix (SIM), a key component in SiNG, can be computed in logarithmic time in contrast to WIM in WNG. Concretely, we list our contributions as follows:

1. We derive the Sinkhorn Natural Gradient (SiNG) update rule as the exact direction that minimizes the objective functional $\mathcal{F}$ within the Sinkhorn ball of radius $\epsilon$ centered at the current measure. In the asymptotic case $\epsilon \to 0$, we show that the SiNG direction only depends on the Hessian of the Sinkhorn divergence and the gradient of the function $F$, while the effect of the Hessian of $F$ becomes negligible. Further, we prove that SiNG is invariant to reparameterization in its continuous-time limit (i.e. using the infinitesimal step size).

2. We explicitly derive the expression of the Sinkhorn information matrix (SIM), i.e. the Hessian of the Sinkhorn divergence with respect to the parameter $\theta$. We then show the SIM can be computed using logarithmic (w.r.t. the target accuracy) function operations and integrals with respect to $\alpha_\theta$.

3. When only Monte-Carlo integration w.r.t. $\alpha_\theta$ is available, we propose to approximate SIM with its empirical counterpart (eSIM), i.e. the Hessian of the empirical Sinkhorn divergence. Further, we prove stability of eSIM. Our analysis relies on the fact that the Fréchet derivative of Sinkhorn potential with respect to the parameter $\theta$ is continuous with respect to the underlying measure $\mu$. Such result can be of general interest.

In our experiments, we pretrain the discriminators for the celebA and cifar10 datasets. Fixing the discriminator, we compare SiNG with state-of-the-art SGD-type solvers in terms of the generator loss. The result shows the remarkable superiority of SiNG in both efficacy and efficiency.

**Notation:** Let $\mathcal{X} \subseteq \mathbb{R}^q$ be a compact ground set. We use $\mathcal{M}_1^+(\mathcal{X})$ to denote the space of probability measures on $\mathcal{X}$ and use $\mathcal{C}(\mathcal{X})$ to denote the family of continuous functions mapping from $\mathcal{X}$ to $\mathbb{R}$. For a function $f \in \mathcal{C}(\mathcal{X})$, we denote its $L^\infty$ norm by $\|f\|_\infty := \max_{x \in \mathcal{X}} |f(x)|$ and its gradient by $\nabla f$. For a functional on general vector spaces, the Fréchet derivative is formally defined as follows. Let V and W be normed vector spaces, and $U \subseteq V$ be an open subset of V. A function $\mathcal{F} : U \to W$ is called Fréchet differentiable at $x \in U$ if there exists a bounded linear operator $A : V \to W$ such that

$$\lim_{\|h\| \to 0} \frac{\|\mathcal{F}(x + h) - \mathcal{F}(x) - Ah\|_W}{\|h\|_V} = 0. \qquad (2)$$

If there exists such an operator $A$, it will be unique, so we denote $D\mathcal{F}(x) = A$ and call it the *Fréchet derivative*. From the above definition, we know that $D\mathcal{F} : U \to T(V, W)$ where $T(V, W)$ is the family of bounded linear operators from V to W. Given $x \in U$, the linear map $D\mathcal{F}(x)$ takes one input $y \in V$ and outputs $z \in W$. This is denoted by $z = D\mathcal{F}(x)[y]$. We then define the operator norm of $D\mathcal{F}$ at $x$ as $\|D\mathcal{F}(x)\|_{op} := \max_{h \in V} \frac{\|D\mathcal{F}(x)[h]\|_W}{\|h\|_V}$. Further, the second-order Fréchet derivative of $\mathcal{F}$ is denoted as $D^2\mathcal{F} : U \to L^2(V \times V, W)$, where $L^2(V \times V, W)$ is the family of all continuous bilinear maps from V to W. Given $x \in U$, the bilinear map $D^2\mathcal{F}(x)$ takes two inputs $y_1, y_2 \in V$ and outputs $z \in W$. We denote this by $z = D^2\mathcal{F}(x)[y_1, y_2]$. If a function $\mathcal{F}$ has multiple variables, we use $D_i f$ to denote the Fréchet derivative with its $i^{th}$ variable and use $D^2_{ij}\mathcal{F}$ to denote the corresponding second-order terms. Finally, $\circ$ denotes the composition of functions.

## 2 Related Work on Natural Gradient

The Fisher-Rao natural gradient (FNG) [Amari, 1998] is a now classical algorithm for the functional minimization over a class of parameterized probability measures. However, unlike SiNG, FNG as well as its many variants [Martens and Grosse, 2015, Thomas et al., 2016, Song et al., 2018] requires to evaluate the score function $\nabla_\theta \log p_\theta$ ($p_\theta$ denotes the p.d.f. of $\alpha_\theta$). Leaving aside its existence issue, the score function for the generative model $\alpha_\theta$ is difficult to compute as it involves $T_\theta^{-1}$, the inversion of the push-forward mapping, and $\det(JT_\theta^{-1})$, the determinant of the Jacobian of $T_\theta^{-1}(z)$. One can possibly recast the computation of the score function as a dual functional minimization problem over all continuous functions on $\mathcal{X}$ [Essid et al., 2019]. However, such functional minimization problem itself is difficult to solve. As a result, FNG has limited applicability in our problem of interest.

Instead of using the KL divergence, Li and Montúfar [2018] propose to measure the distance between (discrete) probability distributions using the optimal transport and develop the Wasserstein Natural Gradient (WNG). WNG inherits FNG's merit of reparameterization invariance. However, WNG requires to compute the Wasserstein information matrix (WIM), which does not attain a closed form expression when $d > 1$, rendering WNG impractical [Li and Zhao, 2019, Li and Montúfar, 2020]. As a workaround, one can recast a single WNG step to a dual functional maximization problem via the Legendre duality. While itself remains challenging and can hardly be globally optimized, Li et al. [2019] simplify the dual subproblem by restricting the optimization domain to an affine space of functions (a linear combinations of several bases). Clearly, the quality of this solver depends heavily on the accuracy of this affine approximation. Alternatively, Arbel et al. [2019] restrict the dual functional optimization to a Reproducing Kernel Hilbert Space (RKHS). By adding two additional regularization terms, the simplified dual subproblem admits a closed form solution. However, in this way, the gap between the original WNG update and its kernelized version cannot be properly quantified without overstretched assumptions.

## 3 Preliminaries

We first introduce the entropy-regularized optimal transport distance and then its debiased version, i.e. the Sinkhorn divergence. Given two probability measures $\alpha, \beta \in \mathcal{M}_1^+(\mathcal{X})$, the entropy-regularized optimal transport distance $\mathrm{OT}_\gamma(\alpha, \beta) : \mathcal{M}_1^+(\mathcal{X}) \times \mathcal{M}_1^+(\mathcal{X}) \to \mathbb{R}_+$ is defined as

$$\mathrm{OT}_\gamma(\alpha, \beta) = \min_{\pi \in \Pi(\alpha, \beta)} \langle c, \pi \rangle + \gamma \mathrm{KL}(\pi || \alpha \otimes \beta). \tag{3}$$

Here, $\gamma > 0$ is a fixed regularization parameter, $\Pi(\alpha, \beta)$ is the set of joint distributions over $\mathcal{X}^2$ with marginals $\alpha$ and $\beta$, and we use $\langle c, \pi \rangle$ to denote $\langle c, \pi \rangle = \int_{\mathcal{X}^2} c(x, y) \mathbf{d}\pi(x, y)$. We also use $\mathrm{KL}(\pi || \alpha \otimes \beta)$ to denote the *Kullback-Leibler divergence* between the candidate transport plan $\pi$ and the product measure $\alpha \otimes \beta$.

Note that $\mathrm{OT}_\gamma(\alpha, \beta)$ is not a valid metric as there exists $\alpha \in \mathcal{M}_1^+(\mathcal{X})$ such that $\mathrm{OT}_\gamma(\alpha, \alpha) \neq 0$ when $\gamma \neq 0$. To remove this bias, consider the *Sinkhorn divergence* $\mathcal{S}(\alpha, \beta) : \mathcal{M}_1^+(\mathcal{X}) \times \mathcal{M}_1^+(\mathcal{X}) \to \mathbb{R}_+$ introduced in Peyré et al. [2019]:

$$\mathcal{S}(\alpha, \beta) := \mathrm{OT}_\gamma(\alpha, \beta) - \frac{\mathrm{OT}_\gamma(\alpha, \alpha)}{2} - \frac{\mathrm{OT}_\gamma(\beta, \beta)}{2}, \tag{4}$$

which can be regarded as a *debiased version* of $\mathrm{OT}_\gamma(\alpha, \beta)$. Since $\gamma$ is fixed throughout this paper, we omit the subscript $\gamma$ for simplicity. It has been proved that $\mathcal{S}(\alpha, \beta)$ is nonnegative, bi-convex and metrizes the convergence in law for a compact $\mathcal{X}$ and a Lipschitz metric $c$ Peyré et al. [2019].

**The Dual Formulation and Sinkhorn Potentials.** The entropy-regularized optimal transport problem $\mathrm{OT}_\gamma$, given in (3), is convex with respect to the joint distribution $\pi$: Its objective is a sum of a linear functional and the convex KL-divergence, and the feasible set $\Pi(\alpha, \beta)$ is convex. Consequently, there is no gap between the primal problem (3) and its Fenchel dual. Specifically, define

$$\mathcal{H}_2(f, g; \alpha, \beta) := \langle f, \alpha \rangle + \langle g, \beta \rangle - \gamma \langle \exp(\frac{1}{\gamma}(f \oplus g - c)) - 1, \alpha \otimes \beta \rangle, \tag{5}$$

where we denote $(f \oplus g)(x, y) = f(x) + g(y)$. We have

$$\mathrm{OT}_\gamma(\alpha, \beta) = \max_{f, g \in \mathcal{C}(\mathcal{X})} \{\mathcal{H}_2(f, g; \alpha, \beta)\} = \langle f_{\alpha, \beta}, \alpha \rangle + \langle g_{\alpha, \beta}, \beta \rangle, \tag{6}$$

where $f_{\alpha, \beta}$ and $g_{\alpha, \beta}$, called the *Sinkhorn potentials* of $\mathrm{OT}_\gamma(\alpha, \beta)$, are the maximizers of (6).

**Training Adversarial Generative Models.** We briefly describe how (1) captures the generative adversarial model (GAN): In training a GAN, the objective functional in (1) itself is defined through a maximization subproblem $\mathcal{F}(\alpha_\theta) = \max_{\xi \in \Xi} \mathcal{G}(\xi; \alpha_\theta)$. Here $\xi \in \Xi \subseteq \mathbb{R}^{\bar{d}}$ is some dual adversarial variable encoding an adversarial discriminator or ground cost. For example, in the ground cost adversarial optimal transport formulation of GAN [Salimans et al., 2018, Genevay et al., 2018], we have $\mathcal{G}(\xi; \alpha_\theta) = \mathcal{S}_{c_\xi}(\alpha_\theta, \beta)$. Here, with a slight abuse of notation, $\mathcal{S}_{c_\xi}(\alpha_\theta, \beta)$ denotes the Sinkhorn divergence between the parameterized measure $\alpha_\theta$ and a given target measure $\beta$. Notice that the symmetric ground cost $c_\xi$ in $\mathcal{S}_{c_\xi}$ is no longer fixed to any pre-specified distance like $\ell_1$ or $\ell_2$ norm. Instead, $c_\xi$ is encoded by a parameter $\xi$ so that $\mathcal{S}_{c_\xi}$ can distinguish $\alpha_\theta$ and $\beta$ in an adaptive and adversarial manner. By plugging the above $\mathcal{F}(\alpha_\theta)$ to (1), we recover the generative adversarial model proposed in [Genevay et al., 2018]:

$$\min_{\theta \in \Theta} \max_{\xi \in \Xi} \mathcal{S}_{c_\xi}(\alpha_\theta, \beta). \tag{7}$$

## 4 Methodology

In this section, we derive the Sinkhorn Natural Gradient (SiNG) algorithm as a steepest descent method in the probability space endowed with the Sinkhorn divergence metric. Specifically, SiNG updates the parameter $\theta^t$ by

$$\theta^{t+1} := \theta^t + \eta \cdot \mathbf{d}^t \tag{8}$$

where $\eta > 0$ is the step size and the update direction $\mathbf{d}^t$ is obtained by solving the following problem. Recall the objective $F$ in (1) and the Sinkhorn divergence $\mathcal{S}$ in (4). Let $\mathbf{d}^t = \lim_{\epsilon \to 0} \frac{\Delta \theta_\epsilon^t}{\sqrt{\epsilon}}$, where

$$\Delta \theta_\epsilon^t := \underset{\Delta \theta \in \mathbb{R}^d}{\operatorname{argmin}} F(\theta^t + \Delta \theta) \quad \text{s.t.} \quad \|\Delta \theta\| \leqslant \epsilon^{c_1}, \mathcal{S}(\alpha_{\theta^t + \Delta \theta}, \alpha_{\theta^t}) \leqslant \epsilon + \epsilon^{c_2}. \tag{9}$$

Here the exponent $c_1$ and $c_2$ can be arbitrary real satisfying $1 < c_2 < 1.5$, $c_1 < 0.5$ and $3c_1 - 1 \geqslant c_2$. Proposition 4.1 depicts a simple expression of $\mathbf{d}^t$. Before proceeding to derive this expression, we note that $\Delta \theta = 0$ globally minimizes the non-negative function $\mathcal{S}(\alpha_{\theta^t + \Delta \theta}, \alpha_{\theta^t})$, which leads to the following first and second order optimality criteria:

$$\nabla_\theta \mathcal{S}(\alpha_\theta, \alpha_{\theta^t})_{|\theta = \theta^t} = 0 \quad \text{and} \quad \mathbf{H}(\theta^t) := \nabla_\theta^2 \mathcal{S}(\alpha_\theta, \alpha_{\theta^t})_{|\theta = \theta^t} \succcurlyeq 0. \tag{10}$$

This property is critical in deriving the explicit formula of the Sinkhorn natural gradient. From now on, the term $\mathbf{H}(\theta^t)$, which is a key component of SiNG, will be referred to as the *Sinkhorn information matrix (SIM)*.

**Proposition 4.1.** *Assume that the minimum eigenvalue of $\mathbf{H}(\theta^t)$ is strictly positive (but can be arbitrary small) and that $\nabla_\theta^2 F(\theta)$ and $\mathbf{H}(\theta)$ are continuous w.r.t. $\theta$. The SiNG direction has the following explicit expression*

$$\mathbf{d}^t = -\frac{\sqrt{2}}{\sqrt{\langle \mathbf{H}(\theta^t)^{-1} \nabla_\theta F(\theta^t), \nabla_\theta F(\theta^t) \rangle}} \cdot \mathbf{H}(\theta^t)^{-1} \nabla_\theta F(\theta^t). \tag{11}$$

Interestingly, the SiNG direction does not involve the Hessian of $F$. This is due to a Lagrangian-based argument that we sketch here. Note that the continuous assumptions on $\nabla_\theta^2 F(\theta)$ and $\mathbf{H}(\theta)$ enable us to approximate the objective and the constraint in (9) via the second-order Taylor expansion.

*Proof sketch for Proposition 4.1.* The second-order Taylor expansion of the Lagrangian of (9) is

$$\bar{G}(\Delta \theta) = F(\theta^t) + \langle \nabla_\theta F(\theta^t), \Delta \theta \rangle + \frac{1}{2} \langle \nabla_\theta^2 F(\theta^t) \Delta \theta, \Delta \theta \rangle + \frac{\lambda}{2} \langle \mathbf{H}(\theta^t) \Delta \theta, \Delta \theta \rangle - \lambda \epsilon - \lambda \epsilon^{c_2}, \tag{12}$$

where $\lambda \geqslant 0$ is the dual variable. Since the minimum eigenvalue of $\mathbf{H}(\theta^t)$ is strictly positive, for a sufficiently small $\epsilon$, by taking $\lambda = \mathcal{O}(\frac{1}{\sqrt{\epsilon}})$, we have that $\mathbf{H}(\theta^t) + \frac{1}{\lambda} \nabla_\theta^2 F(\theta^t)$ is also positive definite. In such case, a direct computation reveals that $\bar{G}$ is minimized at

$$\overline{\Delta \theta^*} = -\frac{1}{\lambda} \left( \mathbf{H}(\theta^t) + \frac{1}{\lambda} \nabla_\theta^2 F(\theta^t) \right)^{-1} \nabla_\theta F(\theta^t). \tag{13}$$

Consequently, the term involving $\nabla_\theta^2 F(\theta^t)$ vanishes when $\epsilon$ approaches zero and we obtain the result.

The above argument is made precise in Appendix A.1. $\qquad\square$

**Remark 4.1.** *Note that our derivation also applies to the Fisher-Rao natural gradient or the Wasserstein natural gradient: If we replace the Sinkhorn divergence by the KL divergence (or the Wasserstein distance), the update direction $\mathbf{d}^t \simeq [\mathbf{H}(\theta^t)]^{-1} \nabla_\theta F(\theta^t)$ still holds, where $\mathbf{H}(\theta^t)$ is the Hessian matrix of the KL divergence (or the Wasserstein distance). This observation works for a general functional as a local metric Thomas et al. [2016] as well.*

The following proposition states that SiNG is invariant to reparameterization in its continuous time limit ($\eta \to 0$). The proof is stated in Appendix A.2.

**Proposition 4.2.** *Let $\Phi$ be an invertible and smoothly differentiable function and denote a reparameterization $\phi = \Phi(\theta)$. Define $\tilde{\mathbf{H}}(\bar{\phi}) := \nabla_\phi^2 \mathcal{S}(\alpha_{\Phi^{-1}(\phi)}, \alpha_{\Phi^{-1}(\bar{\phi})})_{|\phi = \bar{\phi}}$ and $\tilde{F}(\bar{\phi}) := F(\Phi^{-1}(\bar{\phi}))$. Use $\dot{\theta}$ and $\dot{\phi}$ to denote the time derivative of $\theta$ and $\phi$ respectively. Consider SiNG in its continuous-time limit under these two parameterizations:*

$$\dot{\theta}_s = -\mathbf{H}(\theta_s)^{-1} \nabla F(\theta_s) \quad \text{and} \quad \dot{\phi}_s = -\tilde{\mathbf{H}}(\phi_s)^{-1} \nabla \tilde{F}(\phi_s) \quad \text{with} \quad \phi_0 = \Phi(\theta_0). \quad (14)$$

*Then $\theta_s$ and $\phi_s$ are related by the equation $\phi_s = \Phi(\theta_s)$ at all time $s \geqslant 0$.*

The SiNG direction is a "curved" negative gradient of the loss function $F(\theta)$ and the "curvature" is exactly given by the Sinkhorn Information Matrix (SIM), i.e. the Hessian $\mathbf{H}(\theta^t) = \nabla_\theta^2 \mathcal{S}(\alpha_\theta, \alpha_{\theta^t})_{|\theta = \theta^t}$ of the Sinkhorn divergence. An important question is whether SIM is computationally tractable. In the next section, we derive its explicit expression and describe how it can be efficiently computed. This is in sharp contrast to the Wasserstein information matrix (WIM) as in the WNG method proposed in Li and Montúfar [2018], which does not attain an explicit form for $d > 1$ ($d$ is the parameter dimension).

While computing the update direction $\mathbf{d}_t$ involves the inversion of $\mathbf{H}(\theta^t)$, it can be computed using the classical conjugate gradient algorithm, requiring only a matrix-vector product. Consequently, our Sinkhorn Natural Gradient (SiNG) admits a simple and elegant implementation based on modern auto-differential mechanisms such as PyTorch. We will elaborate this point in Appendix E.

## 5 Sinkhorn Information Matrix

In this section, we describe the explicit expression of the *Sinkhorn information matrix (SIM)* and show that it can be computed very efficiently using simple function operations (e.g. $\log$ and $\exp$) and integrals with respect to $\alpha_\theta$ (with complexity logarithmic in terms of the reciprocal of the target accuracy). The computability of SIM and hence SiNG is the key contribution of our paper. In the case when we can only compute the integration with respect to $\alpha_\theta$ in a Monte Carlo manner, an empirical estimator of SIM (eSIM) is proposed in the next section with a delicate stability analysis.

Since $\mathcal{S}(\cdot, \cdot)$ is a linear combination of terms like $\mathrm{OT}_\gamma(\cdot, \cdot)$–see (4), we can focus on the term $\nabla_\theta^2 \mathrm{OT}_\gamma(\alpha_\theta, \alpha_{\theta^t})_{|\theta = \theta^t}$ in $\mathbf{H}(\theta^t)$ and the other term $\nabla_\theta^2 \mathrm{OT}_\gamma(\alpha_\theta, \alpha_\theta)_{|\theta = \theta^t}$ can be handled similarly. Having these two terms, SIM is computed as $\mathbf{H}(\theta^t) = [\nabla_\theta^2 \mathrm{OT}_\gamma(\alpha_\theta, \alpha_{\theta^t}) + \nabla_\theta^2 \mathrm{OT}_\gamma(\alpha_\theta, \alpha_\theta)]_{|\theta = \theta^t}$.

Recall that the entropy regularized optimal transport distance $\mathrm{OT}_\gamma$ admits an equivalent dual concave-maximization form (6). Due to the concavity of $\mathcal{H}_2$ w.r.t. $g$ in (5), the corresponding optimal $g_f = \arg\max_{g \in \mathcal{C}(\mathcal{X})} \mathcal{H}_2(f, g; \alpha, \beta)$ can be explicitly computed for any fixed $f \in \mathcal{C}(\mathcal{X})$: Given a function $\bar{f} \in \mathcal{C}(\mathcal{X})$ and a measure $\alpha \in \mathcal{M}_1^+(\mathcal{X})$, define the Sinkhorn mapping as

$$\mathcal{A}(\bar{f}, \alpha)(y) := -\gamma \log \int_{\mathcal{X}} \exp\left(-\frac{1}{\gamma} c(x, y) + \frac{1}{\gamma} \bar{f}(x)\right) \mathbf{d}\alpha(x). \quad (15)$$

The first-order optimality of $g_f$ writes $g_f = \mathcal{A}(f, \alpha)$. Then, (6) can be simplified to the following problem with a single potential variable:

$$\mathrm{OT}_\gamma(\alpha_\theta, \beta) = \max_{f \in \mathcal{C}(\mathcal{X})} \left\{ \mathcal{H}_1(f, \theta) := \langle f, \alpha_\theta \rangle + \langle \mathcal{A}(f, \alpha_\theta), \beta \rangle \right\}, \quad (16)$$

where we emphasize the impact of $\theta$ to $\mathcal{H}_1$ by writing it explicitly as a variable for $\mathcal{H}_1$. Moreover, in $\mathcal{H}_1$ the dependence on $\beta$ is dropped as $\beta$ is fixed. We also denote the optimal solution to the R.H.S. of (16) by $f_\theta$ which is one of the Sinkhorn potentials for $\mathrm{OT}_\gamma(\alpha_\theta, \beta)$.

The following proposition describes the explicit expression of $\nabla_\theta^2 \mathrm{OT}_\gamma(\alpha_\theta, \alpha_{\theta^t})_{|\theta = \theta^t}$ based on the above dual representation. The proof is provided in Appendix B.1.

**Proposition 5.1.** *Recall the definition of the dual-variable function* $\mathcal{H}_1 : \mathcal{C}(\mathcal{X}) \times \Theta \to \mathbb{R}$ *in* (16) *and the definition of the second-order Fréchet derivative at the end of Section 1. For a parameterized push-forward measure* $\alpha_\theta = T_{\theta\sharp}\mu$ *and a fixed measure* $\beta \in \mathcal{M}_1^+(\mathcal{X})$, *we have*

$$\nabla_\theta^2 \mathrm{OT}_\gamma(\alpha_\theta, \beta) = -D_{11}^2 \mathcal{H}_1(f_\theta, \theta) \circ (Df_\theta, Df_\theta) + D_{22}^2 \mathcal{H}_1(f_\theta, \theta), \qquad (17)$$

*where* $Df_\theta$ *denotes the Fréchet derivative of the Sinkhorn potential* $f_\theta$ *w.r.t. the parameter* $\theta$.

**Remark 5.1** (SIM for $1d$-Gaussian)**.** *It is in general difficult to give closed form expression of the SIM. However, in the simplest case when* $\alpha_\theta$ *is a one-dimensional Gaussian distribution with a parameterized mean, i.e.* $\alpha_\theta = \mathcal{N}(\mu(\theta), \sigma^2)$, *SIM can be explicitly computed as* $\nabla_\theta^2 \mathcal{S}(\alpha_\theta, \beta) = 2\nabla_\theta^2 \mu(\theta)$ *due to the closed form expression of the entropy regularized optimal transport between Gaussian measures [Janati et al., 2020].*

Suppose that we have the Sinkhorn potential $f_\theta$ and its the Fréchet derivative $Df_\theta$. Then the terms $D_{ij}^2 \mathcal{H}_1(f, \theta), i, j = 1, 2$ can all be evaluated using a constant amount of simple function operations, e.g. $\log$ and $\exp$, since we know the explicit expression of $\mathcal{H}_1$. Consequently, it is sufficient to have estimators $f_\theta^\epsilon$ and $g_\theta^\epsilon$ of $f_\theta$ and $Df_\theta$ respectively, such that $\|f_\theta^\epsilon - f_\theta\|_\infty \leqslant \epsilon$ and $\|g_\theta^\epsilon - Df_\theta\|_{op} \leqslant \epsilon$ for an arbitrary target accuracy $\epsilon$. This is because the high accuracy approximation of $f_\theta$ and $Df_\theta$ imply the high accuracy approximation of $\nabla_\theta^2 \mathrm{OT}_\gamma(\alpha_\theta, \beta)$ due to the Lipschitz continuity of the terms $D_{ij}^2 \mathcal{H}_1(f, \theta), i, j = 1, 2$. We derive these expressions and their Lipschitz continuity in Appendix B.

For the Sinkhorn Potential $f_\theta$, its estimator $f_\theta^\epsilon$ can be efficiently computed using the Sinkhorn-Knopp algorithm Sinkhorn and Knopp [1967]. We provide more details on this in Appendix B.2.

**Proposition 5.2** (Computation of the Sinkhorn Potential $f_\theta$ – (Theorem 7.1.4 in [Lemmens and Nussbaum, 2012] and Theorem B.10 in [Luise et al., 2019])**.** *Assume that the ground cost function* $c$ *is bounded, i.e.* $0 \leqslant c(x, y) \leqslant M_c, \forall x, y \in \mathcal{X}$. *Denote* $\lambda := \frac{\exp(M_c/\gamma)-1}{\exp(M_c/\gamma)+1} < 1$ *and define*

$$\mathcal{B}(f, \theta) := \mathcal{A}\big(\mathcal{A}(f, \alpha_\theta), \beta\big). \qquad (18)$$

*Then the fixed point iteration* $f^{t+1} = \mathcal{B}\big(f^t, \theta\big)$ *converges linearly:* $\|f^{t+1} - f_\theta\|_\infty = \mathcal{O}(\lambda^t)$.

For the Fréchet derivative $Df_\theta$, we construct its estimator in the following proposition.

**Proposition 5.3** (Computation of the Fréchet derivative $Df_\theta$)**.** *Let* $f_\theta^\epsilon$ *be an approximation of* $f_\theta$ *such that* $\|f_\theta^\epsilon - f_\theta\|_\infty \leqslant \epsilon$. *Choose a large enough* $l$, *for instance* $l = \lceil \log_\lambda \frac{1}{3} \rceil/2$. *Define* $\mathcal{E}\big(f, \theta\big) = \mathcal{B}\big(\cdots \mathcal{B}(f, \theta) \cdots, \theta\big)$, *the* $l$ *times composition of* $\mathcal{B}$ *in its first variable. Then the sequence*

$$g_\theta^{t+1} = D_1 \mathcal{E}\big(f_\theta^\epsilon, \theta\big) \circ g_\theta^t + D_2 \mathcal{E}\big(f_\theta^\epsilon, \theta\big) \qquad (19)$$

*converges linearly to a* $\epsilon$-*neighborhood of* $Df_\theta$, *i.e.* $\|g_\theta^{t+1} - Df_\theta\|_{op} = \mathcal{O}(\epsilon + (\frac{2}{3})^t \|g_\theta^0 - Df_\theta\|_{op})$.

We deferred the proof to the above proposition to Appendix B.3. The high-accuracy estimators $f_\theta^\epsilon$ and $g_\theta^\epsilon$ derived in the above propositions can both be obtained using $\mathcal{O}(\log \frac{1}{\epsilon})$ function operations and integrals. With the expression of SIM and the two propositions discussing the efficient computation of $f_\theta$ and $Df_\theta$, we obtain the following theorem.

**Theorem 5.1** (Computability of SIM)**.** *For any given target accuracy* $\epsilon > 0$, *there exists an estimator* $\mathbf{H}_\epsilon(\theta)$, *such that* $\|\mathbf{H}_\epsilon(\theta) - \mathbf{H}(\theta)\|_{op} \leqslant \epsilon$, *and the estimator can be computed using* $\mathcal{O}(\log \frac{1}{\epsilon})$ *simple function operations and integrations with respect to* $\alpha_\theta$.

This result shows a significantly broader applicability of SiNG than WNG, as the latter can only be used in limited situations due to the intractability of computing WIM.

## 6 Empirical Estimator of SIM

In the previous section, we derived an explicit expression for the Sinkhorn information matrix (SIM) and described how it can be computed efficiently. In this section, we provide an empirical estimator for SIM (eSIM) in the case where the integration w.r.t. $\alpha_\theta$ can only be computed in a Monte-Carlo manner. Moreover, we prove the stability of eSIM by showing that the Fréchet derivative of the Sinkhorn potential with respect to the parameter $\theta$ is continuous with respect to the underlying measure $\mu$, which is interesting on its own.

Recall that the parameterized measure has the structure $\alpha_\theta = T_{\theta\sharp}\mu$, where $\mu \in \mathcal{M}_1^+(\mathcal{Z})$ is some probability measure on the latent space $\mathcal{Z} \subseteq \mathbb{R}^{\bar{q}}$ and $T_\theta : \mathcal{Z} \to \mathcal{X}$ is some push-forward mapping parameterized by $\theta \in \Theta$. We use $\bar{\mu}$ to denote an empirical measure of $\mu$ with $n$ Dirac measures: $\bar{\mu} = \frac{1}{n}\sum_{i=1}^n \delta_{z_i}$ with $z_i \overset{\text{iid}}{\sim} \mu$ and we use $\bar{\alpha}_\theta$ to denote the corresponding empirical measure of $\alpha_\theta$: $\bar{\alpha}_\theta = T_{\theta\sharp}\bar{\mu} = \frac{1}{n}\sum_{i=1}^n \delta_{T_\theta(z_i)}$. Based on the above definition, we propose the following empirical estimator for the Sinkhorn information matrix (eSIM)

$$\bar{\mathbf{H}}(\theta^t) = \nabla_\theta^2 \mathcal{S}(\bar{\alpha}_\theta, \bar{\alpha}_{\theta^t})_{|\theta=\theta^t}. \tag{20}$$

The following theorem shows stability of eSIM. The proof is provided in Appendix C.

**Theorem 6.1.** *Define the bounded Lipschitz metric of measures* $d_{bl} : \mathcal{M}_1^+(\mathcal{X}) \times \mathcal{M}_1^+(\mathcal{X}) \to \mathbb{R}_+$ *by*

$$d_{bl}(\alpha, \beta) := \sup_{\|\xi\|_{bl} \leqslant 1} |\langle \xi, \alpha \rangle - \langle \xi, \beta \rangle|, \tag{21}$$

*where we denote* $\|\xi\|_{bl} := \max\{\|\xi\|_\infty, \|\xi\|_{Lip}\}$ *with* $\|\xi\|_{Lip} := \max_{x,y \in \mathcal{X}} \frac{|\xi(x)-\xi(y)|}{\|x-y\|}$. *Assume that the ground cost function is bounded and Lipschitz continuous. Then*

$$\|\bar{\mathbf{H}}(\theta^t) - \mathbf{H}(\theta^t)\|_{op} = \mathcal{O}(d_{bl}(\mu, \bar{\mu})). \tag{22}$$

In the rest of this subsection, we analyze the structure of $\bar{\mathbf{H}}(\theta^t)$ and describe how it can be efficiently computed. Similar to the previous section, we focus on the term $\nabla_\theta^2 \text{OT}_\gamma(\bar{\alpha}_\theta, \beta)$ with $\bar{\alpha}_\theta = \frac{1}{n}\sum_{i=1}^n \delta_{T_\theta(z_i)}$ and $\beta = \frac{1}{n}\sum_{i=1}^n \delta_{y_i}$ for arbitrary $y_i \in \mathcal{X}$.

First, notice that the output of the Sinkhorn mapping (15) is determined solely by the function values of the input $\bar{f}$ at the support of $\alpha$. Using $\mathbf{f} = [\mathbf{f}_1, \dots, \mathbf{f}_n] \in \mathbb{R}^n$ with $\mathbf{f}_i = \bar{f}(x_i)$ to denote the value extracted from $\bar{f}$ on $\text{supp}(\bar{\alpha})$, we define for a discrete probability measures $\bar{\alpha} = \frac{1}{n}\sum_{i=1}^n \delta_{x_i}$ the discrete Sinkhorn mapping $\bar{\mathcal{A}}(\mathbf{f}, \bar{\alpha}) : \mathbb{R}^n \times \mathcal{M}_1^+(\mathcal{X}) \to \mathcal{C}(\mathcal{X})$ as

$$\bar{\mathcal{A}}(\mathbf{f}, \bar{\alpha})(y) := -\gamma \log \left( \frac{1}{n}\sum_{i=1}^n \exp\left( -\frac{1}{\gamma}c(x_i, y) + \frac{1}{\gamma}\mathbf{f}_i \right) \right) = \mathcal{A}(\bar{f}, \bar{\alpha})(y), \tag{23}$$

where the last equality should be understood as two functions being identical. Since both $\bar{\alpha}_\theta$ and $\beta$ in $\text{OT}_\gamma(\bar{\alpha}_\theta, \beta)$ are discrete, (16) can be reduced to

$$\text{OT}_\gamma(\bar{\alpha}_\theta, \beta) = \max_{\mathbf{f} \in \mathbb{R}^n} \left\{ \bar{\mathcal{H}}_1(\mathbf{f}, \theta) = \frac{1}{n}\mathbf{f}^\top \mathbf{1}_n + \frac{1}{n}\sum_{i=1}^n \bar{\mathcal{A}}(\mathbf{f}, \bar{\alpha}_\theta)(y_i) \right\}. \tag{24}$$

Now, let $\mathbf{f}_\theta$ be the solution to the above problem. We can compute the first order gradient of $\text{OT}_\gamma(\bar{\alpha}_\theta, \beta)$ with respect to $\theta$ by

$$\nabla_\theta \text{OT}_\gamma(\bar{\alpha}_\theta, \beta) = J_{\mathbf{f}_\theta}^\top \cdot \nabla_1 \bar{\mathcal{H}}_1(\mathbf{f}_\theta, \theta) + \nabla_2 \bar{\mathcal{H}}_1(\mathbf{f}_\theta, \theta). \tag{25}$$

Here $J_{\mathbf{f}_\theta} = \frac{\partial \mathbf{f}_\theta}{\partial \theta} \in \mathbb{R}^{n \times d}$ denotes the Jacobian matrix of $\mathbf{f}_\theta$ with respect to $\theta$ and $\nabla_i \bar{\mathcal{H}}_1$ denotes the gradient of $\bar{\mathcal{H}}_1$ with respect to its $i^{th}$ variable for $i = 1, 2$. Importantly, the optimality condition of $\mathbf{f}_\theta$ implies $\nabla_1 \bar{\mathcal{H}}_1(\mathbf{f}_\theta, \theta) = \mathbf{0}_n$. Further, we compute the second order gradient of $\text{OT}_\gamma(\bar{\alpha}_\theta, \beta)$ with respect to $\theta$ by (we omit the parameter $(\mathbf{f}_\theta, \theta)$ of $\bar{\mathcal{H}}_1$)

$$\nabla_\theta^2 \text{OT}_\gamma(\bar{\alpha}_\theta, \beta) = T_{\mathbf{f}_\theta} \times_1 \nabla_1 \bar{\mathcal{H}}_1 + J_{\mathbf{f}_\theta}^\top \cdot \nabla_{11} \bar{\mathcal{H}}_1 \cdot J_{\mathbf{f}_\theta} + J_{\mathbf{f}_\theta}^\top \cdot \nabla_{12} \bar{\mathcal{H}}_1 + \nabla_{21} \bar{\mathcal{H}}_1^\top \cdot J_{\mathbf{f}_\theta} + \nabla_{22} \bar{\mathcal{H}}_1, \tag{26}$$

where $T_{\mathbf{f}_\theta} = \frac{\partial^2 \mathbf{f}_\theta}{\partial \theta^2} \in \mathbb{R}^{n \times d \times d}$ is a tensor denoting the second-order Jacobian matrix of $\mathbf{f}_\theta$ with respect to $\theta$ and $\times_1$ denotes the tensor product along its first dimension. Using the fact that $\nabla_1 \bar{\mathcal{H}}_1(\mathbf{f}_\theta, \theta) = \mathbf{0}_n$, we drop the first term and simplify $\nabla_\theta^2 \text{OT}_\gamma(\bar{\alpha}_\theta, \beta)$ to (again we omit the parameter $(\mathbf{f}_\theta, \theta)$ of $\bar{\mathcal{H}}_1$)

$$\nabla_\theta^2 \text{OT}_\gamma(\bar{\alpha}_\theta, \beta) = J_{\mathbf{f}_\theta}^\top \cdot \nabla_{11} \bar{\mathcal{H}}_1 \cdot J_{\mathbf{f}_\theta} + J_{\mathbf{f}_\theta}^\top \cdot \nabla_{12} \bar{\mathcal{H}}_1 + \nabla_{21} \bar{\mathcal{H}}_1^\top \cdot J_{\mathbf{f}_\theta} + \nabla_{22} \bar{\mathcal{H}}_1. \tag{27}$$

As we have the explicit expression of $\bar{\mathcal{H}}_1$, we can explicitly compute $\nabla_{ij} \bar{\mathcal{H}}_1$ given that we have the Sinkhorn potential $\mathbf{f}_\theta$. Further, if we can compute $J_{\mathbf{f}_\theta}$, we are then able to compute $\nabla_\theta^2 \text{OT}_\gamma(\bar{\alpha}_\theta, \beta)$. The following propositions can be viewed as discrete counterparts of Proposition 5.2 and Proposition 5.3 respectively. Both $\mathbf{f}_\theta$ and $J_{\mathbf{f}_\theta}$ can be well-approximated using a number of finite dimensional vector/matrix operations which is logarithmic in the desired accuracy. Besides, given these two quantities, one can easily check that $\nabla_{ij} \bar{\mathcal{H}}_1$ can be evaluated within $\mathcal{O}((n+d)^2)$ arithmetic operations. Consequently, we can compute an $\epsilon$-accurate approximation of eSIM in time $\mathcal{O}((n+d)^2 \log \frac{1}{\epsilon})$.

**Proposition 6.1** (Computation of the Sinkhorn Potential $\mathbf{f}_\theta$). *Assume that the ground cost function $c$ is bounded, i.e. $0 \leqslant c(x,y) \leqslant M_c, \forall x,y \in \mathcal{X}$. Denote $\lambda := \frac{\exp(M_c/\gamma) - 1}{\exp(M_c/\gamma) + 1} < 1$ and define*

$$\bar{\mathcal{B}}(\mathbf{f}, \theta) := \bar{\mathcal{A}}(\mathbf{g}, \beta) \text{ with } \mathbf{g} = [\bar{\mathcal{A}}(\mathbf{f}, \bar{\alpha}_\theta)(y_1), \dots, \bar{\mathcal{A}}(\mathbf{f}, \bar{\alpha}_\theta)(y_n)] \in \mathbb{R}^n. \tag{28}$$

*Then the fixed point iteration $\mathbf{f}^{t+1} = \bar{\mathcal{B}}(\mathbf{f}^t, \theta)$ converges linearly: $\|\mathbf{f}^{t+1} - \mathbf{f}_\theta\|_\infty = \mathcal{O}(\lambda^t)$*

**Proposition 6.2** (Computation of the Jacobian $J_{\mathbf{f}_\theta}$). *Let $\mathbf{f}_\epsilon$ be an approximation of $\mathbf{f}_\theta$ such that $\|\mathbf{f}_\epsilon - \mathbf{f}_\theta\|_\infty \leqslant \epsilon$. Pick $l = \lceil \log_\lambda \frac{1}{3} \rceil / 2$. Define $\bar{\mathcal{E}}(\mathbf{f}, \theta) = \bar{\mathcal{B}}(\cdots \bar{\mathcal{B}}(\mathbf{f}, \theta) \cdots, \theta)$, the $l$ times composition of $\bar{\mathcal{B}}$ in its first variable. Then the sequence of matrices*

$$\mathbf{J}^{t+1} = J_1 \bar{\mathcal{E}}(\mathbf{f}_\epsilon, \theta) \cdot \mathbf{J}^t + J_2 \bar{\mathcal{E}}(\mathbf{f}_\epsilon, \theta), \tag{29}$$

*converges linearly to an $\epsilon$ neighbor of $J_{\mathbf{f}_\theta}$: $\|\mathbf{J}^{t+1} - J_{\mathbf{f}_\theta}\|_{op} = \mathcal{O}(\epsilon + (\frac{2}{3})^t \|\mathbf{J}^0 - J_{\mathbf{f}_\theta}\|_{op})$. Here $J_i \bar{\mathcal{E}}$ denotes the Jacobian matrix of $\bar{\mathcal{E}}$ with respect to its $i^{th}$ variable.*

The SiNG direction $\mathbf{d}_t$ involves the inversion of $\bar{\mathbf{H}}(\theta^t)$. This can be (approximately) computed using the classical conjugate gradient (CG) algorithm, using only matrix-vector products. Combining eSIM and CG, we describe a simple and elegant PyTorch-based implementation for SiNG in Appendix E,

## 7 Experiment

In this section, we compare SiNG with other SGD-type solvers by training generative models. We did not compare with WNG Li and Montúfar [2018] since WNG can only be implemented for the case where the parameter dimension $d$ is $1$. We also tried to implement KWNG Arbel et al. [2019], which however diverges in our setting. In particular, we encounter the case when the KWNG direction has negative inner product with the euclidean gradient direction, leading to its divergence. As we discussed in the related work, the gap between KWNG and WNG cannot be quantified with reasonable assumptions, which explains our observation. In all the following experiments, we pick the push-forward map $T_\theta$ to be the generator network in DC-GAN [Radford et al., 2015]. For more detailed experiment settings, please see Appendix D.

### 7.1 Squared-$\ell_2$-norm as Ground Metric

We first consider the distribution matching problem, where our goal is to minimize the Sinkhorn divergence between the parameterized generative model $\alpha_\theta = T_{\theta\sharp}\mu$ and a given target distribution $\beta$,

$$\min_{\theta \in \Theta} F(\theta) = \mathcal{S}(\alpha_\theta, \beta). \tag{30}$$

Here, $T_\theta$ is a neural network describing the push-forward map with its parameter summarized in $\theta$ and $\mu$ is a zero-mean isometric Gaussian distribution. In particular, the metric on the ground set $\mathcal{X}$ is set to the vanilla squared-$\ell_2$ norm, i.e. $c(x,y) = \|x - y\|^2$ for $x,y \in \mathcal{X}$. Our experiment considers a specific instance of problem (30) where we take the measure $\beta$ to be the distribution of the images in the CelebA dataset. We present the comparison of the generator loss (the objective value) vs time plot in right figure. The entropy regularization parameter $\gamma$ is set to $0.01$ for both the objective and the constraint. We can see that SiNG is much more efficient at reducing the objective value than ADAM given the same amount of time.

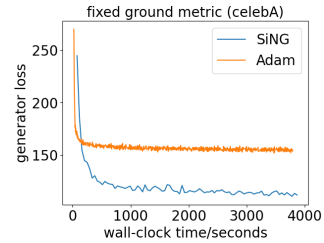

### 7.2 Squared-$\ell_2$-norm with an Additional Encoder as Ground Metric

We then consider a special case of problem (7), where the metric on the ground set $\mathcal{X}$ is set to squared-$\ell_2$-norm with a *fixed* parameterized encoder (i.e. we fix the variable $\xi$ in the max part of (7)): $c_\xi(x,y) = \|\phi_\xi(x) - \phi_\xi(y)\|^2$. Here $\phi_\xi(\cdot) : \mathcal{X} \to \mathbb{R}^{\hat{q}}$ is a neural network encoder that outputs an embedding of the input in a high dimensional space ($\hat{q} > q$, where we recall $q$ is the dimension of the ground set $\mathcal{X}$). In particular, we set $\phi_\xi(\cdot)$ to be the discriminator network in DC-GAN without the last classification layer [Radford et al., 2015]. Two specific instances are considered: we take the measure $\beta$ to be the distribution of the images in either the CelebA or the Cifar10 dataset. The

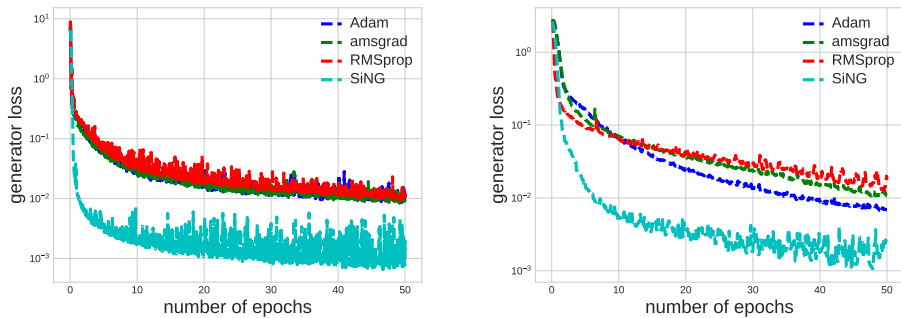

Figure 1: Generator losses on CelebA (left) and Cifar10 (right).

parameter $\xi$ of the encoder $\phi$ is obtained in the following way: we first use SiNG to train a generative model by alternatively taking a SiNG step on $\theta$ and taking an SGD step on $\xi$. After sufficiently many iterations (when the generated image looks real or specifically 50 epochs), we fix the encoder $\phi_\xi$. We then set the objective functional (1) to be $\mathcal{F}(\alpha_\theta) = \mathcal{S}_{c_\xi}(\alpha_\theta, \beta)$ (see (7)), and compare SiNG and SGD-type algorithms in the minimization of $\mathcal{F}$ under a consensus random initialization. We report the comparison in Figure 1, where we observe the significant improvement from SiNG in both accuracy and efficiency. Such phenomenon is due to the fact that SiNG is able to use geometry information by considering SIM while other method does not. Moreover, the pretrained ground cost $c_\xi$ may capture some non-trivial metric structure of the images and consequently geometry-faithfully method like our SiNG can thus do better.

## 7.3 Training GAN with SiNG

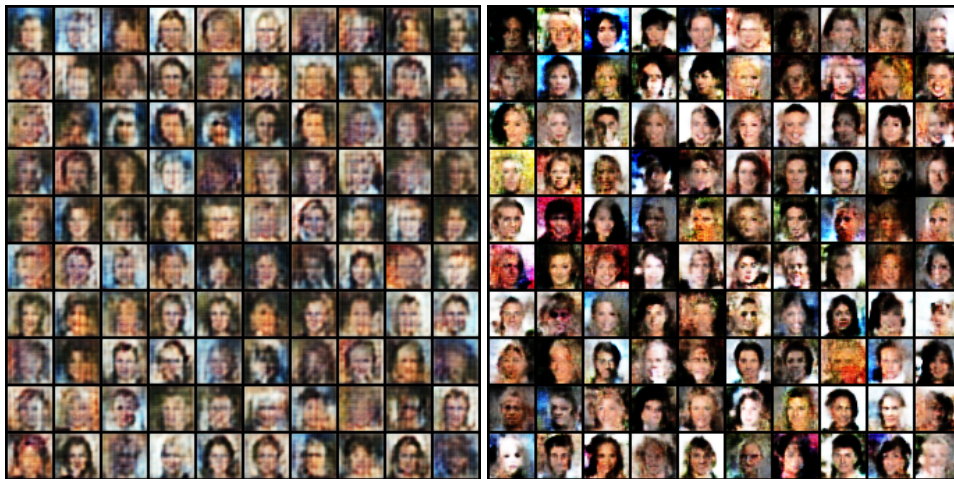

Figure 2: Comparison of the visual quality of the images generated by Adam (left) and SiNG (right).

Finally, we showcase the the advantage of training a GAN model using SiNG over SGD-based solvers. Specifically, we consider the GAN model (7). The entropy regularization of the Sinkhorn divergence objective is set to $\gamma = 100$ as suggested in Table 2 of [Genevay et al., 2018]. The regularization for the constraint is set to $\gamma = 1$ in SiNG. We used ADAM as the optimizer for the discriminators (with step size $10^{-3}$ and batch size 4000). The result is reported in Figure 2. We can see that the images generated using SiNG are much more vivid than the ones obtained using SGD-based optimizers. We remark that our main goal has been to showcase that SiNG is more efficient in reducing the objective value compared to SGD-based solvers, and hence, we have used a relatively simpler DC-GAN type generator and discriminator (details given in the supplementary materials). If more sophisticated ResNet type generators and discriminators are used, the image quality can be further improved.

## 8 Broader Impact

We propose the Sinkhorn natural gradient (SiNG) algorithm for minimizing an objective functional over a parameterized family of generative-model type measures. While our results do not immediately lead to broader societal impacts (as they are mostly theoretical), they can lead to new potential positive impacts. SiNG admits explicit update rule which can be efficiently carried out in an exact manner under both continuous and discrete settings. Being able to exploit the geometric information provided in the Sinkhorn information matrix, we observe the remarkable advantage of SiNG over existing state-of-the-art SGD-type solvers. Such algorithm is readily applicable to many types of existing generative adversarial models and possibly helps the development of the literature.

## Acknowledgment

This work is supported by NSF CPS-1837253.

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
