[Supplementary Material]

# A   Appendix Section for Methodology

## A.1   Proof of Proposition 4.1

Denote the Lagrangian function by

$$G_\lambda(\Delta\theta) = F(\theta^t + \Delta\theta) + \lambda\left(\mathcal{S}(\alpha_{\theta^t + \Delta\theta}, \alpha_{\theta^t}) - \epsilon - \epsilon^{c_2}\right). \tag{31}$$

We have the following inequality which characterize a lower bound of the solution to (9) (recall that $1 < c_2 < 1.5$, $c_1 < 0.5$ and $3c_1 - 1 \geqslant c_2$),

$$
\begin{aligned}
\min_{\Delta\theta \in \mathbb{R}^d} \; & F(\theta^t + \Delta\theta) \\
\text{s.t. } & \|\Delta\theta\| \leqslant \epsilon^{c_1} \\
& \mathcal{S}(\alpha_{\theta^t + \Delta\theta}, \alpha_{\theta^t}) \leqslant \epsilon + \epsilon^{c_2}
\end{aligned}
\quad = \min_{\|\Delta\theta\| \leqslant \epsilon^{c_1}} \max_{\lambda \geqslant 0} G_\lambda(\Delta\theta) \geqslant \max_{\lambda \geqslant 0} \min_{\|\Delta\theta\| \leqslant \epsilon^{c_1}} G_\lambda(\Delta\theta). \tag{32}
$$

We now focus on the R.H.S. of the above inequality. Denote the second-order Taylor expansion of the Lagrangian $G_\lambda$ by $\bar{G}_\lambda$:

$$\bar{G}_\lambda(\Delta\theta) = F(\theta^t) + \langle \nabla_\theta F(\theta^t), \Delta\theta \rangle + \frac{1}{2}\langle \nabla_\theta^2 F(\theta^t)\Delta\theta, \Delta\theta \rangle + \frac{\lambda}{2}\langle \mathbf{H}(\theta^t)\Delta\theta, \Delta\theta \rangle - \lambda\epsilon - \lambda\epsilon^{c_2},$$

where we used the optimality condition (10) of $\mathcal{S}(\alpha, \alpha^t)$ so that the first-order term of $\mathcal{S}(\alpha, \alpha^t)$ vanishes. Besides, $\mathbf{H}(\theta)$ is defined in (10). The error of such approximation can be bounded as

$$G_\lambda(\Delta\theta) - \bar{G}_\lambda(\Delta\theta) = \mathcal{O}((\lambda + 1)\|\Delta\theta\|^3). \tag{33}$$

Further, for any fixed $\lambda$, denote $\Delta\theta_\lambda^* = \operatorname{argmin}_{\|\Delta\theta\| \leqslant \epsilon^{c_1}} G_\lambda(\Delta\theta)$.

We can then derive the following lower bound on the minimization subproblem of the R.H.S. of (32):

$$
\begin{aligned}
\max_{\lambda \geqslant 0} \min_{\|\Delta\theta\| \leqslant \epsilon^{c_1}} G_\lambda(\Delta\theta) &= \max_{\lambda \geqslant 0} \bar{G}_\lambda(\Delta\theta_\lambda^*) - \mathcal{O}((\lambda + 1)\|\Delta\theta_\lambda^*\|^3) \\
&\geqslant \max_{\lambda \geqslant 0} \bar{G}_\lambda(\Delta\theta_\lambda^*) - \mathcal{O}((\lambda + 1)\epsilon^{3c_1}) \\
&\geqslant \max_{\lambda \geqslant 0} \min_{\|\Delta\theta\| \leqslant \epsilon^{c_1}} \bar{G}_\lambda(\Delta\theta) - \mathcal{O}((\lambda + 1)\epsilon^{3c_1}),
\end{aligned}
$$

Note that for sufficiently large $\lambda$, $\mathbf{H}(\theta^t) + \frac{1}{\lambda}\nabla_\theta^2 F(\theta^t) > 0$ by recalling the positive definiteness of $\mathbf{H}(\theta^t)$. In this case, as a convex program, $\min_{\|\Delta\theta\| \leqslant \epsilon^{c_1}} \bar{G}_\lambda(\Delta\theta)$ admits the closed form solution: Denote $\overline{\Delta\theta_\lambda^*} = \operatorname{argmin} \bar{G}_\lambda(\Delta\theta)$. We have

$$\overline{\Delta\theta_\lambda^*} = -\frac{1}{\lambda}\left(\mathbf{H}(\theta^t) + \frac{1}{\lambda}\nabla_\theta^2 F(\theta^t)\right)^{-1}\nabla_\theta F(\theta^t) \text{ and } \bar{G}(\overline{\Delta\theta_\lambda^*}) = F(\theta^t) - \frac{\bar{a}}{2\lambda} - \lambda\epsilon - \lambda\epsilon^{c_2}, \tag{34}$$

where we denote $\bar{a} := \langle\left[\mathbf{H}(\theta^t) + \frac{1}{\lambda}\nabla_\theta^2 F(\theta^t)\right]^{-1}\nabla_\theta F(\theta^t), \nabla_\theta F(\theta^t)\rangle > 0$.

For sufficiently small $\epsilon$, by taking $\lambda = \sqrt{\frac{a}{2\epsilon}}$ with $a := \langle[\mathbf{H}(\theta^t)]^{-1}\nabla_\theta F(\theta^t), \nabla_\theta F(\theta^t)\rangle > 0$ (note that $\|\overline{\Delta\theta_\lambda^*}\| = \mathcal{O}(\sqrt{\epsilon}) < \epsilon^{c_1}$ and is hence feasible for $c_1 < 0.5$), the R.H.S. of (32) has the following lower bound (recall that we have $3c_1 - 1 \geqslant c_2$)

$$\max_{\lambda \geqslant 0} \min_{\|\Delta\theta\| \leqslant \epsilon^{c_1}} G_\lambda(\Delta\theta) \geqslant F(\theta^t) - \left(\frac{\bar{a}}{\sqrt{2a}} + \sqrt{\frac{a}{2}}\right)\sqrt{\epsilon} - \mathcal{O}(\epsilon^{c_2 - 0.5}). \tag{35}$$

This result leads to the following lower bound on (9):

$$\lim_{\epsilon \to 0} \frac{F(\theta^t + \Delta\theta_\epsilon^t) - F(\theta^t)}{\sqrt{\epsilon}} \geqslant -\sqrt{2\langle[\mathbf{H}(\theta^t)]^{-1}\nabla_\theta F(\theta^t), \nabla_\theta F(\theta^t)\rangle}, \tag{36}$$

where $\Delta\theta_\epsilon^t$ is the solution to (9). Finally, observe that the equality is achieved by taking $\Delta\theta_\epsilon^t = -\frac{\sqrt{2\epsilon}(\mathbf{H}(\theta^t))^{-1}\nabla_\theta F(\theta^t)}{\sqrt{\langle[\mathbf{H}(\theta^t)]^{-1}\nabla_\theta F(\theta^t), \nabla_\theta F(\theta^t)\rangle}}$:

$$\lim_{\epsilon \to 0} \frac{F(\theta^t + \Delta\theta_\epsilon^t) - F(\theta^t)}{\sqrt{\epsilon}} = \lim_{\epsilon \to 0} \frac{1}{\sqrt{\epsilon}}\langle\nabla F(\theta^t), \Delta\theta_\epsilon^t\rangle = -\sqrt{2\langle[\mathbf{H}(\theta^t)]^{-1}\nabla_\theta F(\theta^t), \nabla_\theta F(\theta^t)\rangle}, \tag{37}$$

and $\Delta\theta_\epsilon^t$ is feasible for sufficiently small $\epsilon$ (note that we have $\frac{1}{2}\langle\mathbf{H}(\theta^t)\Delta\theta_\epsilon^t, \Delta\theta_\epsilon^t\rangle = \epsilon$):

$$\mathcal{S}(\alpha_{\theta^t + \Delta\theta_\epsilon^t}, \alpha_{\theta^t}) \leqslant \frac{1}{2}\langle\mathbf{H}(\theta^t)\Delta\theta_\epsilon^t, \Delta\theta_\epsilon^t\rangle + \mathcal{O}(\epsilon^{1.5}) < \epsilon + \epsilon^{c_2}, \tag{38}$$

and $\|\Delta\theta_\epsilon^t\| = \mathcal{O}(\sqrt{\epsilon}) < \epsilon^{c_1}$ for $c_1 < 0.5$. This leads to our conclusion.

## A.2 Proof of Proposition 4.2

Our goal is to show that the continuous-time limit of $\Phi(\theta_s)$ satisfies the same differential equation as $\phi_s$ provided that $\Phi(\theta_0) = \phi_0$. To do so, first compute the differential equation of $\Phi(\theta_s)$

$$\frac{\partial \Phi(\theta_s)}{\partial s} = \nabla_\theta \Phi(\theta_s)\dot{\theta}_s = -\nabla_\theta \Phi(\theta_s)\mathbf{H}(\theta_s)^{-1}\nabla F(\theta_s), \tag{39}$$

where $\nabla_\theta \Phi(\theta_s)$ is the Jacobian matrix of $\Phi(\theta)$ w.r.t. $\theta$ at $\theta = \theta_s$. We then compute the differential equation of $\phi_s$ (note that $\nabla_\phi \Phi^{-1}(\phi_s)$ is the Jacobian matrix of $\Phi^{-1}(\phi)$ w.r.t. $\phi$ at $\phi = \phi_s$)

$$\begin{aligned}
\dot{\phi}_s &= -\left[\nabla_\phi^2 \mathcal{S}(\alpha_{\Phi^{-1}(\phi)}, \alpha_{\Phi^{-1}(\phi_s)})_{|\phi=\phi_s}\right]^{-1} \nabla_\phi F(\Phi^{-1}(\phi))_{|\phi=\phi_s} \\
&= -\left[\nabla_\phi \Phi^{-1}(\phi_s)^\top \nabla_\theta^2 \mathcal{S}(\alpha_\theta, \alpha_{\theta^s})_{|\theta=\theta_s}\nabla_\phi \Phi^{-1}(\phi_s)\right]^{-1} \nabla_\phi \Phi^{-1}(\phi_s)^\top \nabla F(\theta)_{|\theta=\theta_s} \tag{40} \\
&= -\left[\nabla_\phi \Phi^{-1}(\phi_s)\right]^{-1} \nabla_\theta^2 \mathcal{S}(\alpha_\theta, \alpha_{\theta^s})_{|\theta=\theta_s}\nabla F(\theta)_{|\theta=\theta_s} \\
&= -\nabla_\theta \Phi(\theta_s)\mathbf{H}(\theta_s)^{-1}\nabla F(\theta_s) \tag{41} \\
&= \frac{\partial \Phi(\theta_s)}{\partial s}.
\end{aligned}$$

Here we use the following lemma in (40). We use $\Phi^{-1}(\phi_s) = \theta_s$ and the inverse function theorem $\nabla_\theta \Phi(\theta_s) = \left[\nabla_\phi \Phi^{-1}(\phi_s)\right]^{-1}$ in (41).

**Lemma A.1.**

$$\nabla_\phi^2 \mathcal{S}(\alpha_{\Phi^{-1}(\phi)}, \alpha_{\Phi^{-1}(\phi_s)})_{|\phi=\phi_s} = \nabla_\phi \Phi^{-1}(\phi_s)^\top \nabla_\theta^2 \mathcal{S}(\alpha_\theta, \alpha_{\theta^s})_{|\theta=\theta_s}\nabla_\phi \Phi^{-1}(\phi_s) \tag{42}$$

*Proof.* This lemma can be proved with simple computations. We compute only for the terms in $\nabla_\theta^2 \mathrm{OT}_\gamma(\alpha_\theta, \alpha_{\theta^s})$ as example. The terms in $\nabla_\theta^2 \mathrm{OT}_\gamma(\alpha_\theta, \alpha_\theta)$ can be computed similarly. Recall the expression

$$\begin{aligned}
\nabla_\theta^2 \mathrm{OT}_\gamma(\alpha_\theta, \beta) = \quad &D_{11}^2 \mathcal{H}_1(f_\theta, \theta) \circ (Df_\theta, Df_\theta) + \quad D_{12}^2 \mathcal{H}_1(f_\theta, \theta) \circ (Df_\theta, \mathcal{I}_d) \\
&+ D_{21}^2 \mathcal{H}_1(f_\theta, \theta) \circ (\mathcal{I}_d, Df_\theta) + \quad D_{22}^2 \mathcal{H}_1(f_\theta, \theta) \circ (\mathcal{I}_d, \mathcal{I}_d).
\end{aligned} \tag{43}$$

We compute

$$\begin{aligned}
\nabla_\phi^2 \mathrm{OT}_\gamma(\alpha_{\Phi^{-1}(\phi)}, \beta) = \; &D_{11}^2 \mathcal{H}_1(f_{\Phi^{-1}(\phi)}, \Phi^{-1}(\phi)) \circ (Df_{\Phi^{-1}(\phi)} \circ J_{\Phi^{-1}}(\phi), Df_{\Phi^{-1}(\phi)} \circ J_{\Phi^{-1}}(\phi)) \\
&+ D_{12}^2 \mathcal{H}_1(f_{\Phi^{-1}(\phi)}, \Phi^{-1}(\phi)) \circ (Df_{\Phi^{-1}(\phi)} \circ J_{\Phi^{-1}}(\phi), J_{\Phi^{-1}}(\phi)) \\
&+ D_{21}^2 \mathcal{H}_1(f_{\Phi^{-1}(\phi)}, \Phi^{-1}(\phi)) \circ (J_{\Phi^{-1}}(\phi), Df_{\Phi^{-1}(\phi)} \circ J_{\Phi^{-1}}(\phi)) \\
&+ D_{22}^2 \mathcal{H}_1(f_{\Phi^{-1}(\phi)}, \Phi^{-1}(\phi)) \circ (J_{\Phi^{-1}}(\phi), J_{\Phi^{-1}}(\phi)).
\end{aligned} \tag{44}$$

Plugging $\Phi^{-1}(\phi_s) = \theta_s$ to the above equality, we have

$$\nabla_\phi^2 \mathrm{OT}_\gamma(\alpha_{\Phi^{-1}(\phi)}, \beta)_{|\phi=\phi_s} = \nabla_\phi \Phi^{-1}(\phi_s)^\top \nabla_\theta^2 \mathrm{OT}_\gamma(\alpha_\theta, \beta)_{|\theta=\theta_s}\nabla_\phi \Phi^{-1}(\phi_s). \tag{45}$$

$\square$

## B   Appendix on SIM

### B.1   Proof of Proposition 5.1

We will derive the explicit expression of $\nabla_\theta^2 \mathrm{OT}_\gamma(\alpha_\theta, \alpha_{\theta^t})_{|\theta=\theta^t}$ based on the dual representation (16). Recall the definition of the Fréchet derivative in Definition 2 and its chain rule $D(f \circ g)(x) = Df(g(x)) \circ Dg(x)$. We compute the first-order gradient by

$$\nabla_\theta \mathrm{OT}_\gamma(\alpha_\theta, \beta) = \nabla_\theta \mathcal{H}_1(f_\theta, \theta) = \underbrace{D_1 \mathcal{H}_1(f_\theta, \theta) \circ Df_\theta}_{\mathcal{G}_1(f_\theta, \theta)} + \underbrace{D_2 \mathcal{H}_1(f_\theta, \theta)}_{\mathcal{G}_2(f_\theta, \theta)}, \tag{46}$$

where $D_i \mathcal{H}_1$ denote the Fréchet derivative of $\mathcal{H}_1$ with respect to its $i^{th}$ variable. Importantly, the optimality condition of (16) implies that $D_1 \mathcal{H}_1(f_\theta, \theta)[g] = 0, \forall g \in \mathcal{C}(\mathcal{X})$.
Further, in order to compute the second order gradient of $\mathrm{OT}_\gamma(\alpha_\theta, \beta)$ with respect to $\theta$, we first compute the gradient of $\mathcal{G}_i, i = 1, 2$:

$$\nabla_\theta \mathcal{G}_1(f_\theta, \theta) = D_1 \mathcal{H}_1(f_\theta, \theta) \circ D^2 f_\theta + D_{11}^2 \mathcal{H}_1(f_\theta, \theta) \circ (Df_\theta, Df_\theta) + D_{12}^2 \mathcal{H}_2(f_\theta, \theta) \circ (Df_\theta, \mathcal{I}_d), \tag{47}$$

$$\nabla_\theta \mathcal{G}_2(f_\theta, \theta) = D_{21}^2 \mathcal{H}_1(f_\theta, \theta) \circ (\mathcal{I}_d, Df_\theta) + D_{22}^2 \mathcal{H}_1(f_\theta, \theta) \circ (\mathcal{I}_d, \mathcal{I}_d). \tag{48}$$

Using the fact that $D_1 \mathcal{H}_1(f_\theta, \theta)[g] = 0, \forall g \in \mathcal{C}(\mathcal{X})$, we can drop the first term in the R.H.S. of (47). Combining the above results, we have

$$\begin{aligned}\nabla_\theta^2 \mathrm{OT}_\gamma(\alpha_\theta, \beta) = & \quad D_{11}^2 \mathcal{H}_1(f_\theta, \theta) \circ (Df_\theta, Df_\theta) + \quad D_{12}^2 \mathcal{H}_1(f_\theta, \theta) \circ (Df_\theta, \mathcal{I}_d) \\ & + D_{21}^2 \mathcal{H}_1(f_\theta, \theta) \circ (\mathcal{I}_d, Df_\theta) + \quad D_{22}^2 \mathcal{H}_1(f_\theta, \theta) \circ (\mathcal{I}_d, \mathcal{I}_d).\end{aligned} \tag{49}$$

Moreover, we can further simplify the above expression by noting that for any $g \in T(\mathbb{R}^d, \mathcal{C}(\mathcal{X}))$, i.e. any bounded linear operators from $\mathbb{R}^d$ to $\mathcal{C}(\mathcal{X})$,

$$\nabla_\theta \left( D_1 \mathcal{H}_1(f_\theta, \theta) \circ g \right) = D_{11}^2 \mathcal{H}_1(f_\theta, \theta) \circ (g, Df_\theta) + D_{12}^2 \mathcal{H}_1(f_\theta, \theta) \circ (g, \mathcal{I}_d) = 0. \tag{50}$$

Plugging in $g = Df_\theta$ in the above equality we have

$$D_{11}^2 \mathcal{H}_1(f_\theta, \theta) \circ (Df_\theta, Df_\theta) = -D_{12}^2 \mathcal{H}_1(f_\theta, \theta) \circ (Df_\theta, \mathcal{I}_d). \tag{51}$$

Consequently we derive (we omit the identity operator $(\mathcal{I}_d, \mathcal{I}_d)$ for the second term)

$$\nabla_\theta^2 \mathrm{OT}_\gamma(\alpha_\theta, \beta) = -D_{11}^2 \mathcal{H}_1(f_\theta, \theta) \circ (Df_\theta, Df_\theta) + D_{22}^2 \mathcal{H}_1(f_\theta, \theta), \tag{52}$$

where we note that $D_{12}^2 \mathcal{H}_1(f_\theta, \theta) \circ (Df_\theta, \mathcal{I}_d)$ is symmetric from (51) and

$$D_{21}^2 \mathcal{H}_1(f_\theta, \theta) \circ (\mathcal{I}_d, Df_\theta) = \left[ D_{12}^2 \mathcal{H}_1(f_\theta, \theta) \circ (Df_\theta, \mathcal{I}_d) \right]^\top = D_{12}^2 \mathcal{H}_1(f_\theta, \theta) \circ (Df_\theta, \mathcal{I}_d). \tag{53}$$

These two terms can be computed explicitly and involve only simple function operations like $\exp$ and $\log$ and integration with respect to $\alpha_\theta$ and $\beta$, as discussed in the following.

### B.1.1   Explicit Expression of $\nabla_\theta^2 \mathrm{OT}_\gamma(\alpha_\theta, \beta)$

Denote $\mathbf{A}_1 = D_{11}^2 \mathcal{H}_1(f_\theta, \theta) \circ (Df_\theta, Df_\theta)$ as the first term of (52). We note that $\mathbf{A}_1 \in \mathbb{R}^{d \times d}$ is a matrix and hence is a bilinear operator. If we can compute $h_1^\top \mathbf{A}_1 h_2$ for any two directions $h_1, h_2 \in \mathbb{R}^d$, we are able to compute entries of $\mathbf{A}_1$ by taking $h_1$ and $h_2$ to be the canonical bases. We compute this quantity $h_1^\top \mathbf{A}_1 h_2$ as follows.

For a fixed $y \in \mathcal{X}$, denote $\mathcal{T}_y : \mathcal{X} \times \mathcal{C}(\mathcal{X}) \to \mathbb{R}$ by

$$\mathcal{T}_y(x, f) := \exp(-c(x, y)/\gamma) \exp(f(x)/\gamma).$$

Denote $g_1 = Df_\theta[h_1] \in \mathcal{C}(\mathcal{X})$ for some direction $h_1 \in \mathbb{R}^d$ (recall that $Df_\theta \in T(\mathbb{R}^d, \mathcal{C}(\mathcal{X}))$, where $T(V, W)$ is the family of bounded linear operators from set $V$ to set $W$). Use the chain rule of Fréchet derivative to compute

$$\left( D_1 \mathcal{A}(f, \alpha_\theta)[g_1] \right)(y) = -\frac{\int_\mathcal{X} \mathcal{T}_y(x, f) g_1(x) \mathrm{d}\alpha_\theta(x)}{\int_\mathcal{X} \mathcal{T}_y(x, f) \mathrm{d}\alpha_\theta(x)}. \tag{54}$$

Let $h_2 \in \mathbb{R}^d$ be another direction and denote $g_2 = Df_\theta[h_2] \in \mathcal{C}(\mathcal{X})$. We compute

$$
\begin{aligned}
&\left(D_{11}^2 \mathcal{A}(f, \alpha_\theta)[g_1, g_2]\right)(y) \\
&= \frac{\int_{\mathcal{X}} \mathcal{T}_y(x, f) g_1(x) g_2(x) \mathbf{d}\alpha_\theta(x)}{\gamma \int_{\mathcal{X}} \mathcal{T}_y(x, f) \mathbf{d}\alpha_\theta(x)} - \frac{\int_{\mathcal{X}^2} \mathcal{T}_y(x, f) \mathcal{T}_y(x', f) g_1(x) g_2(x') \mathbf{d}\alpha_\theta(x) \mathbf{d}\alpha_\theta(x')}{\gamma \left[\int_{\mathcal{X}} \mathcal{T}_y(x, f) \mathbf{d}\alpha_\theta(x)\right]^2}.
\end{aligned}
\tag{55}
$$

Moreover, for any two directions $h_1, h_2 \in \mathbb{R}^d$, we compute $D_{11}^2 \mathcal{H}_1(f, \theta)\left[Df_\theta[h_1], Df_\theta[h_2]\right]$ by

$$
D_{11}^2 \mathcal{H}_1(f, \theta)\left[Df_\theta[h_1], Df_\theta[h_2]\right] = \int_{\mathcal{X}} \left(D_{11}^2 \mathcal{A}(f_\theta, \alpha_\theta)\left[Df_\theta[h_1], Df_\theta[h_2]\right]\right)(y) \mathbf{d}\beta(y),
\tag{56}
$$

which by plugging in (55) yields closed a form expression with only simple function operations like $\exp$ and $\log$ and integration with respect to $\alpha_\theta$ and $\beta$.

We then compute the second term of (52). Using the change-of-variable formula, we have

$$
\mathcal{A}(f, T_{\theta\sharp}\mu)(y) = -\gamma \log \int_{\mathcal{Z}} \exp\left(-\frac{1}{\gamma} c(T_\theta(z), y) + \frac{1}{\gamma} f(T_\theta(z))\right) \mathbf{d}\mu(z).
\tag{57}
$$

For any $f \in \mathcal{C}(\mathcal{X})$, the first-order Fréchet derivative of $\mathcal{H}_1(f, \theta)$ w.r.t. its second variable is given by

$$
\begin{aligned}
D_2 \mathcal{H}_1(f, \theta) = &\int_{\mathcal{Z}} \langle \nabla_\theta T_\theta(z), \nabla f(T_\theta(z)) \rangle \mathbf{d}\mu(z) \\
&+ \int_{\mathcal{X}} \frac{\int_{\mathcal{Z}} \mathcal{T}_y(T_\theta(z), f) \langle \nabla_\theta T_\theta(z), \nabla_1 c(T_\theta(z), y) - \nabla f(T_\theta(z)) \rangle \mathbf{d}\mu(z)}{\int_{\mathcal{Z}} \mathcal{T}_y(T_\theta(z), f) \mathbf{d}\mu(z)} \mathbf{d}\beta(y).
\end{aligned}
$$

Denote $u_z(\theta, f) = \nabla_1 c(T_\theta(z), y) - \nabla f(T_\theta(z))$. The second-order Fréchet derivative is given by

$$
\begin{aligned}
&D_{22}^2 \mathcal{H}_1(f, \theta) \tag{58} \\
&= \int_{\mathcal{Z}} \nabla_\theta^2 T_\theta(z) \times_1 \nabla f(T_\theta(z)) + \nabla_\theta T_\theta(z)^\top \nabla^2 f(T_\theta(z)) \nabla_\theta T_\theta(z) \mathbf{d}\mu(z) \\
&\quad + \frac{1}{\gamma} \int_{\mathcal{X}} \frac{\int_{\mathcal{Z}} \mathcal{T}_y(T_\theta(z), f) \nabla_\theta T_\theta(z)^\top u_z(\theta, f) u_z(\theta, f)^\top \nabla_\theta T_\theta(z) \mathbf{d}\mu(z)}{\int_{\mathcal{Z}} \mathcal{T}_y(T_\theta(z), f) \mathbf{d}\mu(z)} \mathbf{d}\beta(y) \\
&\quad + \int_{\mathcal{X}} \frac{\int_{\mathcal{Z}} \mathcal{T}_y(T_\theta(z), f) \nabla_\theta^2 T_\theta(z) \times_1 u_z(\theta, f) \mathbf{d}\mu(z)}{\int_{\mathcal{Z}} \mathcal{T}_y(T_\theta(z), f) \mathbf{d}\mu(z)} \mathbf{d}\beta(y) \\
&\quad + \int_{\mathcal{X}} \frac{\int_{\mathcal{Z}} \mathcal{T}_y(T_\theta(z), f) \nabla_\theta T_\theta(z)^\top [\nabla_{11} c(T_\theta(z), y) - \nabla^2 f(T_\theta(z))] \nabla_\theta T_\theta(z) \mathbf{d}\mu(z)}{\int_{\mathcal{Z}} \mathcal{T}_y(T_\theta(z), f) \mathbf{d}\mu(z)} \mathbf{d}\beta(y) \\
&\quad + \frac{1}{\gamma} \int_{\mathcal{X}} \frac{\int_{\mathcal{Z}} \mathcal{T}_y(T_\theta(z), f) \nabla_\theta T_\theta(z)^\top u_z(\theta, f) \mathbf{d}\mu(z) \left[\int_{\mathcal{Z}} \mathcal{T}_y(T_\theta(z), f) \nabla_\theta T_\theta(z)^\top u_z(\theta, f) \mathbf{d}\mu(z)\right]^\top}{\left[\int_{\mathcal{Z}} \mathcal{T}_y(T_\theta(z), f) \mathbf{d}\mu(z)\right]^2} \mathbf{d}\beta(y).
\end{aligned}
$$

Here $\nabla_\theta T_\theta(z) \in \mathbb{R}^{q \times d}$ and $\nabla_\theta^2 T_\theta(z) \in \mathbb{R}^{q \times d \times d}$ denote the first and second order Jacobian of $T_\theta(z)$ w.r.t. to $\theta$; $\times_1$ denotes the tensor product along the first dimension; $\nabla f \in \mathbb{R}^q$ and $\nabla^2 f \in \mathbb{R}^{q \times q}$ denote the first and second order gradient of $f$ w.r.t. its input; $\nabla_1 c \in \mathbb{R}^q$ and $\nabla_{11} c \in \mathbb{R}^{q \times q}$ denote the first and second order gradient of $c$ w.r.t. its first input. By plugging in $f = f_\theta$, we have the explicit expression of the second term of (52).

## B.2 More details in Proposition 5.2

First, we recall some existing results about the Sinkhorn potential $f_\theta$.

**Assumption B.1.** *The ground cost function $c$ is bounded and we denote $M_c := \max_{x, y \in \mathcal{X}} c(x, y)$.*

It is known that, under the above boundedness assumption on the ground cost function $c$, $f_\theta$ is a solution to the generalized DAD problem (eq. (7.4) in [Lemmens and Nussbaum, 2012]), which is the fixed point to the operator $\mathcal{B} : \mathcal{C}(\mathcal{X}) \times \Theta \to \mathcal{C}(\mathcal{X})$ defined as

$$
\mathcal{B}(f, \theta) := \mathcal{A}(\mathcal{A}(f, \alpha_\theta), \beta).
\tag{59}
$$

Further, the Birkhoff-Hopf Theorem (Sections A.4 and A.7 in [Lemmens and Nussbaum, 2012]) states that $\exp(\mathcal{B}/\gamma)$ is a contraction operator under the Hilbert metric with a contraction factor $\lambda^2$ where $\lambda := \frac{\exp(M_c/\gamma)-1}{\exp(M_c/\gamma)+1} < 1$ (see also Theorem B.5 in [Luise et al., 2019]): For strictly positive functions $u, u' \in \mathcal{C}(\mathcal{X})$, define the Hilbert metric as

$$d_H(u, u') := \log \max_{x,y \in \mathcal{X}} \frac{u(x)u'(y)}{u'(x)u(y)}. \tag{60}$$

For any measure $\alpha \in \mathcal{M}_1^+(\mathcal{X})$, we have

$$d_H(\exp(\mathcal{A}(f, \alpha_\theta)/\gamma), \exp(\mathcal{A}(f', \alpha_\theta)/\gamma)) \leqslant \lambda d_H(\exp(f/\gamma), \exp(f'/\gamma)). \tag{61}$$

Consequently, by applying the fixed point iteration

$$f^{t+1} = \mathcal{B}(f^t, \theta), \tag{62}$$

also known as the Sinkhorn-Knopp algorithm, one can compute $f_\theta$ in logarithmic time: $\|f^{t+1} - f_\theta\|_\infty = \mathcal{O}(\lambda^t)$ (Theorem. 7.1.4 in [Lemmens and Nussbaum, 2012] and Theorem B.10 in [Luise et al., 2019]).

While the above discussion shows that the output of the Sinkhorn-Knopp algorithm well approximates the Sinkhorn potential $f_\theta$, it would be useful to discuss more about the boundedness property of the sequence $\{f^t\}$ produced by the above Sinkhorn-Knopp algorithm. We first show that under bounded initialization $f^0$, the entire sequence $\{f^t\}$ is bounded.

**Lemma B.1.** *Suppose that we initialize the Sinkhorn-Knopp algorithm with $f^0 \in \mathcal{C}(\mathcal{X})$ such that $\|f^0\|_\infty \leqslant M_c$. One has $\|f^t\|_\infty \leqslant M_c$, for $t = 1, 2, 3, \cdots$.*

*Proof.* For $\|f\|_\infty \leqslant M_c$ and any measure $\alpha \in \mathcal{M}_1^+(\mathcal{X})$, we have

$$\|\mathcal{A}(f, \alpha)\|_\infty = \gamma \| \log \int_\mathcal{X} \exp\{-c(x, \cdot)/\gamma\} \exp\{f(x)/\gamma\} \mathbf{d}\alpha(x)\|_\infty \leqslant \gamma \log \exp(M_c/\gamma) \leqslant M_c.$$

One can then check the lemma via induction. $\qquad\square$

We then show that the sequence $\{f^t\}$ has bounded first, second and third-order gradients under the following assumptions on the ground cost function $c$.

**Assumption B.2.** *The cost function $c$ is $G_c$-Lipschitz continuous with respect to one of its inputs: For all $x, x' \in \mathcal{X}$,*

$$|c(x, y) - c(x', y)| \leqslant G_c \|x - x'\|.$$

**Assumption B.3.** *The gradient of the cost function $c$ is $L_c$-Lipschitz continuous: for all $x, x' \in \mathcal{X}$,*

$$\|\nabla_1 c(x, y) - \nabla_1 c(x', y)\| \leqslant L_c \|x - x'\|.$$

**Assumption B.4.** *The Hessian matrix of the cost function $c$ is $L_{2,c}$-Lipschitz continuous: for all $x, x' \in \mathcal{X}$,*

$$\|\nabla_{11}^2 c(x, y) - \nabla_{11}^2 c(x', y)\| \leqslant L_{2,c} \|x - x'\|.$$

**Lemma B.2.** *Assume that the initialization $f^0 \in \mathcal{C}(\mathcal{X})$ satisfies $\|f^0\|_\infty \leqslant M_c$.*
*(i.) Under Assumptions B.1 and B.2, $\exists G_f$ such that $\|\nabla f^t\|_{2,\infty} \leqslant G_f, \forall t > 0$.*
*(ii.) Under Assumptions B.1 - B.3, $\exists L_f$ such that $\|\nabla^2 f^t(x)\| \leqslant L_f, \forall t > 0$.*
*(iii.) Under Assumptions B.1 - B.4, $\exists L_{2,f}$ such that $\|\nabla^2 f^t(x) - \nabla^2 f^t(y)\|_{op} \leqslant L_{2,f}\|x - y\|, \forall t > 0$.*
*(iv). For $\|f\|_\infty \leqslant M_c$, the function $\mathcal{B}(f, \theta)(x)$ is $G_f$-Lipschitz continuous.*

*Proof.* We denote $k(x, y) := \exp\{-c(x, y)/\gamma\}$ in this proof.

(i) Under Assumptions B.1 and B.2, $k$ is $[G_c/\gamma]$-Lipschitz continuous w.r.t. its first variable. For $f \in \mathcal{C}(\mathcal{X})$ such that $\|f\|_\infty \leqslant M_c$, we bound

$$|\mathcal{A}(f, \alpha)(x) - \mathcal{A}(f, \alpha)(y)| = \gamma |\log \int_\mathcal{X} [k(z, y) - k(z, x)] \exp\{f(z)/\gamma\} \mathbf{d}\alpha(z)|$$

$$\leqslant \gamma \exp(M_c/\gamma) G_c/\gamma \|x - y\|_2 = \exp(M_c/\gamma) G_c \|x - y\|_2.$$

Using Lemma B.1, we know that $\{f^t\}$ is $M_c$-bounded and hence
$$\|\nabla f^{t+1}\|_{2,\infty} \leqslant G_f = \exp(2M_c/\gamma)G_c^2.$$

(ii) Under Assumption B.1, $k(x,y) \geqslant \exp(-M_c/\gamma)$. We compute
$$\nabla\big(\mathcal{A}(f,\alpha)\big)(x) = \frac{\int_{\mathcal{X}} k(z,x)\exp\{f(z)/\gamma\}\nabla_1 c(x,z)\mathbf{d}\alpha(z)}{\int_{\mathcal{X}} k(z,x)\exp\{f(z)/\gamma\}\mathbf{d}\alpha(z)}. \qquad \#\frac{g_1(x)}{g_2(x)}$$

Let $g_1 : \mathbb{R}^q \to \mathbb{R}^q$ and $g_2 : \mathbb{R}^q \to \mathbb{R}$ be the numerator and denominator of the above expression. If we have (a) $\|g_1\|_{2,\infty} \leqslant G_1$, (b) $\|g_1(x) - g_1(y)\| \leqslant L_1\|x-y\|$ and (c) $\|g_2\|_\infty \leqslant G_2$, (d) $|g_2(x) - g_2(y)| \leqslant L_2\|x-y\|$, (e) $g_2 \geqslant \bar{G}_2 > 0$, we can bound
$$\|\frac{g_1(x)}{g_2(x)} - \frac{g_1(y)}{g_2(y)}\| = \|\frac{g_1(x)g_2(y) - g_1(y)g_2(x)}{g_2(x)g_2(y)}\| \leqslant \frac{G_2 L_1 + G_1 L_2}{\bar{G}_2^2}\|x-y\|, \qquad (63)$$

which means that $\nabla\big(\mathcal{A}(f,\alpha)\big)$ is $L$-Lipschitz continuous with $L = \frac{G_2 L_1 + G_1 L_2}{\bar{G}_2^2}$. We now prove (a)-(e).

(a) $\|\int_{\mathcal{X}} k(z,x)\exp\{f(z)/\gamma\}\nabla_1 c(x,z)\mathbf{d}\alpha(z)\|_{2,\infty} \leqslant \exp(M_c/\gamma) \cdot G_c$ (Assumption B.2).

(b) Note that for any two bounded and Lipschitz continuous functions $h_1 : \mathcal{X} \to \mathbb{R}$ and $h_2 : \mathcal{X} \to \mathbb{R}^q$, their product is also Lipschitz continuous:
$$\|h_1(x) \cdot h_2(x) - h_1(y) \cdot h_2(y)\| \leqslant [|h_1|_\infty \cdot G_{h_2} + \|h_2\|_{2,\infty} \cdot G_{h_1}]\|x-y\|, \qquad (64)$$
where $G_{h_i}$ denotes the Lipschitz constant of $h_i$, $i = 1,2$. Hence for $g_1$, we have
$$\|g_1(x) - g_1(y)\| \leqslant \exp(M_c/\gamma) \cdot (L_c + G_c^2/\gamma) \cdot \|x-y\|,$$
since $k(x,y) \leqslant 1$, $\|\nabla_1 k(x,y)\| \leqslant G_c/\gamma$, $\|\nabla_1 c(x,y)\| \leqslant G_c$, $\|\nabla_{11}^2 c(x,y)\|_{op} \leqslant L_c$.

(c) $\|\int_{\mathcal{X}} k(z,\cdot)\exp\{f(z)/\gamma\}\mathbf{d}\alpha(z)\|_\infty \leqslant \exp(M_c/\gamma)$.

(d) $|\int_{\mathcal{X}}[k(z,x) - k(z,y)]\exp\{f(z)/\gamma\}\mathbf{d}\alpha(z)| \leqslant \exp(M_c/\gamma) \cdot G_c/\gamma \cdot \|x-y\|$.

(e) $\int_{\mathcal{X}} k(z,x)\exp\{f(z)/\gamma\}\mathbf{d}\alpha(z) \geqslant \exp(-2M_c/\gamma) > 0$.

Combining the above points, we prove the existence of $L_f$.

For (iii), compute that
$$\nabla^2\big(\mathcal{A}(f,\alpha)\big)(x)$$
$$= \frac{\int_{\mathcal{X}} k(z,x)\exp\{f(z)/\gamma\}\nabla_1 c(x,z)\nabla_1 c(x,z)^\top \mathbf{d}\alpha(z)}{\int_{\mathcal{X}} k(z,x)\exp\{f(z)/\gamma\}\mathbf{d}\alpha(z)} \qquad \#1$$
$$+ \frac{\int_{\mathcal{X}} k(z,x)\exp\{f(z)/\gamma\}\nabla_{11}^2 c(x,z)\mathbf{d}\alpha(z)}{\int_{\mathcal{X}} k(z,x)\exp\{f(z)/\gamma\}\mathbf{d}\alpha(z)} \qquad \#2$$
$$- \frac{\int_{\mathcal{X}} k(z,x)\exp\{f(z)/\gamma\}\nabla_1 c(x,z)\mathbf{d}\alpha(z)\big[\int_{\mathcal{X}} k(z,x)\exp\{f(z)/\gamma\}\nabla_1 c(x,z)\mathbf{d}\alpha(z)\big]^\top}{\big[\int_{\mathcal{X}} k(z,x)\exp\{f(z)/\gamma\}\mathbf{d}\alpha(z)\big]^2}. \qquad \#3$$

We now analyze #1-#3 individually.

#1 Note that for any two bounded and Lipschitz continuous functions $h_1 : \mathcal{X} \to \mathbb{R}$ and $h_2 : \mathcal{X} \to \mathbb{R}^{q \times q}$, their product is also Lipschitz continuous:
$$\|h_1(x) \cdot h_2(x) - h_1(y) \cdot h_2(y)\|_{op} \leqslant [|h_1|_\infty \cdot G_{h_2} + \|h_2\|_{op,\infty} \cdot G_{h_1}]\|x-y\|, \qquad (65)$$
where $G_{h_i}$ denotes the Lipschitz constant of $h_i$, $i = 1,2$.

Take $h_1(x) = k(z',x)\exp\{f(z')/\gamma\}/\int_{\mathcal{X}} k(z,x)\exp\{f(z)/\gamma\}\mathbf{d}\alpha(z)$. $h_1$ is bounded since $k(z',x) \leqslant 1$ and $\int_{\mathcal{X}} k(z,x)\exp\{f(z)/\gamma\}\mathbf{d}\alpha(z) \geqslant \exp(-2M_c/\gamma) > 0$. $h_1$ is Lipschitz continuous since we additionally have $k(z',x)$ being Lipschitz continuous (see (63)).

Take $h_2(x) = \nabla_1 c(x,z)\nabla_1 c(x,z)^\top$. $h_2$ is bounded since $\|\nabla_1 c(x,z)\| \leqslant G_c$ (Assumption B.2). $h_2$ is Lipschitz continuous due to Assumption B.3.

#2 Following the similar argument as #1, we have the result. Note that $h_2(x) = \nabla^2_{11} c(x, z)$ is Lipschitz continuous due to Assumption B.4.

#3 We follow the similar argument as #1 by taking

$$h_1(x) = \frac{k(z', x) \exp\{f(z')/\gamma\} k(z', x) \exp\{f(z')/\gamma\}}{\left[ \int_{\mathcal{X}} k(z, x) \exp\{f(z)/\gamma\} \mathbf{d}\alpha(z) \right]^2},$$

and taking

$$h_2(x) = \nabla_1 c(x, z) [\nabla_1 c(x, z)]^\top.$$

Combining the above points, we prove the existence of $L_{2,f}$.

(iv) As a composition of $\mathcal{A}$, we also have that $\mathcal{B}(f, \theta)$ is $G_f$-Lipschitz continuous (see $G_f$ in (i)). $\square$

Moreover, based on the above continuity results, we can show that the first-order gradient $\nabla f_\theta^\epsilon$ (and second-order gradient $\nabla^2 f_\theta^\epsilon$) also converges to $\nabla f_\theta$ (and $\nabla^2 f_\theta$) in time logarithmically depending on $1/\epsilon$.

**Lemma B.3.** *Under Assumptions B.1-B.3, the Sinkhorn-Knopp algorithm, i.e. the fixed point iteration*

$$f^{t+1} = \mathcal{B}(f^t, \theta), \tag{66}$$

*computes $\nabla f_\theta$ in logarithm time:* $\|\nabla f^{t+1} - \nabla f_\theta\|_{2,\infty} = \epsilon$ *with* $t = \mathcal{O}(\log \frac{1}{\epsilon})$.

*Proof.* For a fix point $x \in \mathcal{X}$ and any direction $h \in \mathbb{R}^q$, we have

$$f^t(x + \eta \cdot h) - f^t(x) = \eta [\nabla f^t(x)]^\top h + \frac{\eta^2}{2} h^\top \nabla^2 f^t(x + \tilde{\eta}_1 \cdot h) h,$$

where $\eta > 0$ is some constant to be determined later and $0 \leqslant \tilde{\eta}_1 \leqslant \eta$ is obtained from the mean value theorem. Similarly, we have for $0 \leqslant \tilde{\eta}_2 \leqslant \eta$

$$f_\theta(x + \eta \cdot h) - f_\theta(x) = \eta [\nabla f_\theta(x)]^\top h + \frac{\eta^2}{2} h^\top \nabla^2 f_\theta(x + \tilde{\eta}_2 \cdot h) h.$$

We can then compute

$$|[\nabla f^t(x) - \nabla f_\theta(x)]^\top h| \leqslant \frac{2}{\eta} \|f^t - f_\theta\|_\infty + \eta L_f \|h\|^2.$$

Take $h = \nabla f^t(x) - \nabla f_\theta(x)$ and $\eta = \frac{2}{L_f}$. We derive from the above inequality

$$\|\nabla f^t(x) - \nabla f_\theta(x)\|^2 \leqslant 2L_f \|f^t - f_\theta\|_\infty.$$

Consequently, if we have $2L_f \|f^t - f_\theta\|_\infty \leqslant \epsilon^2$, we can prove that $\|\nabla f^t - \nabla f_\theta\|_{2,\infty} \leqslant \epsilon$ since $x$ is arbitrary. This can be achieve in logarithmic time using the Sinkhorn-Knopp algorithm. $\square$

**Lemma B.4.** *Under Assumptions B.1-B.4, the Sinkhorn-Knopp algorithm, i.e. the fixed point iteration*

$$f^{t+1} = \mathcal{B}(f^t, \theta), \tag{67}$$

*computes $\nabla^2 f_\theta$ in logarithm time:* $\|\nabla^2 f^{t+1} - \nabla^2 f_\theta\|_{op,\infty} = \epsilon$ *with* $t = \mathcal{O}(\log \frac{1}{\epsilon})$.

*Proof.* This follows a similar argument as Lemma B.3 by noticing that the third order gradient of $f^t$ (and $f_\theta$) is bounded due to Assumption B.4. $\square$

## B.3  Proof of Proposition 5.3

We now construct a sequence $\{g^t\}$ to approximate the Fréchet derivative of the Sinkhorn potential $Df_\theta$ such that for all $t \geqslant T(\epsilon)$ with some integer function $T(\epsilon)$ of the target accuracy $\epsilon$, we have $\|g_\theta^t - Df_\theta\|_{op} \leqslant \epsilon$. In particular, we show that such $\epsilon$-accurate approximation can be achieved using a logarithmic amount of simple function operations and integrations with respect to $\alpha_\theta$.

For a given target accuracy $\epsilon > 0$, denote $\bar{\epsilon} = \epsilon/L_l$, where $L_l$ is a constant defined in Lemma B.5. First, Use the Sinkhorn-Knopp algorithm to compute $f_\theta^{\bar{\epsilon}}$, an approximation of $f_\theta$ such that $\|f_\theta^{\bar{\epsilon}} - f_\theta\|_\infty \leqslant \bar{\epsilon}$. This computation can be done in $\mathcal{O}(\log \frac{1}{\epsilon})$ from Proposition 5.2.

Denote $\mathcal{E}(f, \theta) = \mathcal{B}^l(f, \theta) = \mathcal{B}(\cdots \mathcal{B}(f, \theta), \cdots, \theta)$, the $l$ times composition of $\mathcal{B}$ in its first variable. Pick $l = \lceil \log_\lambda \frac{1}{3} \rceil / 2$. From the contraction of $\mathcal{A}$ under the Hilbert metric (61), we have

$$\|\mathcal{E}(f, \theta) - \mathcal{E}(f', \theta)\|_\infty \leqslant \gamma d_H(\exp(\mathcal{E}(f, \theta)/\gamma), \exp(\mathcal{E}(f', \theta)/\gamma))$$

$$\leqslant \gamma \lambda^{2l} d_H(\exp(f/\gamma), \exp(f'/\gamma)) \leqslant 2\lambda^{2l}\|f - f'\|_\infty \leqslant \frac{2}{3}\|f - f'\|_\infty,$$

where we use $\|f - f'\|_\infty \leqslant d_H(\exp(f), \exp(f')) \leqslant 2\|f - f'\|_\infty$ in the first and third inequalities. Consequently, $\mathcal{E}[f, \theta]$ is a contraction operator w.r.t. $f$ under the $l_\infty$ norm, which is equivalent to

$$\|D^1 \mathcal{E}(f, \theta)\|_{op} \leqslant \frac{2}{3}. \tag{68}$$

Now, given arbitrary initialization $g_\theta^0 : \Theta \to T(\mathbb{R}^d, \mathcal{C}(\mathcal{X}))^1$, construct iteratively

$$g_\theta^{t+1} = D_1 \mathcal{E}(f_\theta^{\bar{\epsilon}}, \theta) \circ g_\theta^t + D_2 \mathcal{E}(f_\theta^{\bar{\epsilon}}, \theta), \tag{69}$$

where $\circ$ denotes the composition of (linear) mappings. In the following, we show that

$$\|g_\theta^{t+1} - Df_\theta\|_{op} \leqslant 3\epsilon + (\frac{2}{3})^t\|g_\theta^0 - Df_\theta\|_{op}.$$

First, note that $f_\theta$ is a fixed point of $\mathcal{E}(\cdot, \theta)$

$$f_\theta = \mathcal{E}(f_\theta, \theta).$$

Take the Fréchet derivative w.r.t. $\theta$ on both sides of the above equation. Using the chain rule, we compute

$$Df_\theta = D_1 \mathcal{E}(f_\theta, \theta) \circ Df_\theta + D_2 \mathcal{E}(f_\theta, \theta). \tag{70}$$

For any direction $h \in \mathbb{R}^d$, we bound the difference of the directional derivatives by

$$\|g_\theta^{t+1}[h] - Df_\theta[h]\|_\infty$$

$$\leqslant \|D_1 \mathcal{E}(f_\theta, \theta)[Df_\theta[h]] - D_1 \mathcal{E}(f_\theta^{\bar{\epsilon}}, \theta)[g_\theta^t[h]]\|_\infty + \|D_2 \mathcal{E}(f_\theta^{\bar{\epsilon}}, \theta)[h] - D_2 \mathcal{E}(f_\theta, \theta)[h]\|_\infty$$

$$\leqslant \frac{2}{3}\|Df_\theta[h] - g_\theta^t[h]\|_\infty + L_l(\|f_\theta^{\bar{\epsilon}} - f_\theta\|_\infty + \|\nabla f_\theta^{\bar{\epsilon}} - \nabla f_\theta\|_\infty)\|h\|_\infty$$

$$\leqslant \frac{2}{3}\|Df_\theta - g_\theta^t\|_{op}\|h\|_\infty + \epsilon\|h\|_\infty,$$

where in the second inequality we use the bound on $D_1\mathcal{E}$ in (68) and the $L_l$-Lipschitz continuity of $D_2\mathcal{E}$ with respect to its first argument (recall that $f_\theta^{\bar{\epsilon}}$ is obtained from the Sinkhorn-Knopp algorithm and hence $\|f_\theta^{\bar{\epsilon}}\|_\infty \leqslant M_c$ from Lemma B.1 and $\|\nabla f_\theta^{\bar{\epsilon}}\|_{2,\infty} \leqslant G_f$ from (i) of Lemma B.2). The above inequality is equivalent to

$$\|g_\theta^{t+1} - Df_\theta\|_{op} - 3\epsilon \leqslant \frac{2}{3}\left(\|Df_\theta - g_\theta^t\|_{op} - 3\epsilon\right) \Rightarrow \|g_\theta^{t+1} - Df_\theta\|_{op} \leqslant 3\epsilon + (\frac{2}{3})^t\|g_\theta^0 - Df_\theta\|_{op}.$$

Therefore, after $T(\epsilon) = \mathcal{O}(\log \frac{1}{\epsilon})$ iterations, we find $g_\theta^{T(\epsilon)}$ such that $\|g_\theta^{T(\epsilon)} - Df_\theta\|_{op} \leqslant 4\epsilon$.

**Assumption B.5** (Boundedness of $\nabla_\theta T_\theta(x)$). *There exists some $G_T > 0$ such that for any $x \in \mathcal{X}$ and $\theta \in \Theta$, $\|\nabla_\theta T_\theta(x)\|_{op} \leqslant G_T$.*

**Lemma B.5** (Lipschitz continuity of $D_2\mathcal{E}$). *Under Assumptions B.1 - B.3 and B.5, $D_2\mathcal{E}$ is Lipschitz continuous with respect to its first variable: For $f, f' \in \mathcal{C}(\mathcal{X})$ such that $\|f\|_\infty \leqslant M_c$ ($\|f'\|_\infty \leqslant M_c$) and $\|\nabla f\|_\infty \leqslant G_f$ ($\|\nabla f'\|_\infty \leqslant G_f$), and $\theta \in \Theta$ there exists some $L_l$ such that*

$$\|D_2\mathcal{E}(f,\theta) - D_2\mathcal{E}(f',\theta)\|_{op} \leqslant L_l\big(\|f - f'\|_\infty + \|\nabla f - \nabla f'\|_{2,\infty}\big). \tag{71}$$

*Proof.* Recall that $\mathcal{E}(\cdot, \theta) = \mathcal{B}^l(\cdot, \theta)$. Using the chain rule of Fréchet derivative, we compute

$$D_2\mathcal{B}^l(f,\theta) = D_1\mathcal{B}\big(\mathcal{B}^{l-1}(f,\theta),\theta\big) \circ D_2\mathcal{B}^{l-1}(f,\theta) + D_2\mathcal{B}\big(\mathcal{B}^{l-1}(f,\theta),\theta\big). \tag{72}$$

We bound the two terms on the R.H.S. individually.

**Analyze the first term of** (72). For a given $f$, use $A_f$ and $B_f$ to denote two linear operators depending on $f$. We have $\|A_f \circ B_f - A_{f'} \circ B_{f'}\|_{op} = \mathcal{O}(\|f - f'\|_\infty + \|\nabla f - \nabla f'\|_{2,\infty})$ if both $A_f$ and $B_f$ are bounded, $\|A_f - A_{f'}\|_{op} = \mathcal{O}(\|f - f'\|_\infty + \|\nabla f - \nabla f'\|_{2,\infty})$, and $\|B_f - B_{f'}\|_{op} = \mathcal{O}(\|f - f'\|_\infty + \|\nabla f - \nabla f'\|_{2,\infty})$:

$$\|A_f \circ B_f - A_{f'} \circ B_{f'}\|_{op} \leqslant \|A_f \circ B_f - A_f \circ B_{f'}\|_{op} + \|A_f \circ B_{f'} - A_{f'} \circ B_{f'}\|_{op}$$
$$\leqslant \big[\max_f \|B_f\|_{op} \cdot L_A + \max_f \|A_f\|_{op} \cdot L_B\big]\big[\|f - f'\|_\infty + \|\nabla f - \nabla f'\|_{2,\infty}\big], \tag{73}$$

where $L_A$ and $L_B$ denote the constants of operators $A_f$ and $B_f$ such that

$$\|A_f - A_{f'}\| \leqslant L_A\big[\|f - f'\|_\infty + \|\nabla f - \nabla f'\|_{2,\infty}\big]$$
$$\|B_f - B_{f'}\| \leqslant L_B\big[\|f - f'\|_\infty + \|\nabla f - \nabla f'\|_{2,\infty}\big].$$

We now take

$$A_f = D_1\mathcal{B}\big(\mathcal{B}^{l-1}(f,\theta),\theta\big) \text{ and } B_f = D_2\mathcal{B}^{l-1}(f,\theta).$$

$\|A_f\|_{op}$ is bounded from the following lemma.

**Lemma B.6.** $\mathcal{B}(f,\theta)$ *is 1-Lipschitz continuous with respect to its first variable.*

*Proof.* We compute that for any measure $\kappa$ and any function $g \in \mathcal{C}(\mathcal{X})$,

$$D_1\mathcal{A}(f,\kappa)[g] = \frac{\int_\mathcal{X} \exp\{-\frac{1}{\gamma}\big(c(x,y) - f(x)\big)\}g(x)\mathbf{d}\kappa(x)}{\int_\mathcal{X} \exp\{-\frac{1}{\gamma}\big(c(x,y) - f(x)\big)\}\mathbf{d}\kappa(x)}. \tag{74}$$

Note that

$$\|D_1\mathcal{A}(f,\kappa)[g]\|_\infty \leqslant \|\frac{\int_\mathcal{X} \exp\{-\frac{1}{\gamma}\big(c(x,y) - f(x)\big)\}\mathbf{d}\kappa(x)}{\int_\mathcal{X} \exp\{-\frac{1}{\gamma}\big(c(x,y) - f(x)\big)\}\mathbf{d}\kappa(x)}\|_\infty \cdot \|g\|_\infty = \|g\|_\infty, \tag{75}$$

and consequently we have $\|D_1\mathcal{A}(f,\kappa)\|_{op} \leqslant 1$. Further, since $\mathcal{B}$ is the composition of $\mathcal{A}$ in its first variable, we have that $\|D_1\mathcal{B}(f,\theta)\|_{op} \leqslant 1$. $\square$

$\|B_f\|_{op}$ is bounded from the following lemma.

**Lemma B.7.** *Assume that $f \in \mathcal{C}(\mathcal{X})$ satisfies $\|f\|_\infty \leqslant M_c$ and $\|\nabla f\|_{2,\infty} \leqslant G_f$. Under Assumptions B.2 and B.5, $\forall l \geqslant 1$, $\|D_2\mathcal{B}^l(f,\theta)\|_{op}$ is $M_l$-bounded, with $M_l = l \cdot \exp(3M_c/\gamma) \cdot G_T \cdot (G_c + G_f)$.*

*Proof.* In this proof, we denote $\tilde{\mathcal{A}}(f,\theta):=\mathcal{A}(f,\alpha_\theta)$ to make the dependence of $\mathcal{A}$ on $\theta$ explicit. Using the chain rule of Fréchet derivative, we compute

$$D_2\mathcal{B}^l(f,\theta) = D_1\mathcal{B}\big(\mathcal{B}^{l-1}(f,\theta),\theta\big) \circ D_2\mathcal{B}^{l-1}(f,\theta) + D_2\mathcal{B}\big(\mathcal{B}^{l-1}(f,\theta),\theta\big). \tag{76}$$

We will use $M_l$ to denote the upper bound of $\|D_2\mathcal{B}^l(f,\theta)\|_{op}$. Consequently we have

$$M_l \leqslant \|D_1\mathcal{B}\big(\mathcal{B}^{l-1}(f,\theta),\theta\big)\|_{op}\|D_2\mathcal{B}^{l-1}(f,\theta)\|_{op} + \|D_2\mathcal{B}\big(\mathcal{B}^{l-1}(f,\theta),\theta\big)\|_{op}$$
$$\leqslant M_{l-1} + \|D_2\mathcal{B}\big(\mathcal{B}^{l-1}(f,\theta),\theta\big)\|_{op},$$

where we use Lemma B.6 in the second inequality. Recall that $\mathcal{B}(f, \theta) = \mathcal{A}(\tilde{\mathcal{A}}(f, \theta), \beta)$. Again using the chain rule of the Fréchet derivative, we compute

$$D_2\mathcal{B}(f, \theta) = D_1\mathcal{A}\big(\tilde{\mathcal{A}}(f, \theta), \beta\big) \circ D_2\tilde{\mathcal{A}}(f, \theta), \tag{77}$$

and hence

$$\|D_2\mathcal{B}(f, \theta)\|_{op} \leqslant \|D_1\mathcal{A}\big(\tilde{\mathcal{A}}(f, \theta), \beta\big)\|_{op} \cdot \|D_2\tilde{\mathcal{A}}(f, \theta)\|_{op} \leqslant \|D_2\tilde{\mathcal{A}}(f, \theta)\|_{op}, \tag{78}$$

where we use (75) in the second inequality. We now bound $\|D_2\tilde{\mathcal{A}}(f, \theta)\|_{op}$. Denote

$$\omega_y(x) := \exp(-c(x, y)/\gamma) \exp(f(x)/\gamma).$$

We have $\exp(-2M_c/\gamma) \leqslant \omega_y(x) \leqslant \exp(M_c/\gamma)$ from $\|f\|_\infty \leqslant M_c$ and Assumption B.1. For any direction $h \in \mathbb{R}^q$ (note that $D_2\tilde{\mathcal{A}}(f, \theta)[h] : \mathcal{X} \to \mathbb{R}$) and any $y \in \mathcal{X}$, we compute

$$\big(D_2\tilde{\mathcal{A}}(f, \theta)[h]\big)(y) = \frac{\int_{\mathcal{X}} \omega_y(T_\theta(x))\langle[\nabla_\theta T_\theta(x)]^\top [-\nabla_1 c(T_\theta(x), y) + \nabla f(T_\theta(x))], h\rangle \mathbf{d}\mu(x)}{\int_{\mathcal{X}} \omega_y(T_\theta(x))\mathbf{d}\mu(x)},$$

where $\nabla_\theta T_\theta(x)$ denotes the Jacobian matrix of $T_\theta(x)$ w.r.t. $\theta$. Consequently we bound

$$\|D_2\tilde{\mathcal{A}}(f, \theta)[h]\|_\infty \leqslant \exp(3M_c/\gamma)\|\nabla_\theta T_\theta(x)\|_{op} \cdot [\|\nabla_1 c\big(T_\theta(x), y\big)\| + \|\nabla f\big(T_\theta(x)\big)\|] \cdot \|h\|$$
$$\leqslant \exp(3M_c/\gamma) \cdot G_T \cdot (G_c + G_f)\|h\|,$$

which implies

$$\|D_2\tilde{\mathcal{A}}(f, \theta)\|_{op} \leqslant \exp(3M_c/\gamma) \cdot G_T \cdot (G_c + G_f). \tag{79}$$

$\square$

To show the Lipschitz continuity of $A_f$, i.e. $\|A_f - A_{f'}\| \leqslant L_A\|f - f'\|_\infty$, we first establish the following continuity lemmas of $D_1\mathcal{B}(\cdot, \theta)$ and $\mathcal{B}^{l-1}(\cdot, \theta)$.

**Lemma B.8.** *For $f \in \mathcal{C}(\mathcal{X})$ such $\|f\|_\infty \leqslant M_c$, $D_1\mathcal{B}(f, \theta)$ is $L$-Lipschitz continuous with respect to its first variable with $L = 2L_{\mathcal{A}}$.*

*Proof.* Use the chain rule of Fréchet derivative to compute

$$D_1\mathcal{B}(f, \theta) = \underbrace{D_1\mathcal{A}\big(\mathcal{A}(f, \alpha_\theta), \beta\big)}_{U_f} \circ \underbrace{D_1\mathcal{A}(f, \alpha_\theta)}_{V_f}. \tag{80}$$

We analyze the Lipschitz continuity of $\|D_1\mathcal{B}(f, \theta)\|_{op}$ following the same logic as (73):

- The 1-boundedness of $U_f$ and $V_f$ is from Lemma B.6.

- The $L_{\mathcal{A}}$-Lipschitz continuity of $V_f$ is from Lemma B.11.

- The $L_{\mathcal{A}}$-Lipschitz continuity of $U_f$ is from Lemmas B.6 and B.11.

Consequently, we have that $D_1\mathcal{B}(f, \theta)$ is $2L_{\mathcal{A}}$-Lipschitz continuous w.r.t. its first variable. $\square$

**Lemma B.9.** $\forall l, \mathcal{B}^l(f, \theta)$ *is 1-Lipschitz continuous with respect to its first variable.*

*Proof.* Use the chain rule of Fréchet derivative to compute

$$D_1\mathcal{B}^l(f, \theta) = D_1\mathcal{B}\big(\mathcal{B}^{l-1}(f, \theta), \theta\big) \circ D_1\mathcal{B}^{l-1}(f, \theta). \tag{81}$$

Consequently $\|D_1\mathcal{B}^l(f, \theta)\|_{op} \leqslant \|D_1\mathcal{B}(f, \theta)\|_{op}^l$. Further, we have $\|D_1\mathcal{B}(f, \theta)\|_{op} \leqslant 1$ from Lemma B.6 which leads to the result. $\square$

We have that $A_f$ is Lipschitz continuous since (i) $A_f$ is the composition of Lipschitz continuous operators $D_1\mathcal{B}(\cdot, \theta)$ and $\mathcal{B}^{l-1}(f\cdot, \theta)$ and (ii) for $\|f\|_\infty \leqslant M_c$, $\forall l \geqslant 0, \|\mathcal{B}^l(f, \theta)\|_\infty \leqslant M_c$ (the argument is similar to Lemma B.1).

We prove $\|B_f - B_{f'}\| \leqslant L_l[\|f - f'\|_\infty + \|\nabla f - \nabla f'\|_{2,\infty}]$ via induction. The following lemma establishes the base case for $D_2\mathcal{B}(f, \theta)$ (when $l = 2$). Note that the boundedness of $\|f\|_\infty$ ($\|f'\|_\infty$) and $\|\nabla f\|_\infty$ ($\|\nabla f'\|_\infty$) remains valid after the operator $\mathcal{B}$ (Lemma B.1 and (i) of Lemma (B.2)).

**Lemma B.10.** *There exists constant $L_1$ such that for $\|f\|_\infty \leqslant M_c$ ($\|f'\|_\infty \leqslant M_c$) and $\|\nabla f\|_\infty \leqslant G_f$ ($\|\nabla f'\|_\infty \leqslant G_f$)*

$$\|D_2\mathcal{B}(f,\theta) - D_2\mathcal{B}(f',\theta)\|_{op} \leqslant L_1\big[\|f-f'\|_\infty + \|\nabla f - \nabla f'\|_{2,\infty}\big]. \tag{82}$$

*Proof.* In this proof, we denote $\tilde{\mathcal{A}}(f,\theta):=\mathcal{A}(f,\alpha_\theta)$ to make the dependence of $\mathcal{A}$ on $\theta$ explicit. Recall that $\mathcal{B}(f,\theta) = \mathcal{A}(\tilde{\mathcal{A}}(f,\theta),\beta)$. Use the chain rule of Fréchet derivative to compute

$$D_2\mathcal{B}(f,\theta) = \underbrace{D_1\mathcal{A}\big(\mathcal{A}(f,\alpha_\theta),\beta\big)}_{U_f} \circ \underbrace{D_2\tilde{\mathcal{A}}(f,\theta)}_{V_f}. \tag{83}$$

We analyze the Lipschitz continuity of $\|D_2\mathcal{B}(f,\theta)\|_{op}$ following the same logic as (73):

- The 1-boundedness of $U_f$ is from Lemma B.6.

- The $\exp(3M_c/\gamma) \cdot G_T \cdot (G_c + G_f)$-boundedness of $V_f$ is from (79).

- The $L_A$-Lipschitz continuity of $U_f$ is from Lemmas B.6 and B.11 and the fact that for $\|f\|_\infty \leqslant M_c$, $\|\mathcal{A}(f,\theta)\|_\infty \leqslant M_c$ (the argument is similar to Lemma B.1).

- Denote
$$\mathcal{T}_y(x,f):= \exp(-c(x,y)/\gamma)\exp(f(x)/\gamma).$$
We compute

$$V_f = \frac{\int_{\mathcal{Z}} \mathcal{T}_y\big(T_\theta(z),f\big)[\nabla_\theta T_\theta(z)]^\top \big[-\nabla_1 c(T_\theta(z),y) + \nabla f\big(T_\theta(z)\big)\big]\,\mathbf{d}\mu(z)}{\int_{\mathcal{Z}} \mathcal{T}_y\big(T_\theta(z),f\big)\mathbf{d}\mu(z)}, \quad \#\frac{P_f}{Q_f}$$

Denote the numerator by $P_f$ and the denominator by $Q_f$. Following the similar idea as (63), we show that both $\|P_f\|_{op}$ and $\|Q_f\|_\infty$ are bounded, $Q_f$ is Lipschitz continuous w.r.t. $f$, $Q_f$ is positive and bounded from below, and $\|P_f - P_{f'}\|_{op} \leqslant L_v[\|f-f'\|_\infty + \|\nabla f - \nabla f'\|_{2,\infty}]$ for some constant $L_v$.

  - The boundedness of $\|P_f\|_{op}$ is from the boundedness of $f$, Assumptions B.5, B.2, and the boundedness of $\nabla f$.
  - The boundedness of $\|Q_f\|_\infty$ is from the boundedness of $f$.
  - Use $DQ_f$ to denote the Fréchet derivative of $Q_f$ w.r.t. $f$. For any function $g \in \mathcal{C}(\mathcal{X})$,

  $$DQ_f[g] = \int_{\mathcal{X}} \mathcal{T}_y(x,f)g(x)/\gamma\mathbf{d}\alpha_\theta(x), \tag{84}$$

  where we recall that $\alpha_\theta = T_{\theta\sharp}\mu$. Further, we have $\|DQ_f[g]\|_\infty \leqslant \exp(M_c/\gamma)/\gamma\|g\|_\infty$, which implies the Lipschitz continuity of $Q_f$ (for $\|f\|_\infty \leqslant M_c$).
  - We prove that for $\|f\|_\infty \leqslant M_c$ ($\|f'\|_\infty \leqslant M_c$) and $\|\nabla f\|_\infty \leqslant G_f$ ($\|\nabla f'\|_\infty \leqslant G_f$),

  $$\|P_f - P_{f'}\|_{op} \leqslant L_v[\|f-f'\|_\infty + \|\nabla f - \nabla f'\|_{2,\infty}].$$

  For a fixed $z \in \mathcal{Z}$, denote

  $$p_f^z := \mathcal{T}_y\big(T_\theta(z),f\big)[\nabla_\theta T_\theta(z)]^\top \big[-\nabla_1 c(T_\theta(z),y) + \nabla f\big(T_\theta(z)\big)\big].$$

  Note that $P_f = \int_{\mathcal{Z}} p_f^z \mathbf{d}\mu(z)$. For any direction $h \in \mathbb{R}^d$, we bound

  $\|p_f^z[h] - p_{f'}^z[h]\|_{op}$
  $\leqslant \|D_2\mathcal{T}_y\big(T_\theta(z),f\big)\|_{op}\|f-f'\|_\infty \cdot \max_y |[\nabla_\theta T_\theta(z)h]^\top \big[-\nabla_1 c(T_\theta(z),y) + \nabla f\big(T_\theta(z)\big)\big]|$
  $\quad + [\max_y \mathcal{T}_y\big(T_\theta(z),f\big)] \cdot \|\nabla_\theta T_\theta(z)h\|\|\nabla f\big(T_\theta(z)\big) - \nabla f'\big(T_\theta(z)\big)\|$
  $\leqslant \exp(M_c/\gamma)/\gamma \cdot G_T \cdot (G_c + G_f) \cdot \|f-f'\|_\infty \cdot \|h\| + \exp(M_c/\gamma) \cdot G_T \cdot \|h\| \cdot \|\nabla f - \nabla f'\|_{2,\infty}.$

  Consequently, we have that there exists a constant $L_v$ such that

  $$\|p_f^z[h] - p_{f'}^z[h]\|_\infty \leqslant L_v[\|f-f'\|_\infty + \|\nabla f - \nabla f'\|_{2,\infty}] \cdot \|h\|.$$

$\square$

The above lemma shows the base case for the induction. Now suppose that the inequality $\|D_2\mathcal{B}^k(f,\theta) - D_2\mathcal{B}^k(f',\theta)\|_{op} \leqslant L_k[\|f - f'\|_\infty + \|\nabla f - \nabla f'\|_{2,\infty}]$ holds.
For the case of $k+1$, we compute the Fréchet derivative

$$D_2\mathcal{B}^{k+1}(f,\theta) = D_1\mathcal{B}\big(\mathcal{B}^k(f,\theta),\theta\big) \circ D_2\mathcal{B}^k(f,\theta) + D_2\mathcal{B}\big(\mathcal{B}^k(f,\theta),\theta\big),$$

and hence we can bound

$$
\begin{aligned}
&\|D_2\mathcal{B}^{k+1}(f,\theta) - D_2\mathcal{B}^{k+1}(f',\theta)\|_{op} \\
&\leqslant \|D_1\mathcal{B}\big(\mathcal{B}^k(f,\theta),\theta\big) \circ \big(D_2\mathcal{B}^k(f,\theta) - D_2\mathcal{B}^k(f',\theta)\big)\|_{op} \\
&\quad + \Big\|\Big(D_1\mathcal{B}\big(\mathcal{B}^k(f,\theta),\theta\big) - D_1\mathcal{B}\big(\mathcal{B}^k(f',\theta),\theta\big)\Big) \circ D_2\mathcal{B}^k(f',\theta)\Big\|_{op} \\
&\quad + \|D_2\mathcal{B}\big(\mathcal{B}^k(f,\theta),\theta\big) - D_2\mathcal{B}\big(\mathcal{B}^k(f',\theta),\theta\big)\|_{op} \\
&\leqslant \|D_2\mathcal{B}^k(f,\theta) - D_2\mathcal{B}^k(f',\theta)\|_{op} \qquad\qquad\qquad\qquad\qquad\qquad (85) \\
&\quad + L_{\mathcal{A}}\|\mathcal{B}^k(f,\theta) - \mathcal{B}^k(f',\theta)\|_\infty \|D_2\mathcal{B}^k(f',\theta)\|_{op} \\
&\quad + L_1\big[\|\mathcal{B}^k(f,\theta) - \mathcal{B}^k(f',\theta)\|_\infty + \|\nabla\mathcal{B}^k(f,\theta) - \nabla\mathcal{B}^k(f',\theta)\|_{2,\infty}\big] \\
&\leqslant L_k[\|f - f'\|_\infty + \|\nabla f - \nabla f'\|_{2,\infty}] + L_{\mathcal{A}} \cdot M_k \cdot \|f - f'\|_\infty \\
&\quad + L_1\|f - f'\|_\infty + L_1\|\nabla\mathcal{B}^k(f,\theta) - \nabla\mathcal{B}^k(f',\theta)\|_{2,\infty} \\
&\leqslant (L_k + L_1 + L_{\mathcal{A}}M_k)[\|f - f'\|_\infty + \|\nabla f - \nabla f'\|_{2,\infty}] + L_1\|\nabla\mathcal{B}^k(f,\theta) - \nabla\mathcal{B}^k(f',\theta)\|_{2,\infty}. \quad (86)
\end{aligned}
$$

Here in the third inequality, we use the induction for the first term, Lemma B.7 for the second term. Notice that $\nabla\mathcal{A}(f,\theta)$ is Lipschitz continuous w.r.t. $f$: Denote $k(x,y) := \exp\{-c(x,y)/\gamma\}$. For any fixed $x \in \mathcal{X}$,

$$\nabla\big(\mathcal{A}(f,\alpha)\big)(x) = \frac{\int_\mathcal{X} k(z,x)\exp\{f(z)/\gamma\}\nabla_1 c(x,z)\mathbf{d}\alpha(z)}{\int_\mathcal{X} k(z,x)\exp\{f(z)/\gamma\}\mathbf{d}\alpha(z)}, \qquad \# \frac{g_1(f)}{g_2(f)}$$

where we denote the numerator and denominator of the above expression by $g_1 : \mathcal{C}(\mathcal{X}) \to \mathbb{R}^q$ and $g_2 : \mathcal{C}(\mathcal{X}) \to \mathbb{R}$. From the boundedness of $g_1$ and $g_2$, the Lipschitz continuity of $g_1$ and $g_2$ w.r.t. to $f$, and the fact that $g_2$ is positive and bounded away from zero, we conclude that there exists some constant $L_{\mathcal{A},f}$ such that for any $x \in \mathcal{X}$ (this follows similarly as (63))

$$\|\nabla\big(\mathcal{A}(f,\alpha)\big)(x) - \nabla\big(\mathcal{A}(f',\alpha)\big)(x)\| \leqslant L_{\mathcal{A},f}\|f - f'\|_\infty. \qquad (87)$$

Recall that $\mathcal{B}^k$ is the compositions of operators in the form of $\mathcal{A}$. Consequently, we have that

$$\|\nabla\mathcal{B}^k(f,\theta) - \nabla\mathcal{B}^k(f',\theta)\|_{2,\infty} \leqslant L_{\mathcal{A},f}\|f - f'\|_\infty.$$

Plugging this result into (86), we prove that the induction holds for $k+1$:

$$\|D_2\mathcal{B}^{k+1}(f,\theta) - D_2\mathcal{B}^{k+1}(f',\theta)\|_{op} \leqslant (L_k + L_1 + L_{\mathcal{A}}M_k + L_1L_{\mathcal{A},f})[\|f - f'\|_\infty + \|\nabla f - \nabla f'\|_{2,\infty}].$$

Consequently, for any finite $l$, we have $\|B_f - B_{f'}\| \leqslant L_l[\|f - f'\|_\infty + \|\nabla f - \nabla f'\|_{2,\infty}]$, where $L_l = l \cdot (L_1 + L_{\mathcal{A}}M_k + L_1L_{\mathcal{A},f})$.

**Lemma B.11.** *Under Assumption B.1, for $f \in \mathcal{C}(\mathcal{X})$ such $\|f\|_\infty \leqslant M_c$, there exists constant $L_{\mathcal{A}}$ such that $D_1\mathcal{A}(f,\alpha_\theta)$ is $L_{\mathcal{A}}$-Lipschitz continuous with respect to its first variable.*

*Proof.* Let $g \in \mathcal{C}(\mathcal{X})$ any function. Denote $\mathcal{T}_y(x,f) := \exp(-c(x,y)/\gamma)\exp(f(x)/\gamma)$. For a fixed point $y \in \mathcal{X}$ and any function $g \in \mathcal{C}(\mathcal{X})$, we compute that

$$\big(D_1\mathcal{A}(f,\theta)[g]\big)(y) = \frac{\int_\mathcal{X} \mathcal{T}_y(x,f)g(x)\mathbf{d}\alpha_\theta(x)}{\int_\mathcal{X} \mathcal{T}_y(x,f)\mathbf{d}\alpha_\theta(x)}, \qquad \# \frac{g_1(f)}{g_2(f)}$$

where we denote the numerator and denominator of the above expression by $g_1 : \mathcal{C}(\mathcal{X}) \to \mathbb{R}^q$ and $g_2 : \mathcal{C}(\mathcal{X}) \to \mathbb{R}$. From the boundedness of $g_1$ and $g_2$, the Lipschitz continuity of $g_1$ and $g_2$ w.r.t. to $f$, and the fact that $g_2$ is positive and bounded away from zero, we conclude that there exists some constant $L_{\mathcal{A}}$ such that for any $x \in \mathcal{X}$ (this follows similarly as (63)).

$\square$

**Analyze the second term of** (72). We bound the second term of (72) using Lemma B.10:

$$\|D_2\mathcal{B}(\mathcal{B}^{l-1}(f,\theta),\theta) - D_2\mathcal{B}(\mathcal{B}^{l-1}(f',\theta),\theta)\|_{op}$$
$$\leqslant L_1[\|\mathcal{B}^{l-1}(f,\theta) - \mathcal{B}^{l-1}(f',\theta)\|_\infty + \|\nabla\mathcal{B}^{l-1}(f,\theta) - \nabla\mathcal{B}^{l-1}(f',\theta)\|_{2,\infty}]$$
$$\leqslant L_1[\|f - f'\|_\infty + L_{\mathcal{A},f}\|f - f'\|_\infty] = L_1 \cdot (1 + L_{\mathcal{A},f})\|f - f'\|_\infty,$$

where we use (87) in the second inequality.

Combing the analysis for the two terms of (72), we conclude the result.

$\square$

## B.4 Proof of Theorem 5.1

We prove that the approximation error of $\nabla_\theta^2 \mathrm{OT}_\gamma(\alpha_\theta, \beta)$ using the estimated Sinkhorn potential $f_\theta^\epsilon$ and the estimated Fréchet derivative $g_\theta^\epsilon$ is of the order

$$\mathcal{O}(\|f_\theta^\epsilon - f_\theta\|_\infty + \|\nabla f_\theta^\epsilon - \nabla f_\theta\|_{2,\infty} + \|\nabla^2 f_\theta^\epsilon - \nabla^2 f_\theta\|_{op,\infty} + \|g_\theta^\epsilon - Df_\theta\|_{op}).$$

The other term $\nabla_\theta^2 \mathrm{OT}_\gamma(\alpha_\theta, \alpha_\theta)$ is handled in a similar manner.

Recall the simplified expression of $\nabla_\theta^2 \mathrm{OT}_\gamma(\alpha_\theta, \beta)$ in (52). Given the estimator $f_\theta^\epsilon$ ($g_\theta^\epsilon$) of $f_\theta$ ($Df_\theta$), we need to prove the following bounds of differences in terms of the estimation accuracy: For any $h_1, h_2 \in \mathbb{R}^d$,

$$|D_{11}^2\mathcal{H}_1(f_\theta,\theta)[Df_\theta[h_1], Df_\theta[h_2]] - D_{11}^2\mathcal{H}_1(f_\theta^\epsilon,\theta)[g_\theta^\epsilon[h_1], g_\theta^\epsilon[h_2]]|$$
$$= \mathcal{O}\left(\|h_1\| \cdot \|h_2\| \cdot (\|f_\theta^\epsilon - f_\theta\|_\infty + \|g_\theta^\epsilon - Df_\theta\|_{op})\right), \tag{88}$$
$$\|D_{22}^2\mathcal{H}_1(f_\theta,\theta) - D_{22}^2\mathcal{H}_1(f_\theta^\epsilon,\theta)\|_{op}$$
$$= \mathcal{O}\left(\|f_\theta^\epsilon - f_\theta\|_\infty + \|\nabla f_\theta^\epsilon - \nabla f_\theta\|_{2,\infty} + \|\nabla^2 f_\theta^\epsilon - \nabla^2 f_\theta\|_{op,\infty}\right). \tag{89}$$

Note that from the definition of the operator norm the first results is equivalent to the bound in the operator norm. Using Propositions 5.2 and 5.3 and Lemmas B.3, B.4, we know that we can compute the estimators $f_\theta^\epsilon$ and $g_\theta^\epsilon$ such that $\|f_\theta^\epsilon - f_\theta\|_\infty \leqslant \epsilon$, $\|\nabla f_\theta^\epsilon - \nabla f_\theta\|_{2,\infty} \leqslant \epsilon$, and $\|\nabla^2 f_\theta^\epsilon - \nabla^2 f_\theta\|_{op,\infty} \leqslant \epsilon$, and $\|g_\theta^\epsilon - Df_\theta\|_{op} \leqslant \epsilon$ in logarithm time $\mathcal{O}(\log\frac{1}{\epsilon})$. Together with (88) and (89) proved above, we can compute an $\epsilon$-accurate estimation of $\nabla_\theta^2 \mathrm{OT}_\gamma(\alpha_\theta, \beta)$ (in the operator norm) in logarithm time $\mathcal{O}(\log\frac{1}{\epsilon})$.

**Bounding** (88). Recall the definition of $D_{11}^2\mathcal{H}_1(f_\theta,\theta)[Df_\theta[h_1], Df_\theta[h_2]]$ in (56). Denote

$$A_1 = D_{11}^2\mathcal{A}(f_\theta, \alpha_\theta), v_1 = Df_\theta[h_1], v_2 = Df_\theta[h_2],$$
$$A_2 = D_{11}^2\mathcal{A}(f_\theta^\epsilon, \alpha_\theta), u_1 = g_\theta^\epsilon[h_1], u_2 = g_\theta^\epsilon[h_2].$$

Based on these definitions, we have

$$D_{11}^2\mathcal{H}_1(f_\theta,\theta)[Df_\theta[h_1], Df_\theta[h_2]] = \int_\mathcal{X} A_1[v_1, v_2](y)\mathbf{d}\beta(y)$$

$$D_{11}^2\mathcal{H}_1(f_\theta^\epsilon,\theta)[g_\theta^\epsilon[h_1], g_\theta^\epsilon[h_2]] = \int_\mathcal{X} A_2[u_1, u_2](y)\mathbf{d}\beta(y).$$

Using the triangle inequality, we have

$$\|A_1[v_1, v_2] - A_2[u_1, u_2]\|_\infty \tag{90}$$
$$\leqslant \|A_1[v_1 - u_1, v_2]\|_\infty + \|A_1[u_1, v_2 - u_2]\|_\infty + \|(A_1 - A_2)[u_1, u_2]\|_\infty.$$

We bound the three terms on the R.H.S. individually.

For the first term on the R.H.S. of (90), we recall the explicit expression of $A_1[v_1, v_2](y)$ in (55) as

$$A_1[v_1, v_2](y) = \frac{\int_\mathcal{X} \mathcal{T}_y(x, f_\theta)v_1(x)v_2(x)\mathbf{d}\alpha_\theta(x)}{\gamma\int_\mathcal{X} \mathcal{T}_y(x, f_\theta)\mathbf{d}\alpha_\theta(x)} - \frac{\int_{\mathcal{X}^2} \mathcal{T}_y(x, f_\theta)\mathcal{T}_y(x', f_\theta)v_1(x)v_2(x')\mathbf{d}\alpha_\theta(x)\mathbf{d}\alpha_\theta(x')}{\gamma\left[\int_\mathcal{X} \mathcal{T}_y(x, f_\theta)\mathbf{d}\alpha_\theta(x)\right]^2}.$$

Here we recall $\mathcal{T}_y(x,f) := \exp(-c(x,y)/\gamma)\exp(f(x)/\gamma)$. We bound using the facts that $\mathcal{T}_y(x,f_\theta)$ is bounded from above and bounded away from zero

$$|A_1[v_1 - u_1, v_2](y)| \leqslant \left| \frac{\int_{\mathcal{X}} \mathcal{T}_y(x, f_\theta)\big(v_1(x) - u_1(x)\big)v_2(x)\mathbf{d}\alpha_\theta(x)}{\gamma \int_{\mathcal{X}} \mathcal{T}_y(x, f_\theta)\mathbf{d}\alpha_\theta(x)} \right|$$

$$+ \left| \frac{\int_{\mathcal{X}^2} \mathcal{T}_y(x, f_\theta)\mathcal{T}_y(x', f_\theta)\big(v_1(x) - u_1(x)\big)v_2(x')\mathbf{d}\alpha_\theta(x)\mathbf{d}\alpha_\theta(x')}{\gamma \left[ \int_{\mathcal{X}} \mathcal{T}_y(x, f_\theta)\mathbf{d}\alpha_\theta(x)\right]^2} \right|$$

$$= \mathcal{O}(\|v_1 - u_1\|_\infty \cdot \|v_2\|_\infty).$$

Further, we have $\|u_1 - v_1\|_\infty = \mathcal{O}(\|Df_\theta - g_\theta^\epsilon\|_{op} \cdot \|h_1\|)$ and $\|v_1\|_\infty = \mathcal{O}(\|h_2\|)$. Consequently, the first term on the R.H.S. of (90) is of order $\mathcal{O}(\|Df_\theta - g_\theta^\epsilon\|_{op} \cdot \|h_1\| \cdot \|h_2\|)$.

Following the same argument, we have the second term on the R.H.S. of (90) is of order $\mathcal{O}(\|Df_\theta - g_\theta^\epsilon\|_{op} \cdot \|h_1\| \cdot \|h_2\|)$.

To bound the third term on the R.H.S. of (90), denote

$$A_{11}[u_1, u_2] := \frac{\int_{\mathcal{X}} \mathcal{T}_y(x, f_\theta)u_1(x)u_2(x)\mathbf{d}\alpha_\theta(x)}{\gamma \int_{\mathcal{X}} \mathcal{T}_y(x, f_\theta)\mathbf{d}\alpha_\theta(x)} \text{ and } A_{21}[u_1, u_2] := \frac{\int_{\mathcal{X}} \mathcal{T}_y(x, f_\theta^\epsilon)u_1(x)u_2(x)\mathbf{d}\alpha_\theta(x)}{\gamma \int_{\mathcal{X}} \mathcal{T}_y(x, f_\theta^\epsilon)\mathbf{d}\alpha_\theta(x)},$$

and denote

$$A_{12}[u_1, u_2] := \frac{\int_{\mathcal{X}} \mathcal{T}_y(x, f_\theta)u_1(x)\mathbf{d}\alpha_\theta(x) \int_{\mathcal{X}} \mathcal{T}_y(x', f_\theta)u_2(x')\mathbf{d}\alpha_\theta(x')}{\gamma \left[\int_{\mathcal{X}} \mathcal{T}_y(x, f_\theta)\mathbf{d}\alpha_\theta(x)\right]^2},$$

$$\text{and } A_{22}[u_1, u_2] := \frac{\int_{\mathcal{X}} \mathcal{T}_y(x, f_\theta^\epsilon)u_1(x)\mathbf{d}\alpha_\theta(x) \int_{\mathcal{X}} \mathcal{T}_y(x', f_\theta^\epsilon)u_2(x')\mathbf{d}\alpha_\theta(x')}{\gamma \left[\int_{\mathcal{X}} \mathcal{T}_y(x, f_\theta^\epsilon)\mathbf{d}\alpha_\theta(x)\right]^2}.$$

We show that both $|(A_{11} - A_{21})[u_1, u_2]|$ and $|(A_{12} - A_{22})[u_1, u_2]|$ are of order $\mathcal{O}(\|Df_\theta - g_\theta^\epsilon\|_{op} \cdot \|h_1\| \cdot \|h_2\|)$. This then implies $|(A_1 - A_2)[u_1, u_2]| = \mathcal{O}(\|Df_\theta - g_\theta^\epsilon\|_{op} \cdot \|h_1\| \cdot \|h_2\|)$.

With the argument similar to (63), we obtain that $|(A_{11} - A_{21})[u_1, u_2]| = \mathcal{O}(\|Df_\theta - g_\theta^\epsilon\|_{op} \cdot \|u_1\| \cdot \|u_2\|)$ using the boundedness and Lipschitz continuity of the numerator and denominator of $A_{11}[u_1, u_2]$ w.r.t. to $f_\theta$ and the fact that the denominator is positive and bounded away from zero (see the discussion following (63)). Further, since both $Df_\theta$ and $g_\theta^\epsilon$ are bounded linear operators, we have that $u_1 = \mathcal{O}(h_1)$ and $u_2 = \mathcal{O}(h_2)$. Consequently, we prove that $|(A_{11} - A_{21})[u_1, u_2]| = \mathcal{O}(\|f_\theta - f_\theta^\epsilon\|_{op} \cdot \|h_1\| \cdot \|h_2\|)$.

Similarly, we can prove that $|(A_{12} - A_{22})[u_1, u_2]| = \mathcal{O}(\|f_\theta - f_\theta^\epsilon\|_{op} \cdot \|h_1\| \cdot \|h_2\|)$.

Altogether, we have proved (88).

**Bounding** (89). Recall that the expression of $D_{22}^2\mathcal{H}_1(f, \theta)$ in (58). For a fixed $y \in \mathcal{X}$ and a fixed $z' \in \mathcal{Z}$, denote (recall that $u_z(\theta, f) = \nabla_1 c\big(T_\theta(z), y\big) - \nabla f\big(T_\theta(z)\big)$)

$$B_1(f) = \nabla_\theta^2 T_\theta(z') \times_1 \nabla f\big(T_\theta(z')\big)$$

$$B_2(f) = \nabla_\theta T_\theta(z')^\top \nabla^2 f\big(T_\theta(z')\big)\nabla_\theta T_\theta(z')$$

$$B_3(f) = \frac{\int_{\mathcal{Z}} \mathcal{T}_y\big(T_\theta(z), f\big)\nabla_\theta T_\theta(z)^\top u_z(\theta, f)u_z(\theta, f)^\top \nabla_\theta T_\theta(z)\mathbf{d}\mu(z)}{\int_{\mathcal{Z}} \mathcal{T}_y\big(T_\theta(z), f\big)\mathbf{d}\mu(z)}$$

$$B_4(f) = \frac{\int_{\mathcal{Z}} \mathcal{T}_y\big(T_\theta(z), f\big)\nabla_\theta^2 T_\theta(z) \times_1 u_z(\theta, f)\mathbf{d}\mu(z)}{\int_{\mathcal{Z}} \mathcal{T}_y\big(T_\theta(z), f\big)\mathbf{d}\mu(z)}$$

$$B_5(f) = \frac{\int_{\mathcal{Z}} \mathcal{T}_y\big(T_\theta(z), f\big)\nabla_\theta T_\theta(z)^\top \nabla_{11} c(T_\theta(z), y)\nabla_\theta T_\theta(z)\mathbf{d}\mu(z)}{\int_{\mathcal{Z}} \mathcal{T}_y\big(T_\theta(z), f\big)\mathbf{d}\mu(z)}$$

$$B_6(f) = -\frac{\int_{\mathcal{Z}} \mathcal{T}_y\big(T_\theta(z), f\big)\nabla_\theta T_\theta(z)^\top \nabla^2 f\big(T_\theta(z)\big)\nabla_\theta T_\theta(z)\mathbf{d}\mu(z)}{\int_{\mathcal{Z}} \mathcal{T}_y\big(T_\theta(z), f\big)\mathbf{d}\mu(z)}$$

$$B_7(f) = \frac{\int_{\mathcal{Z}} \mathcal{T}_y\big(T_\theta(z), f\big)\nabla_\theta T_\theta(z)^\top u_z(\theta, f)\mathbf{d}\mu(z) \left[\int_{\mathcal{Z}} \mathcal{T}_y\big(T_\theta(z), f\big)\nabla_\theta T_\theta(z)^\top u_z(\theta, f)\mathbf{d}\mu(z)\right]^\top}{\left[\int_{\mathcal{Z}} \mathcal{T}_y\big(T_\theta(z), f\big)\mathbf{d}\mu(z)\right]^2}$$

Based on these definitions, we have

$$D_{22}^2 \mathcal{H}_1(f, \theta) = \int_{\mathcal{Z}} \sum_{i=1}^2 B_i(f) \mathbf{d}\mu(z') + \int_{\mathcal{X}} \sum_{i=3}^7 B_i(f) \mathbf{d}\beta(y).$$

We bound the above seven terms individually.

**Assumption B.6.** *For a fixed $z \in \mathcal{Z}$ and $\theta \in \Theta$, use $\nabla_\theta^2 T_\theta(z) \in T(\mathbb{R}^d \times \mathbb{R}^d \to \mathbb{R}^q)^2$ to denote the second-order Jacobian of $T_\theta(z)$ w.r.t. $\theta$. Use $\times_1$ to denote the tensor product along the first dimension. For any two vectors $g, g' \in \mathbb{R}^d$, we assume that*

$$\|\nabla_\theta^2 T_\theta(z) \times_1 g - \nabla_\theta^2 T_\theta(z) \times_1 g'\|_{op} = \mathcal{O}(\|g - g'\|). \tag{91}$$

For the first term, using the boundedness of $\nabla_\theta^2 T_\theta(z')$ (Assumption B.6), we have that

$$\|B_1(f_\theta) - B_1(f_\theta^\epsilon)\|_{op} = \mathcal{O}(\|\nabla f_\theta - \nabla f_\theta^\epsilon\|_{2,\infty}).$$

For the second term, using the boundedness of $\nabla_\theta T_\theta(z')$, we have that

$$\|B_2(f_\theta) - B_2(f_\theta^\epsilon)\|_{op} = \mathcal{O}(\|\nabla^2 f_\theta - \nabla^2 f_\theta^\epsilon\|_{op,\infty}).$$

For the third term, note that $\|u_z(\theta, f_\theta) - u_z(\theta, f_\theta^\epsilon)\| = \mathcal{O}(\|\nabla f_\theta - \nabla f_\theta^\epsilon\|_{2,\infty})$. With the argument similar to (63), we obtain that

$$\|B_3(f_\theta) - B_3(f_\theta^\epsilon)\|_{op} = \mathcal{O}(\|f_\theta - f_\theta^\epsilon\|_\infty + \|\nabla f_\theta - \nabla f_\theta^\epsilon\|_{2,\infty}). \tag{92}$$

This is from the boundedness and Lipschitz continuity of $\mathcal{T}_y(T_\theta(z), f)$ w.r.t. to $f$, the boundedness and Lipschitz continuity of $u_z(\theta, f)$ w.r.t. $\nabla f$, and the fact that $\mathcal{T}_y(T_\theta(z), f)$ is positive and bounded away from zero.

For the forth term, following the similar argument as the third term and using the boundedness of $\nabla_\theta^2 T_\theta(z)$, we have that

$$\|B_4(f_\theta) - B_4(f_\theta^\epsilon)\|_{op} = \mathcal{O}(\|f_\theta - f_\theta^\epsilon\|_\infty + \|\nabla f_\theta - \nabla f_\theta^\epsilon\|_{2,\infty}). \tag{93}$$

For the fifth term, following the similar argument as the third term and using the boundedness of $\nabla_\theta T_\theta(z)$ and $\nabla_{11} c(T_\theta(z), y)$, we have that

$$\|B_5(f_\theta) - B_5(f_\theta^\epsilon)\|_{op} = \mathcal{O}(\|f_\theta - f_\theta^\epsilon\|_\infty). \tag{94}$$

For the sixth term, following the similar argument as the third term and using the boundedness of $\nabla_\theta T_\theta(z)$, we have that

$$\|B_6(f_\theta) - B_6(f_\theta^\epsilon)\|_{op} = \mathcal{O}(\|f_\theta - f_\theta^\epsilon\|_\infty + \|\nabla^2 f_\theta - \nabla^2 f_\theta^\epsilon\|_{op,\infty}). \tag{95}$$

For the last term, following the similar argument as the third term and using the boundedness of $\nabla_\theta T_\theta(z)$, we have that

$$\|B_7(f_\theta) - B_7(f_\theta^\epsilon)\|_{op} = \mathcal{O}(\|f_\theta - f_\theta^\epsilon\|_\infty + \|\nabla f_\theta - \nabla f_\theta^\epsilon\|_{2,\infty}). \tag{96}$$

Combing the above results, we obtain (89).

## C  eSIM appendix

### C.1  Proof of Theorem 6.1

In this section, we use $f_\theta^\mu$ to denote the Sinkhorn potential to $\mathrm{OT}_\gamma(T_{\theta\sharp}\mu, \beta)$. This allows us to emphasize the continuity of its Fréchet derivative w.r.t. the underlying measure $\mu$. Similarly, we write $\mathcal{B}_\mu(f, \theta)$ and $\mathcal{E}_\mu(f, \theta)$ instead of $\mathcal{B}(f, \theta)$ and $\mathcal{E}(f, \theta)$, which are used to characterize the fixed point property of the Sinkhorn potential.

To prove Theorem 6.1, we need the following lemmas.

**Lemma C.1.** *The Sinkhorn potential $f_\theta^\mu$ is Lipschitz continuous with respect to $\mu$:*
$$\|f_\theta^\mu - f_\theta^{\bar\mu}\|_\infty = \mathcal{O}(d_{bl}(\mu, \bar\mu)). \tag{97}$$

**Lemma C.2.** *The gradient of the Sinkhorn potential $f_\theta^\mu$ is Lipschitz continuous with respect to $\mu$:*
$$\|\nabla f_\theta^\mu - \nabla f_\theta^{\bar\mu}\|_{2,\infty} = \mathcal{O}(d_{bl}(\mu, \bar\mu)). \tag{98}$$

**Lemma C.3.** *The Hessian of the Sinkhorn potential $f_\theta^\mu$ is Lipschitz continuous with respect to $\mu$:*
$$\|\nabla^2 f_\theta^\mu - \nabla^2 f_\theta^{\bar\mu}\|_{op,\infty} = \mathcal{O}(d_{bl}(\mu, \bar\mu)). \tag{99}$$

**Lemma C.4.** *The Fréchet derivative of the Sinkhorn potential $f_\theta^\mu$ w.r.t. the parameter $\theta$, i.e. $Df_\theta^\mu$, is Lipschitz continuous with respect to $\mu$:*
$$\|Df_\theta^\mu - Df_\theta^{\bar\mu}\|_{op} = \mathcal{O}(d_{bl}(\mu, \bar\mu)). \tag{100}$$

Once we have these lemmas, we can prove 6.1 in the same way as the proof of 5.1 in Appendix B.4.

### C.2  Proof of Lemma C.1

Note that from the definition of the bounded Lipschitz distance, we have
$$d_{bl}(\alpha, \bar\alpha) = \sup_{\|\xi\|_{bl} \leqslant 1} |\langle \xi, \alpha\rangle - \langle \xi, \bar\alpha\rangle| = \sup_{\|\xi\|_{bl} \leqslant 1} |\langle \xi \circ T_\theta, \mu\rangle - \langle \xi \circ T_\theta, \bar\mu\rangle|$$
$$\leqslant \sup_{\|\xi\|_{bl} \leqslant 1} \|\xi \circ T_\theta\|_{bl} \cdot d_{bl}(\mu, \bar\mu) \leqslant G_T \cdot d_{bl}(\mu, \bar\mu), \tag{101}$$

where we use $\|\xi \circ T_\theta\|_{lip} \leqslant G_T$ from Assumption B.5.

We have Lemma C.1 by combining the above results with the following lemma.

**Lemma C.5.** *Under Assumption B.1 and Assumption B.2, the Sinkhorn potential is Lipschitz continuous with respect to the bounded Lipschitz metric: Given measures $\alpha$, $\alpha'$ and $\beta$, we have*
$$\|f_{\alpha,\beta} - f_{\alpha',\beta}\|_\infty \leqslant G_{bl} d_{bl}(\alpha', \alpha) \quad \text{and} \quad \|g_{\alpha,\beta} - g_{\alpha',\beta'}\|_\infty \leqslant G_{bl} d_{bl}(\alpha', \alpha).$$
*where $G_{bl} = 2\gamma \exp(2M_c/\gamma)G'_{bl}/(1 - \lambda^2)$ with $G'_{bl} = \max\{\exp(3M_c/\gamma), 2G_c \exp(3M_c/\gamma)/\gamma\}$ and $\lambda = \frac{\exp(M_c/\gamma) - 1}{\exp(M_c/\gamma) + 1}$.*

*Proof.* Let $(f, g)$ and $(f', g')$ be the Sinkhorn potentials to $\mathrm{OT}_\gamma(\alpha, \beta)$ and $\mathrm{OT}_\gamma(\alpha', \beta)$ respectively. Denote $u := \exp(f/\gamma)$, $v := \exp(g/\gamma)$ and $u' := \exp(f'/\gamma)$, $v' := \exp(g'/\gamma)$. From Lemma C.7, $u$ is bounded in terms of the $L^\infty$ norm:
$$\|u\|_\infty = \max_{x \in \mathcal{X}} |u(x)| = \max_{x \in \mathcal{X}} \exp(f/\gamma) \leqslant \exp(2M_c/\gamma),$$

which also holds for $v, u', v'$. Additionally, from Lemma C.8, $\nabla u$ exists and $\|\nabla u\|$ is bounded:
$$\max_x \|\nabla u(x)\| = \max_x \frac{1}{\gamma}|u(x)|\|\nabla f(x)\| \leqslant \frac{1}{\gamma}\|u(x)\|_\infty \max_x \|\nabla f(x)\| \leqslant \frac{G_c \exp(2M_c/\gamma)}{\gamma}.$$

Define the mapping $A_\alpha\mu := 1/(L_\alpha\mu)$ with
$$L_\alpha\mu = \int_{\mathcal{X}} l(\cdot, y)\mu(y)\mathbf{d}\alpha(y),$$

where $l(x, y) := \exp(-c(x, y)/\gamma)$. From Assumption B.1, we have $\|l\|_\infty \leqslant \exp(M_c/\gamma)$ and from Assumption B.2 we have $\|\nabla_x l(x, y)\| \leqslant \exp(M_c/\gamma)\frac{G_c}{\gamma}$. From the optimality condition of $f$ and $g$, we have $v = A_\alpha u$ and $u = A_\beta v$. Similarly, $v' = A_{\alpha'}u'$ and $u' = A_\beta v'$. Recall the definition of the Hilbert metric in (60). Note that $d_H(\mu, \nu) = d_H(1/\mu, 1/\nu)$ if $\mu(x) > 0$ and $\nu(x) > 0$ for all $x \in \mathcal{X}$ and hence $d_H(L_\alpha\mu, L_\alpha\nu) = d_H(A_\alpha\mu, A_\alpha\nu)$. We recall the result in (61) using the above notations.

**Lemma C.6** (Birkhoff-Hopf Theorem Lemmens and Nussbaum [2012], see Lemma B.4 in Luise et al. [2019]). *Let* $\lambda = \frac{\exp(M_c/\gamma)-1}{\exp(M_c/\gamma)+1}$ *and* $\alpha \in \mathcal{M}_1^+(\mathcal{X})$. *Then for every* $u, v \in \mathcal{C}(\mathcal{X})$, *such that* $u(x) > 0, v(x) > 0$ *for all* $x \in \mathcal{X}$, *we have*

$$d_H(L_\alpha u, L_\alpha v) \leqslant \lambda d_H(u, v).$$

Note that

$$\|\log \mu - \log \nu\|_\infty \leqslant d_H(\mu, \nu) = \|\log \mu - \log \nu\|_\infty + \|\log \nu - \log \mu\|_\infty \leqslant 2\|\log \mu - \log \nu\|_\infty.$$

In the following, we derive upper bound for $d_H(\mu, \nu)$ and use such bound to analyze the Lipschitz continuity of the Sinkhorn potentials $f$ and $g$.
Construct $\tilde{v} := A_\alpha u'$. Using the triangle inequality (which holds since $v(x), v'(x), \tilde{v}(x) > 0$ for all $x \in \mathcal{X}$), we have

$$d_H(v, v') \leqslant d_H(v, \tilde{v}) + d_H(\tilde{v}, v') \leqslant \lambda d_H(u, u') + d_H(\tilde{v}, v'),$$

where the second inequality is due to Lemma C.6. Note that $u' = A_\beta v'$. Apply Lemma C.6 again to obtain

$$d_H(u, u') \leqslant \lambda d_H(v, v').$$

Together, we obtain

$$d_H(v, v') \leqslant \lambda^2 d_H(v, v') + d_H(\tilde{v}, v') + \lambda d_H(\tilde{u}, u') \leqslant \lambda^2 d_H(v, v') + d_H(\tilde{v}, v'),$$

which leads to

$$d_H(v, v') \leqslant \frac{1}{1-\lambda^2}[d_H(\tilde{v}, v')].$$

To bound $d_H(\tilde{v}, v')$, observe the following:

$$
\begin{aligned}
d_H(v', \tilde{v}) &= d_H(L_{\alpha'}u', L_\alpha u') \leqslant 2\|\log L_{\alpha'}u' - \log L_\alpha u'\|_\infty \\
&= 2\max_{x \in \mathcal{X}} |\nabla \log(a_x)([L_{\alpha'}u'](x) - [L_\alpha u'](x))| = 2\max_{x \in \mathcal{X}} \frac{1}{a_x}|[L_{\alpha'}u'](x) - [L_\alpha u'](x)| \\
&\leqslant 2\max\{\|1/L_{\alpha'}u'\|_\infty, \|1/L_\alpha u'\|_\infty\}\|L_{\alpha'}u' - L_\alpha u'\|_\infty, \quad\quad (102)
\end{aligned}
$$

where $a_x \in [[L_{\alpha'}u'](x), [L_\alpha u'](x)]]$ in the second line is from the mean value theorem. Further, in the inequality we use $\max\{\|1/L_\alpha u'\|_\infty, \|1/L_\alpha u'\|_\infty\} = \max\{\|A_{\alpha'}u'\|_\infty, \|A_\alpha u'\|_\infty\} \leqslant \exp(2M_c/\gamma)$. Consequently, all we need to bound is the last term $\|L_{\alpha'}u' - L_\alpha u'\|_\infty$.

We first note that $\forall x \in \mathcal{X}, \|l(x, \cdot)u'(\cdot)\|_{bl} < \infty$: In terms of $\|\cdot\|_\infty$

$$\|l(x, \cdot)u'(\cdot)\|_\infty \leqslant \|l(x, \cdot)\|_\infty \|u'\|_\infty \leqslant \exp(3M_c/\gamma) < \infty.$$

In terms of $\|\cdot\|_{lip}$, we bound

$$
\begin{aligned}
\|l(x, \cdot)u'(\cdot)\|_{lip} &\leqslant \|l(x, \cdot)\|_\infty \|u'\|_{lip} + \|l(x, \cdot)\|_{lip}\|u'\|_\infty \\
&\leqslant \exp(M_c/\gamma)\frac{G_c\exp(2M_c/\gamma)}{\gamma} + \exp(M_c/\gamma)\frac{G_c}{\gamma}\exp(2M_c/\gamma) = \frac{2G_c\exp(3M_c/\gamma)}{\gamma} < \infty.
\end{aligned}
$$

Together we have $\|l(x, y)u'(y)\|_{bl} \leqslant \max\{\exp(3M_c/\gamma), \frac{2G_c\exp(3M_c/\gamma)}{\gamma}\}$. From the definition of the operator $L_\alpha$, we have

$$\|L_{\alpha'}u' - L_\alpha u'\|_\infty = \max_x |\int_{\mathcal{X}} l(x, y)u'(y)\mathbf{d}\alpha'(y) - \int_{\mathcal{X}} l(x, y)u'(y)\mathbf{d}\alpha(y)| \leqslant \|l(x, y)u'(y)\|_{bl}d_{bl}(\alpha', \alpha).$$

All together we derive

$$d_H(v', v) \leqslant \frac{2\exp(2M_c/\gamma)\|l(x, y)u'(y)\|_{bl}}{1-\lambda^2} \cdot d_{bl}(\alpha', \alpha) \quad (\lambda = \frac{\exp(M_c/\gamma)-1}{\exp(M_c/\gamma)+1}).$$

Further, since $d_H(v', v) \geqslant \|\log v' - \log v\|_\infty = \frac{1}{\gamma}\|f' - f\|_\infty$, we have the result:

$$\|f' - f\|_\infty \leqslant \frac{2\gamma\exp(2M_c/\gamma)\|l(x, y)u'(y)\|_{bl}}{1-\lambda^2} \cdot d_{bl}(\alpha', \alpha).$$

Similar argument can be made for $\|g' - g\|_\infty$. $\qquad\qquad\qquad\qquad\qquad\qquad\square$

**Lemma C.7** (Boundedness of the Sinkhorn Potentials). *Let $(f, g)$ be the Sinkhorn potentials of problem* (6) *and assume that there exists $x_o \in \mathcal{X}$ such that $f(x_o) = 0$ (otherwise shift the pair by $f(x_o)$). Then, under Assumption B.1, $\|f\|_\infty \leqslant 2M_c$ and $\|g\|_\infty \leqslant 2M_c$.*

Next, we analyze the Lipschitz continuity of the Sinkhorn potential $f_{\alpha,\beta}(x)$ with respect to the input $x$.

Assumption B.2 implies that $\nabla_x c(x, y)$ exists and for all $x, y \in \mathcal{X}, \|\nabla_x c(x, y)\| \leqslant G_c$. It further ensures the Lipschitz-continuity of the Sinkhorn potential.

**Lemma C.8** (Proposition 12 of Feydy et al. [2019]). *Under Assumption B.2, for a fixed pair of measures $(\alpha, \beta)$, the corresponding Sinkhorn potential $f : \mathcal{X} \to \mathbb{R}$ is $G_c$-Lipschitz continuous, i.e. for $x_1, x_2 \in \mathcal{X}$*

$$|f_{\alpha,\beta}(x_1) - f_{\alpha,\beta}(x_2)| \leqslant G_c \|x_1 - x_2\|. \tag{103}$$

*Further, the gradient $\nabla f_{\alpha,\beta}$ exists at every point $x \in \mathcal{X}$, and $\|\nabla f_{\alpha,\beta}(x)\| \leqslant G_c, \forall x \in \mathcal{X}$.*

**Lemma C.9.** *Under Assumption B.3, for a fixed pair of measures $(\alpha, \beta)$, the gradient of the corresponding Sinkhorn potential $f : \mathcal{X} \to \mathbb{R}$ is Lipschitz continuous,*

$$\|\nabla f(x_1) - \nabla f(x_2)\| \leqslant L_f \|x_1 - x_2\|, \tag{104}$$

*where $L_f := \frac{4G_c^2}{\gamma} + L_c$.*

## C.3 Proof of Lemma C.2

We have Lemma C.2 by combining (101) with the following lemma.

**Lemma C.10** (Lemma C.2 restated). *Under Assumption B.1 and Assumption B.2, the gradient of the Sinkhorn potential is Lipschitz continuous with respect to the bounded Lipschitz metric: Given measures $\alpha$, $\alpha'$ and $\beta$, we have*

$$\|\nabla f_{\alpha,\beta} - \nabla f_{\alpha',\beta}\|_\infty = \mathcal{O}\big(d_{bl}(\alpha', \alpha)\big)$$

*Proof.* From the optimality condition of the Sinkhorn potentials, one have that

$$\int_{\mathcal{X}} h_{\alpha,\beta}(x, y)\mathbf{d}\beta(y) = 1, \text{with } h_{\alpha,\beta}(x, y) := \exp\left(\frac{1}{\gamma}\big(f_{\alpha,\beta}(x) + g_{\alpha,\beta}(y) - c(x, y)\big)\right). \tag{105}$$

Taking gradient w.r.t. $x$ on both sides of the above equation, the expression of $\nabla f_{\alpha,\beta}$ writes

$$\nabla f_{\alpha,\beta}(x) = \frac{\int_{\mathcal{X}} h_{\alpha,\beta}(x, y)\nabla_x c(x, y)\mathbf{d}\beta(y)}{\int_{\mathcal{X}} h_{\alpha,\beta}(x, y)\mathbf{d}\beta(y)} = \int_{\mathcal{X}} h_{\alpha,\beta}(x, y)\nabla_x c(x, y)\mathbf{d}\beta(y). \tag{106}$$

We have that $\forall x, y, h_{\alpha,\beta}(x)$ is Lipschitz continuous w.r.t. $\alpha$, which is due to the boundedness of $f_{\alpha,\beta}(x), g_{\alpha,\beta}(y)$ and the ground cost $c$, and Lemma C.1. Further, since $\|\nabla_x c(x, y)\|$ is bounded from Assumption B.2 we have the Lipschitz continuity of $\nabla f_{\alpha,\beta}$ w.r.t. $\alpha$, i.e.

$$\|\nabla f_{\alpha,\beta}(x) - \nabla f_{\alpha',\beta}(x)\| = \mathcal{O}\big(d_{bl}(\alpha', \alpha)\big).$$

$\square$

## C.4 Proof of Lemma C.3

We have Lemma C.3 by combining (101) with the following lemma.

**Lemma C.11** (Lemma C.3 restated). *Under Assumptions B.1-B.3, the Hessian of the Sinkhorn potential is Lipschitz continuous with respect to the bounded Lipschitz metric: Given measures $\alpha$, $\alpha'$ and $\beta$, we have*

$$\|\nabla^2 f_{\alpha,\beta} - \nabla^2 f_{\alpha',\beta}\|_{op,\infty} = \mathcal{O}\big(d_{bl}(\alpha', \alpha)\big)$$

*Proof.* Taking gradient w.r.t. $x$ on both sides of (106), the expression of $\nabla^2 f_{\alpha,\beta}$ writes

$$\nabla^2 f_{\alpha,\beta}(x) = \int_{\mathcal{X}} \frac{1}{\gamma} h_{\alpha,\beta}(x, y)(\nabla f_{\alpha,\beta}(x) - \nabla_x c(x, y))[\nabla_x c(x, y)]^\top + h_{\alpha,\beta}(x, y)\nabla_{xx}^2 c(x, y)\mathbf{d}\beta(y).$$

From the boundedness of $h_{\alpha,\beta}$, $\nabla f_{\alpha,\beta}$ and $\nabla_x c$, and the Lipschitz continuity of $h_{\alpha,\beta}$ and $\nabla f_{\alpha,\beta}$ w.r.t. $\alpha$, we have that the first integrand of $\nabla^2 f_{\alpha,\beta}$ is Lipschitz continuous w.r.t. $\alpha$. Further, combining the boundedness of $\|\nabla^2_{xx} c(x,y)\|$ from Assumption B.3 and the Lipschitz continuity of $h_{\alpha,\beta}$ w.r.t. $\alpha$, we have the Lipschitz continuity of $\nabla^2 f_{\alpha,\beta}(x)$, i.e.

$$\|\nabla^2 f_{\alpha,\beta}(x) - \nabla^2 f_{\alpha',\beta}(x)\| = \mathcal{O}\big(d_{bl}(\alpha',\alpha)\big).$$

$\square$

## C.5  Proof of Lemma C.4

The optimality of the Sinkhorn potential $f_\theta^\mu$ can be restated as

$$f_\theta^\mu = \mathcal{B}_\mu(f_\theta^\mu, \theta), \tag{107}$$

where we recall the definition of $\mathcal{B}_\mu$ in (18)

$$\mathcal{B}_\mu(f,\theta) = \mathcal{A}\big(\mathcal{A}(f, T_{\theta\sharp}\mu), \beta_\mu\big). \tag{108}$$

Note that it is possible that $\beta_\mu$ depends on $\mu$, which is the case in $\mathrm{OT}_\gamma(\alpha_\theta, \alpha_{\theta^t})$ as $\beta_\mu = \alpha_{\theta^t} = T_{\theta^t\sharp}\mu$. Under Assumption B.1, let $\lambda = \frac{e^{M_c/\gamma}-1}{e^{M_c/\gamma}+1}$. By repeating the above fixed point iteration (107) $l = \lceil \log_\lambda \frac{1}{3}\rceil/2$ times, we have that

$$f_\theta^\mu = \mathcal{E}_\mu(f_\theta^\mu, \theta), \tag{109}$$

where $\mathcal{E}_\mu(f,\theta) = \mathcal{B}_\mu^l(f,\theta) = \mathcal{B}_\mu\big(\cdots\mathcal{B}_\mu(f,\theta)\cdots,\theta\big)$ is the $l$ times composition of $\mathcal{B}_\mu$ in its first variable. We have from (68)

$$||D^1\mathcal{E}_\mu(f,\theta)||_{op} \leqslant \frac{2}{3}, \tag{110}$$

where we recall for a (linear) operator $\mathcal{C}: \mathcal{C}(\mathcal{X}) \to \mathcal{C}(\mathcal{X})$, $\|\mathcal{C}\|_{op} := \max_{f\in\mathcal{C}(\mathcal{X})} \frac{\|\mathcal{C}f\|_\infty}{\|f\|_\infty}$.

Let $h \in \mathbb{R}^d$ be any direction. Taking Fréchet derivative w.r.t. $\theta$ on both sides of (109), we derive

$$Df_\theta^\mu[h] = D_1\mathcal{E}_\mu(f_\theta^\mu, \theta)\big[Df_\theta^\mu[h]\big] + D_2\mathcal{E}_\mu(f_\theta^\mu, \theta)[h]. \tag{111}$$

Using the triangle inequality, we bound

$$\begin{aligned}
&\|Df_\theta^\mu[h] - Df_\theta^{\bar\mu}[h]\|_\infty \\
\leqslant\ & \|D_1\mathcal{E}_\mu(f_\theta^\mu, \theta)\big[Df_\theta^\mu[h]\big] - D_1\mathcal{E}_{\bar\mu}(f_{\theta,\bar\mu}, \theta)\big[Df_\theta^{\bar\mu}[h]\big]\|_\infty \\
& + \|D_2\mathcal{E}_\mu(f_\theta^\mu, \theta)[h] - D_2\mathcal{E}_{\bar\mu}(f_\theta^{\bar\mu}, \theta)[h]\|_\infty \\
\leqslant\ & \|D_1\mathcal{E}_\mu(f_\theta^\mu, \theta)\big[Df_\theta^\mu[h]\big] - D_1\mathcal{E}_\mu(f_\theta^\mu, \theta)\big[Df_\theta^{\bar\mu}[h]\big]\|_\infty && \text{①} \\
& + \|D_1\mathcal{E}_\mu(f_{\theta,\mu}, \theta)\big[Df_\theta^{\bar\mu}[h]\big] - D_1\mathcal{E}_\mu(f_{\theta,\bar\mu}, \theta)\big[Df_\theta^{\bar\mu}[h]\big]\|_\infty && \text{②} \\
& + \|D_1\mathcal{E}_\mu(f_{\theta,\bar\mu}, \theta)\big[Df_\theta^{\bar\mu}[h]\big] - D_1\mathcal{E}_{\bar\mu}(f_{\theta,\bar\mu}, \theta)\big[Df_\theta^{\bar\mu}[h]\big]\|_\infty && \text{③} \\
& + \|D_2\mathcal{E}_\mu(f_\theta^\mu, \theta)[h] - D_2\mathcal{E}_{\bar\mu}(f_\theta^{\bar\mu}, \theta)[h]\|_\infty. && \text{④}
\end{aligned} \tag{112}$$

The following subsections analyze ① to ④ individually. In summary, we have

$$\text{①} \leqslant \frac{2}{3}\|Df_\theta^\mu[h] - Df_\theta^{\bar\mu}[h]\|_\infty, \tag{113}$$

and ②, ③, ④ are all of order $\mathcal{O}(d_{bl}(\mu,\bar\mu) \cdot \|h\|)$. Therefore we conclude

$$\frac{1}{3}\|Df_\theta^\mu[h] - Df_\theta^{\bar\mu}[h]\|_\infty = \mathcal{O}(d_{bl}(\mu,\bar\mu) \cdot \|h\|) \Rightarrow \|Df_\theta^\mu - Df_\theta^{\bar\mu}\|_{op} = \mathcal{O}(d_{bl}(\mu,\bar\mu)). \tag{114}$$

### C.5.1  Bounding ①

From the linearity of $D_1\mathcal{E}_\mu(f_\theta^\mu, \theta)$ and (110), we bound

$$\begin{aligned}
\text{①} &= \|D_1\mathcal{E}_\mu(f_\theta^\mu, \theta)\big[Df_\theta^\mu[h] - Df_\theta^{\bar\mu}[h]\big]\|_\infty \\
&\leqslant \|D_1\mathcal{E}_\mu(f_\theta^\mu, \theta)\|_{op}\|Df_\theta^\mu[h] - Df_\theta^{\bar\mu}[h]\|_\infty \leqslant \frac{2}{3}\|Df_\theta^\mu[h] - Df_\theta^{\bar\mu}[h]\|_\infty.
\end{aligned}$$

### C.5.2 Bounding ②

From Lemma B.8, we know that $D_1\mathcal{B}_\mu(f,\theta)$ is Lipschitz continuous w.r.t. its first variable:

$$\|D_1\mathcal{B}_\mu(f,\theta) - D_1\mathcal{B}_\mu(f',\theta)\|_{op} = \mathcal{O}(\|f - f'\|_\infty). \tag{115}$$

Recall that $\mathcal{E}_\mu(f,\theta) = \mathcal{B}_\mu^l(f,\theta)$. Using the chain rule of the Fréchet derivative, we have

$$D_1\mathcal{E}_\mu(f,\theta) = D_1\mathcal{B}_\mu^l(f,\theta) = D_1\mathcal{B}_\mu\big(\mathcal{B}_\mu^{l-1}(f,\theta),\theta\big) \circ D_1\mathcal{B}_\mu^{l-1}(f,\theta). \tag{116}$$

Consequently, we can bound ② in a recursive way: for any two functions $f, f' \in \mathcal{C}(\mathcal{X})$

$$
\begin{aligned}
&\|D_1\mathcal{B}_\mu^l(f,\theta) - D_1\mathcal{B}_\mu^l(f',\theta)\|_{op} \\
&= \|D_1\mathcal{B}_\mu\big(\mathcal{B}_\mu^{l-1}(f,\theta),\theta\big) \circ D_1\mathcal{B}_\mu^{l-1}(f,\theta) - D_1\mathcal{B}_\mu\big(\mathcal{B}_\mu^{l-1}(f',\theta),\theta\big) \circ D_1\mathcal{B}_\mu^{l-1}(f',\theta)\|_{op} \\
&\leqslant \|D_1\mathcal{B}_\mu\big(\mathcal{B}_\mu^{l-1}(f,\theta),\theta\big) \circ \big(D_1\mathcal{B}_\mu^{l-1}(f,\theta) - D_1\mathcal{B}_\mu^{l-1}(f',\theta)\big)\|_{op} \\
&\quad + \|\left(D_1\mathcal{B}_\mu\big(\mathcal{B}_\mu^{l-1}(f,\theta),\theta\big) - D_1\mathcal{B}_\mu\big(\mathcal{B}_\mu^{l-1}(f',\theta),\theta\big)\right) \circ D_1\mathcal{B}_\mu^{l-1}(f',\theta)\|_{op} \\
&\leqslant \|D_1\mathcal{B}_\mu\big(\mathcal{B}_\mu^{l-1}(f,\theta),\theta\big)\|_{op}\|D_1\mathcal{B}_\mu^{l-1}(f,\theta) - D_1\mathcal{B}_\mu^{l-1}(f',\theta)\|_\infty \\
&\quad + \mathcal{O}(\|\mathcal{B}_\mu^{l-1}(f,\theta) - \mathcal{B}_\mu^{l-1}(f',\theta)\|_\infty \cdot \|D_1\mathcal{B}_\mu^{l-1}(f',\theta)\|_{op}) \\
&= \mathcal{O}(\|f - f'\|_\infty) + \|D_1\mathcal{B}_\mu^{l-1}(f,\theta) - D_1\mathcal{B}_\mu^{l-1}(f',\theta)\|_\infty,
\end{aligned}
$$

where in the first inequality we use the triangle inequality, in the second inequality, we use the definition of $\|\cdot\|_{op}$ and (115), and in the last equality we use (115) and the fact that $\mathcal{B}^k$ is Lipschitz continuous with respect its first argument for any finite $k$ (see Lemma B.9). Besides, since $f_\theta^\mu$ is continuous with respect to $\mu$ (see Lemma C.1), we have

$$\|D_1\mathcal{B}^l(f_\theta^\mu,\theta) - D_1\mathcal{B}^l(f_\theta^{\bar\mu},\theta)\|_{op} = \mathcal{O}(d_{bl}(\mu,\bar\mu)). \tag{117}$$

We then show that $\|Df_\theta^{\bar\mu}[h]\|_\infty = \mathcal{O}(\|h\|_\infty)$: Using (111), we have that

$$\|Df_\theta^{\bar\mu}[h]\|_\infty \leqslant \frac{2}{3}\|Df_\theta^{\bar\mu}[h]\|_\infty + \|D_2\mathcal{E}_\mu(f_\theta^\mu,\theta)[h]\|_\infty \Rightarrow \|Df_\theta^{\bar\mu}[h]\|_\infty \leqslant 3\|D_2\mathcal{E}_\mu(f_\theta^\mu,\theta)\|_{op}\|[h]\|_\infty.$$

Lemma B.7 shows that $\|D_2\mathcal{E}_\mu(f_\theta^\mu,\theta)\|_{op}$ is bounded and therefore we have

$$\|Df_\theta^{\bar\mu}[h]\|_\infty = \mathcal{O}(\|h\|_\infty). \tag{118}$$

Combining the above results, we obtain

$$② \leqslant \|D_1\mathcal{B}^l(f_\theta^\mu,\theta) - D_1\mathcal{B}^l(f_\theta^{\bar\mu},\theta)\|_{op}\|Df_\theta^{\bar\mu}[h]\|_\infty = \mathcal{O}(d_{bl}(\mu,\bar\mu) \cdot \|h\|_\infty).$$

### C.5.3 Bounding ③

Denote $\omega_y(x) = \exp(-\frac{c(x,y)}{\gamma})\exp(\bar f(x)/\gamma)$. Assume that $\|\bar f\|_\infty \leqslant M_c$ and $\|\nabla\bar f\|_{2,\infty} \leqslant G_f$. Then we have for any $y \in \mathcal{X}$,

$$\|\omega_y\|_\infty \leqslant \exp(M_c/\gamma), \quad \|\nabla\omega_y\|_{2,\infty} \leqslant \exp(M_c/\gamma)(G_c + G_f)/\gamma. \tag{119}$$

Therefore, $\|\omega_y\|_{bl} = \max\{\exp(M_c/\gamma), \exp(M_c/\gamma)(G_c + G_f)/\gamma\}$ is bounded (recall the definition of bounded Lipschitz norm in Theorem 6.1). Besides, for any $y \in \mathcal{X}$, $\omega_y(x)$ is positive and bounded away from zero

$$\omega_y(x) \geqslant \exp(-2M_c/\gamma). \tag{120}$$

For a fixed measure $\kappa$ and $g \in \mathcal{C}(\mathcal{X})$, we compute that

$$D_1\mathcal{A}(\bar f,\kappa)[g] = \frac{\int_\mathcal{X} \omega_y(x)g(x)\mathbf{d}\kappa(x)}{\int_\mathcal{X} \omega_y(x)\mathbf{d}\kappa(x)}. \tag{121}$$

This expression allows us to bound for two measures $\kappa$ and $\kappa'$

$$\left\|\left(D_1\mathcal{A}(\bar{f},\kappa)-D_1\mathcal{A}(\bar{f},\kappa')\right)[g]\right\|_\infty = \left\|\frac{\int_\mathcal{X}\omega_y(x)g(x)\mathbf{d}\kappa(x)}{\int_\mathcal{X}\omega_y(x)\mathbf{d}\kappa(x)}-\frac{\int_\mathcal{X}\omega_y(x)g(x)\mathbf{d}\kappa'(x)}{\int_\mathcal{X}\omega_y(x)\mathbf{d}\kappa'(x)}\right\|_\infty$$

$$\leqslant \left\|\frac{\int_\mathcal{X}\omega_y(x)g(x)\mathbf{d}\kappa(x)}{\int_\mathcal{X}\omega_y(x)\mathbf{d}\kappa(x)}-\frac{\int_\mathcal{X}\omega_y(x)g(x)\mathbf{d}\kappa(x)}{\int_\mathcal{X}\omega_y(x)\mathbf{d}\kappa'(x)}\right\|_\infty+\left\|\frac{\int_\mathcal{X}\omega_y(x)g(x)\mathbf{d}\kappa(x)}{\int_\mathcal{X}\omega_y(x)\mathbf{d}\kappa'(x)}-\frac{\int_\mathcal{X}\omega_y(x)g(x)\mathbf{d}\kappa'(x)}{\int_\mathcal{X}\omega_y(x)\mathbf{d}\kappa'(x)}\right\|_\infty.$$

We now bound these two terms individually. For the first term, we have

$$\left\|\frac{\int_\mathcal{X}\omega_y(x)g(x)\mathbf{d}\kappa(x)}{\int_\mathcal{X}\omega_y(x)\mathbf{d}\kappa(x)}-\frac{\int_\mathcal{X}\omega_y(x)g(x)\mathbf{d}\kappa(x)}{\int_\mathcal{X}\omega_y(x)\mathbf{d}\kappa'(x)}\right\|_\infty$$

$$\leqslant \left\|\int_\mathcal{X}\omega_y(x)g(x)\mathbf{d}\kappa(x)\right\|_\infty\left\|\frac{\int_\mathcal{X}\omega_y(x)\left[\mathbf{d}\kappa(x)-\mathbf{d}\kappa'(x)\right]}{\int_\mathcal{X}\omega_y(x)\mathbf{d}\kappa(x)\int_\mathcal{X}\omega_y(x)\mathbf{d}\kappa'(x)}\right\|_\infty$$

$$\leqslant \|\omega_y\|_\infty\cdot\|g\|_\infty\cdot\|\omega_y(x)\|_{bl}\cdot d_{bl}(\kappa,\kappa')\cdot\exp(4M_c/\gamma)=\mathcal{O}(\|g\|_\infty\cdot d_{bl}(\kappa,\kappa')),$$

where we use (119) and (120) in the last equality. For the second term, we bound

$$\left\|\frac{\int_\mathcal{X}\omega_y(x)g(x)\mathbf{d}\kappa(x)}{\int_\mathcal{X}\omega_y(x)\mathbf{d}\kappa(x)}-\frac{\int_\mathcal{X}\omega_y(x)g(x)\mathbf{d}\kappa'(x)}{\int_\mathcal{X}\omega_y(x)\mathbf{d}\kappa(x)}\right\|_\infty \leqslant \left\|\frac{\int_\mathcal{X}\omega_y(x)g(x)[\mathbf{d}\kappa(x)-\mathbf{d}\kappa'(x)]}{\int_\mathcal{X}\omega_y(x)\mathbf{d}\kappa(x)}\right\|_\infty$$

$$\leqslant \exp(M_c/\gamma)\cdot\|\omega_y(x)\|_{bl}\cdot\|g\|_{bl}\cdot d_{bl}(\kappa,\kappa')=\mathcal{O}(\|g\|_{bl}\cdot d_{bl}(\kappa,\kappa')).$$

Combining the above inequalities, we have

$$\left\|\left(D_1\mathcal{A}(\bar{f},\kappa)-D_1\mathcal{A}(\bar{f},\kappa')\right)[g]\right\|_\infty = \mathcal{O}(\|g\|_{bl}\cdot d_{bl}(\kappa,\kappa')). \tag{122}$$

Denote $\alpha=T_{\theta\sharp}\mu$ and $\bar{\alpha}=T_{\theta\sharp}\bar{\mu}$. From the chain rule of the Fréchet derivative, we compute

$$\left\|\left(D_1\mathcal{B}_\mu(f,\theta)-D_1\mathcal{B}_{\bar{\mu}}(f,\theta)\right)[g]\right\|_\infty$$

$$=\left\|\left(D_1\mathcal{A}\big(\mathcal{A}(f,\alpha),\beta_\mu\big)\circ D_1\mathcal{A}(f,\alpha)-D_1\mathcal{A}\big(\mathcal{A}(f,\bar{\alpha}),\beta_{\bar{\mu}}\big)\circ D_1\mathcal{A}(f,\bar{\alpha})\right)[g]\right\|_\infty$$

$$\leqslant \left\|D_1\mathcal{A}\big(\mathcal{A}(f,\alpha),\beta_\mu\big)\big[\big(D_1\mathcal{A}(f,\alpha)-D_1\mathcal{A}(f,\bar{\alpha})\big)[g]\big]\right\|_\infty$$

$$+\left\|\left(D_1\mathcal{A}\big(\mathcal{A}(f,\alpha),\beta_\mu\big)-D_1\mathcal{A}\big(\mathcal{A}(f,\alpha),\beta_{\bar{\mu}}\big)\right)\big[D_1\mathcal{A}(f,\bar{\alpha})[g]\big]\right\|_\infty$$

$$+\left\|\left(D_1\mathcal{A}\big(\mathcal{A}(f,\alpha),\beta_{\bar{\mu}}\big)-D_1\mathcal{A}\big(\mathcal{A}(f,\bar{\alpha}),\beta_{\bar{\mu}}\big)\right)\big[D_1\mathcal{A}(f,\bar{\alpha})[g]\big]\right\|_\infty.$$

We now bound these three terms one by one.
For the first term, use (110) to derive

$$\left\|D_1\mathcal{A}\big(\mathcal{A}(f,\alpha),\beta_\mu\big)\big[\big(D_1\mathcal{A}(f,\alpha)-D_1\mathcal{A}(f,\bar{\alpha})\big)[g]\big]\right\|_\infty$$
$$\leqslant \left\|D_1\mathcal{A}(f,\alpha)[g]-D_1\mathcal{A}(f,\bar{\alpha})[g]\right\|_\infty=\mathcal{O}(\|g\|_{bl}\cdot d_{bl}(\alpha,\bar{\alpha})),$$

where we use $\|D_1\mathcal{A}\big(\mathcal{A}(f,\alpha),\beta_\mu\big)\|_{op}\leqslant 1$ (75) and (122) in the second equality.

Combining the above result with (101) gives

$$\left\|D_1\mathcal{A}\big(\mathcal{A}(f,\alpha),\beta_\mu\big)\big[\big(D_1\mathcal{A}(f,\alpha)-D_1\mathcal{A}(f,\bar{\alpha})\big)[g]\big]\right\|_\infty=\mathcal{O}(\|g\|_{bl}\cdot d_{bl}(\mu,\bar{\mu})).$$

For the second term, use (122) to derive

$$\left\|\left(D_1\mathcal{A}\big(\mathcal{A}(f,\alpha),\beta_\mu\big)-D_1\mathcal{A}\big(\mathcal{A}(f,\alpha),\beta_{\bar{\mu}}\big)\right)\big[D_1\mathcal{A}(f,\bar{\alpha})[g]\big]\right\|_\infty$$
$$=\mathcal{O}(\|D_1\mathcal{A}(f,\bar{\alpha})[g]\|_{bl}\cdot d_{bl}(\beta_\mu,\beta_{\bar{\mu}})).$$

We now bound $\|D_1\mathcal{A}(f,\bar{\alpha})[g]\|_{bl}$. From (75), we have that $\|D_1\mathcal{A}(f,\bar{\alpha})[g]\|_\infty\leqslant\|g\|_\infty$. Besides, note that $D_1\mathcal{A}(f,\bar{\alpha})[g]$ is a function mapping from $\mathcal{X}$ to $\mathbb{R}$ and recall the expression of $D_1\mathcal{A}(f,\bar{\alpha})[g]$ in (121). To show that $D_1\mathcal{A}(f,\bar{\alpha})[g](y)$ is Lipschitz continuous w.r.t. $y$, we use the similar argument as (63): Under Assumption B.1 and assume that $\|f\|_\infty\leqslant M_c$, the numerator and denominator of (63)

are both Lipschitz continuous w.r.t. $y$ and bounded; the denominator is positive and bounded away from zero. Consequently, we can bound for any $y \in \mathcal{X}$

$$\|\nabla_y D_1 \mathcal{A}(f, \bar{\alpha})[g](y)\| \leqslant 2 \exp(4M_c/\gamma)\|g\|_\infty \cdot G_c, \tag{123}$$

and therefore

$$\left\|\left(D_1 \mathcal{A}\big(\mathcal{A}(f, \alpha), \beta_\mu\big) - D_1 \mathcal{A}\big(\mathcal{A}(f, \alpha), \beta_{\bar{\mu}}\big)\right)\big[D_1 \mathcal{A}(f, \bar{\alpha})[g]\big]\right\|_\infty = \mathcal{O}(\|g\|_\infty \cdot d_{bl}(\beta_\mu, \beta_{\bar{\mu}})).$$

For the third term, first note that we can use (101) and the mean value theorem to bound

$$\|\mathcal{A}(f, \alpha) - \mathcal{A}(f, \bar{\alpha})\|_\infty = \mathcal{O}(\max_{y \in \mathcal{X}} \|\omega_y\|_{bl} \cdot d_{bl}(\alpha, \bar{\alpha})) = \mathcal{O}(d_{bl}(\mu, \bar{\mu})). \tag{124}$$

Hence, we use Lemma B.11 to derive

$$\left\|\left(D_1 \mathcal{A}\big(\mathcal{A}(f, \alpha), \beta_{\bar{\mu}}\big) - D_1 \mathcal{A}\big(\mathcal{A}(f, \bar{\alpha}), \beta_{\bar{\mu}}\big)\right)\big[D_1 \mathcal{A}(f, \bar{\alpha})[g]\big]\right\|_\infty$$
$$= \mathcal{O}(\|\mathcal{A}(f, \alpha) - \mathcal{A}(f, \bar{\alpha})\|_\infty \cdot \|D_1 \mathcal{A}(f, \bar{\alpha})[g]\|_\infty) = \mathcal{O}(\|g\|_\infty \cdot d_{bl}(\mu, \bar{\mu})),$$

where we use (124) and the fact that $\|D_1 \mathcal{A}(f, \bar{\alpha})\|_{op}$ is bounded in the last equality.
Combing the above three results, we have

$$\left\|\big(D_1 \mathcal{B}_\mu(f, \theta) - D_1 \mathcal{B}_{\bar{\mu}}(f, \theta)\big)[g]\right\|_\infty = \mathcal{O}(\|g\|_{bl} \cdot d_{bl}(\mu, \bar{\mu})). \tag{125}$$

Recall that $\mathcal{E}_\mu(f, \theta) = \mathcal{B}_\mu^l(f, \theta)$. Using the chain rule of the Fréchet derivative, we have

$$D_1 \mathcal{E}_\mu(f, \theta) = D_1 \mathcal{B}_\mu^l(f, \theta) = D_1 \mathcal{B}_\mu\big(\mathcal{B}_\mu^{l-1}(f, \theta), \theta\big) \circ D_1 \mathcal{B}_\mu^{l-1}(f, \theta). \tag{126}$$

Denote $g = Df_\theta^{\bar{\mu}}[h]$. We can bound ③ in the following way:

$$③ = \|D_1 \mathcal{B}_\mu\big(\mathcal{B}_\mu^{l-1}(f, \theta), \theta\big)\big[D_1 \mathcal{B}_\mu^{l-1}(f, \theta)[g]\big] - D_1 \mathcal{B}_{\bar{\mu}}\big(\mathcal{B}_{\bar{\mu}}^{l-1}(f, \theta), \theta\big)\big[D_1 \mathcal{B}_{\bar{\mu}}^{l-1}(f, \theta)[g]\big]\|_\infty$$
$$\leqslant \|D_1 \mathcal{B}_\mu\big(\mathcal{B}_\mu^{l-1}(f, \theta), \theta\big)\big[\big(D_1 \mathcal{B}_\mu^{l-1}(f, \theta) - D_1 \mathcal{B}_{\bar{\mu}}^{l-1}(f, \theta)\big)[g]\big]\|_\infty$$
$$\quad + \|\left(D_1 \mathcal{B}_\mu\big(\mathcal{B}_\mu^{l-1}(f, \theta), \theta\big) - D_1 \mathcal{B}_\mu\big(\mathcal{B}_{\bar{\mu}}^{l-1}(f, \theta), \theta\big)\right)\big[D_1 \mathcal{B}_{\bar{\mu}}^{l-1}(f, \theta)[g]\big]\|_\infty$$
$$\quad + \|\left(D_1 \mathcal{B}_\mu\big(\mathcal{B}_{\bar{\mu}}^{l-1}(f, \theta), \theta\big) - D_1 \mathcal{B}_{\bar{\mu}}\big(\mathcal{B}_{\bar{\mu}}^{l-1}(f, \theta), \theta\big)\right)\big[D_1 \mathcal{B}_{\bar{\mu}}^{l-1}(f, \theta)[g]\big]\|_\infty$$
$$\leqslant \|D_1 \mathcal{B}_\mu\big(\mathcal{B}_\mu^{l-1}(f, \theta), \theta\big)\|_{op}\|\big(D_1 \mathcal{B}_\mu^{l-1}(f, \theta) - D_1 \mathcal{B}_{\bar{\mu}}^{l-1}(f, \theta)\big)[g]\|_\infty \qquad \#1$$
$$\quad + \mathcal{O}(\|\mathcal{B}_\mu^{l-1}(f, \theta) - \mathcal{B}_{\bar{\mu}}^{l-1}(f, \theta)\|_\infty \cdot \|D_1 \mathcal{B}_{\bar{\mu}}^{l-1}(f, \theta)[g]\|_\infty) \qquad \#2$$
$$\quad + \mathcal{O}(\|D_1 \mathcal{B}_{\bar{\mu}}^{l-1}(f, \theta)[g]\|_{bl} \cdot d_{bl}(\mu, \bar{\mu})), \qquad \#3$$

where in the first inequality we use the triangle inequality, in the second inequality we use the definition of $\|\cdot\|_{op}$, (115) and (125). We now analyze the R.H.S. of the above inequality one by one. For the first term, use $\|D_1 \mathcal{B}_\mu\big(\mathcal{B}_\mu^{l-1}(f, \theta), \theta\big)\|_{op} \leqslant 1$ and then use (125). We have

$$\#1 \leqslant \|\big(D_1 \mathcal{B}_\mu(f, \theta) - D_1 \mathcal{B}_{\bar{\mu}}(f, \theta)\big)[g]\|_\infty = \mathcal{O}(\|g\|_{bl} \cdot d_{bl}(\mu, \bar{\mu})).$$

For the second term, note that $\mathcal{B}_\mu^k$ is the composition of the terms $\mathcal{A}(f, \alpha)$ and $\mathcal{A}(f, \beta_\mu)$. Using a similar argument like (124), for any finite $k$, we have

$$\|\mathcal{B}_\mu^{l-1}(f, \theta) - \mathcal{B}_{\bar{\mu}}^{l-1}(f, \theta)\|_\infty = \mathcal{O}(d_{bl}(\mu, \bar{\mu})).$$

Together with the fact that $\|D_1 \mathcal{B}(f, \theta)\|_{op} \leqslant 1$, we have

$$\#2 = \mathcal{O}(\|g\|_\infty \cdot d_{bl}(\mu, \bar{\mu})).$$

Finally, for the third term, note that $\mathcal{B}_\mu$ is the composition of the terms $\mathcal{A}(f, \alpha)$ and $\mathcal{A}(f, \beta_\mu)$. Using a similar argument like (123) to bound

$$\#3 = \mathcal{O}(\|g\|_\infty \cdot d_{bl}(\mu, \bar{\mu})).$$

Combining these three results, we have

$$\text{③} = \|\big(D_1\mathcal{B}^l_\mu(f,\theta) - D_1\mathcal{B}^l_{\bar\mu}(f,\theta)\big)[g]\|_\infty = \mathcal{O}(\|g\|_{bl} \cdot d_{bl}(\mu,\bar\mu)). \tag{127}$$

We now bound $\|Df^{\bar\mu}_\theta[h]\|_{bl}$ $(g = Df^{\bar\mu}_\theta[h])$. From the fixed point definition of the Sinkhorn potential in (107), we can compute the Fréchet derivative $Df^\mu_\theta$ by

$$Df^\mu_\theta = D_1\mathcal{A}\big(\mathcal{A}(f^\mu_\theta,\alpha_\theta),\beta_\mu\big)\circ D_1\mathcal{A}(f^\mu_\theta,\alpha_\theta)\circ Df^\mu_\theta + D_1\mathcal{A}\big(\mathcal{A}(f^\mu_\theta,\alpha_\theta),\beta_\mu\big)\circ D_2\tilde{\mathcal{A}}(f^\mu_\theta,\theta), \tag{128}$$

where we recall $\tilde{\mathcal{A}}(f,\theta):=\mathcal{A}(f,\alpha_\theta)$. For any direction $h \in \mathbb{R}^d$ and any $y \in \mathcal{X}$, $Df^\mu_\theta[h]$ is a function with its gradient bounded by

$$\|\nabla_y Df^\mu_\theta[h](y)\| \leqslant \|\nabla_y\bigg(D_1\mathcal{A}\big(\mathcal{A}(f^\mu_\theta,\alpha_\theta),\beta_\mu\big)\Big[D_1\mathcal{A}(f^\mu_\theta,\alpha_\theta)\big[Df^\mu_\theta[h]\big]\Big]\bigg)(y)\| \qquad \text{\#1}$$

$$+\|\nabla_y\Big(D_1\mathcal{A}\big(\mathcal{A}(f^\mu_\theta,\alpha_\theta),\beta_\mu\big)\big[D_2\tilde{\mathcal{A}}(f^\mu_\theta,\theta)[h]\big]\Big)(y)\|. \qquad \text{\#2}$$

We now bound the R.H.S. individually:
For #1, take $\bar f = \mathcal{A}[f,\alpha_\theta]$, $\kappa = \beta_\mu$ and $g = D_1\mathcal{A}(f^\mu_\theta,\alpha_\theta)\big[Df^\mu_\theta[h]\big]$ in (121). Using (123) and (118), we have

$$\text{\#1} = \mathcal{O}(\|g\|_\infty) = \mathcal{O}(\|Df^\mu_\theta[h]\|_\infty) = \mathcal{O}(\|h\|). \tag{129}$$

For #2, take $\bar f = \mathcal{A}[f,\alpha_\theta]$, $\kappa = \beta_\mu$ and $g = D_2\tilde{\mathcal{A}}(f^\mu_\theta,\theta)[h]$ in (121). Using (123) and (79), we have

$$\text{\#2} = \mathcal{O}(\|g\|_\infty) = \mathcal{O}(\|D_2\tilde{\mathcal{A}}(f^\mu_\theta,\theta)[h]\|_\infty) = \mathcal{O}(\|h\|). \tag{130}$$

Combining these two bounds, we have

$$\|Df^\mu_\theta[h]\|_{bl} = \mathcal{O}(\|h\|). \tag{131}$$

By plugging the above result to (127), we bound

$$\text{③} = \|\big(D_1\mathcal{B}^l_\mu(f,\theta) - D_1\mathcal{B}^l_{\bar\mu}(f,\theta)\big)[g]\|_\infty = \mathcal{O}(d_{bl}(\mu,\bar\mu) \cdot \|h\|). \tag{132}$$

### C.5.4 Bounding ④

We have from the triangle inequality

$$\text{④} \leqslant \|D_2\mathcal{E}_\mu(f^\mu_\theta,\theta)[h] - D_2\mathcal{E}_{\bar\mu}(f^\mu_\theta,\theta)[h]\|_\infty + \|D_2\mathcal{E}_{\bar\mu}(f^\mu_\theta,\theta)[h] - D_2\mathcal{E}_{\bar\mu}(f^{\bar\mu}_\theta,\theta)[h]\|_\infty. \tag{133}$$

We analyze these two terms on the R.H.S..

For the first term of (133), use the chain rule of Fréchet derivative to compute

$$D_2\mathcal{E}_\mu(f,\theta)[h] = D_1\mathcal{B}_\mu\big(\mathcal{B}^{l-1}_\mu(f,\theta),\theta\big)\big[D_2\mathcal{B}^{l-1}_\mu(f,\theta)[h]\big] + D_2\mathcal{B}_\mu\big(\mathcal{B}^{l-1}_\mu(f,\theta),\theta\big)[h]. \tag{134}$$

Consequently, we can bound

$$\|\big(D_2\mathcal{E}_\mu(f,\theta) - D_2\mathcal{E}_{\bar\mu}(f,\theta)\big)[h]\|_\infty$$
$$\leqslant\|D_1\mathcal{B}_\mu\big(\mathcal{B}^{l-1}_\mu(f,\theta),\theta\big)\big[D_2\mathcal{B}^{l-1}_\mu(f,\theta)[h]\big] - D_1\mathcal{B}_{\bar\mu}\big(\mathcal{B}^{l-1}_{\bar\mu}(f,\theta),\theta\big)\big[D_2\mathcal{B}^{l-1}_{\bar\mu}(f,\theta)[h]\big]\|_\infty \quad \text{\#1}$$
$$+ \|D_2\mathcal{B}_\mu\big(\mathcal{B}^{l-1}_\mu(f,\theta),\theta\big)[h] - D_2\mathcal{B}_{\bar\mu}\big(\mathcal{B}^{l-1}_{\bar\mu}(f,\theta),\theta\big)[h]\|_\infty. \qquad \text{\#2}$$

We analyze #1 and #2 individually.

**Bounding #1.** We first note that $\mathcal{A}(f,\alpha)$ is Lipschitz continuous w.r.t. $\alpha$ (see also (124)):

$$\|\mathcal{A}(f,\alpha) - \mathcal{A}(f,\alpha')\|_\infty \leqslant \exp(2M_c/\gamma) \cdot \|\omega_y\|_{bl} \cdot d_{bl}(\alpha,\alpha') = \mathcal{O}(d_{bl}(\alpha,\alpha')), \tag{135}$$

where in the equality we use (119). As $\mathcal{B}^k_\mu$ is the composition of $\mathcal{A}$, it is Lipschitz continuous with respect to $\mu$ for finite $k$. Note that the boundedness of $\|f\|_\infty$ and $\|\nabla f\|_\infty$ remains valid after the

operator $\mathcal{B}$ (Lemma B.1 and (i) of Lemma (B.2)). We then bound

$$\#1 \leqslant \|D_1\mathcal{B}_\mu\big(\mathcal{B}_\mu^{l-1}(f,\theta),\theta\big)\big[\big(D_2\mathcal{B}_\mu^{l-1}(f,\theta) - D_2\mathcal{B}_{\bar{\mu}}^{l-1}(f,\theta)\big)[h]\big]\|_\infty$$

$$+ \|\Big(D_1\mathcal{B}_\mu\big(\mathcal{B}_{\bar{\mu}}^{l-1}(f,\theta),\theta\big) - D_1\mathcal{B}_\mu\big(\mathcal{B}_{\bar{\mu}}^{l-1}(f,\theta),\theta\big)\Big)\big[D_2\mathcal{B}_{\bar{\mu}}^{l-1}(f,\theta)[h]\big]\|_\infty$$

$$+ \|\Big(D_1\mathcal{B}_\mu\big(\mathcal{B}_{\bar{\mu}}^{l-1}(f,\theta),\theta\big) - D_1\mathcal{B}_{\bar{\mu}}\big(\mathcal{B}_{\bar{\mu}}^{l-1}(f,\theta),\theta\big)\Big)\big[D_2\mathcal{B}_{\bar{\mu}}^{l-1}(f,\theta)[h]\big]\|_\infty$$

$$\leqslant \|D_1\mathcal{B}_\mu\big(\mathcal{B}_\mu^{l-1}(f,\theta),\theta\big)\|_{op}\|D_2\mathcal{B}_\mu^{l-1}(f,\theta)[h] - D_2\mathcal{B}_{\bar{\mu}}^{l-1}(f,\theta)[h]\|_\infty$$

$$+ \mathcal{O}(\|\mathcal{B}_\mu^{l-1}(f,\theta) - \mathcal{B}_{\bar{\mu}}^{l-1}(f,\theta)\|_\infty \cdot \|D_2\mathcal{B}_{\bar{\mu}}^{l-1}(f,\theta)[h]\|_\infty)$$

$$+ \mathcal{O}(d_{bl}(\mu,\bar{\mu}) \cdot \|D_2\mathcal{B}_{\bar{\mu}}^{l-1}(f,\theta)[h]\|_\infty)$$

$$\leqslant \|D_2\mathcal{B}_\mu^{l-1}(f,\theta)[h] - D_2\mathcal{B}_{\bar{\mu}}^{l-1}(f,\theta)[h]\|_\infty + \mathcal{O}(d_{bl}(\mu,\bar{\mu}) \cdot \|h\|),$$

where in the second inequality we use the definition of $\|\cdot\|_{op}$, (115) and (125), and in the last inequality we use the fact that $\|D_1\mathcal{B}_\mu(f,\theta)\|_{op} \leqslant 1$, $\mathcal{B}_\mu^k$ is Lipschitz continuous with respect to $\mu$ for finite $k$ (see the discussion above) and that $\|D_2\mathcal{B}_{\bar{\mu}}^{l-1}(f,\theta)\|_{op}$ is bounded (see Lemma B.7.

**Bounding #2.** To make the dependences of $\mathcal{A}$ on $\theta$ and $\mu$ explicit, we denote

$$\hat{\mathcal{A}}(f,\theta,\mu) = \mathcal{A}(f, T_{\theta\sharp}\mu).$$

To bound the second term, we first establish that for any $k \geqslant 0$, $\nabla\mathcal{B}_\mu^{k+1}(f,\theta)$ is Lipschitz continuous w.r.t. $\mu$, i.e.

$$\|\nabla\mathcal{B}_\mu^{k+1}(f,\theta) - \nabla\mathcal{B}_{\bar{\mu}}^{k+1}(f,\theta)\|_{2,\infty} = \mathcal{O}(d_{bl}(\mu,\bar{\mu})), \tag{136}$$

as follows: First note that $\nabla\hat{\mathcal{A}}(f,\theta,\mu)$ is Lipschitz continuous w.r.t. $\mu$, i.e.

$$\|\nabla\hat{\mathcal{A}}(f,\theta,\mu)(y) - \nabla\hat{\mathcal{A}}(f,\theta,\bar{\mu})(y)\| = \mathcal{O}(d_{bl}(\mu,\bar{\mu})). \tag{137}$$

This is because for any $y \in \mathcal{X}$ (note that $\hat{\mathcal{A}}(f,\theta,\mu)(\cdot) : \mathcal{X} \to \mathbb{R}$ is a function of $y$),

$$\|\nabla\hat{\mathcal{A}}(f,\theta,\mu)(y) - \nabla\hat{\mathcal{A}}(f,\theta,\bar{\mu})(y)\|$$

$$= \|\frac{\int_\mathcal{X} \omega_y(x)\nabla_1 c(y,x)\mathbf{d}\alpha_\theta(x)}{\int_\mathcal{X} \omega_y(x)\mathbf{d}\alpha_\theta(x)} - \frac{\int_\mathcal{X} \omega_y(x)\nabla_1 c(y,x)\mathbf{d}\bar{\alpha}_\theta(x)}{\int_\mathcal{X} \omega_y(x)\mathbf{d}\bar{\alpha}_\theta(x)}\|$$

$$\leqslant \|\frac{\int_\mathcal{X} \omega_y(x)\nabla_1 c(y,x)\big(\mathbf{d}\alpha_\theta(x) - \mathbf{d}\bar{\alpha}_\theta(x)\big)}{\int_\mathcal{X} \omega_y(x)\mathbf{d}\alpha_\theta(x)}\|$$

$$+ \|\int_\mathcal{X} \omega_y(x)\nabla_1 c(y,x)\mathbf{d}\bar{\alpha}_\theta(x)\| \cdot \|\frac{\int_\mathcal{X} \omega_y(x)\big(\mathbf{d}\alpha_\theta(x) - \mathbf{d}\bar{\alpha}_\theta(x)\big)}{\int_\mathcal{X} \omega_y(x)\mathbf{d}\alpha_\theta(x) \int_\mathcal{X} \omega_y(x)\mathbf{d}\bar{\alpha}_\theta(x)}\|$$

$$= \mathcal{O}(d_{bl}(\mu,\bar{\mu})).$$

Here in the last equality, we use the facts that $\|\omega_y(\cdot)\nabla_1 c(y,\cdot)\|_{bl}$ and $\|\omega_y\|_{bl}$ are bounded, and $\int_\mathcal{X} \omega_y(x)\mathbf{d}\alpha_\theta(x)$ is strictly positive and bounded away from zero. Recall that $\mathcal{B}_\mu(f,\theta) = \mathcal{A}(\hat{\mathcal{A}}(f,\theta,\mu),\beta_\mu)$. We can then prove (136) by bounding

$$\|\nabla\mathcal{B}_\mu^{k+1}(f,\theta) - \nabla\mathcal{B}_{\bar{\mu}}^{k+1}(f,\theta)\|$$

$$= \|\nabla\mathcal{A}(\hat{\mathcal{A}}(\mathcal{B}_\mu^k(f,\theta),\theta,\mu),\beta_\mu) - \nabla\mathcal{A}(\hat{\mathcal{A}}(\mathcal{B}_{\bar{\mu}}^k(f,\theta),\theta,\bar{\mu}),\bar{\beta}_\mu)\|$$

$$\leqslant \|\nabla\mathcal{A}(\hat{\mathcal{A}}(\mathcal{B}_\mu^k(f,\theta),\theta,\mu),\beta_\mu) - \nabla\mathcal{A}(\hat{\mathcal{A}}(\mathcal{B}_\mu^k(f,\theta),\theta,\mu),\bar{\beta}_\mu)\| \qquad \&1$$

$$+ \|\nabla\mathcal{A}(\hat{\mathcal{A}}(\mathcal{B}_\mu^k(f,\theta),\theta,\mu),\bar{\beta}_\mu) - \nabla\mathcal{A}(\hat{\mathcal{A}}(\mathcal{B}_\mu^k(f,\theta),\theta,\bar{\mu}),\bar{\beta}_\mu)\| \qquad \&2$$

$$+ \|\nabla\mathcal{A}(\hat{\mathcal{A}}(\mathcal{B}_\mu^k(f,\theta),\theta,\bar{\mu}),\bar{\beta}_\mu) - \nabla\mathcal{A}(\hat{\mathcal{A}}(\mathcal{B}_{\bar{\mu}}^k(f,\theta),\theta,\bar{\mu}),\bar{\beta}_\mu)\| \qquad \&3$$

$$= \mathcal{O}(d_{bl}(\mu,\bar{\mu}))$$

Here we bound $\&1$ using (137), the Lipschitz continuity of $\nabla\mathcal{A}$ w.r.t. its second variable; we bound $\&2$ using the Lipschitz continuity of $\nabla\hat{\mathcal{A}}$ w.r.t. its first variable and (124), the Lipschitz continuity of $\hat{\mathcal{A}}$ w.r.t. $\mu$; we bound $\&3$ using (124), the Lipschitz continuity of $\hat{\mathcal{A}}$ w.r.t. $\mu$, and the fact that $\mathcal{B}_\mu^k$ is the composition of the terms $\mathcal{A}(f,\alpha)$ and $\mathcal{A}(f,\beta_\mu)$.

We then establish that $D_2\mathcal{B}_\mu(f,\theta)$ is Lipschitz continuous w.r.t. $\mu$.

**Assumption C.1.** $\|\nabla_z[\nabla_\theta T_\theta(z)]\|_{op}$ *is bounded*

**Lemma C.12.** *Assume that* $\|f\|_\infty \leqslant M_c$, $\|\nabla f\|_{2,\infty} \leqslant G_f$, $\|\nabla^2 f\|_{op,\infty} \leqslant L_f$ *Under Assumptions B.5, C.1 and B.1, we have*

$$\|D_2 \mathcal{B}_\mu(f,\theta) - D_2 \mathcal{B}_{\bar\mu}(f,\theta)\|_{op} = \mathcal{O}(d_{bl}(\mu,\bar\mu)). \tag{138}$$

*Proof.* Denote $\omega_y(x) = \exp\left(\frac{-c(x,y)+f(x)}{\gamma}\right)$ and

$$\phi_y(z) = [\nabla_\theta T_\theta(z)]^\top [-\nabla_1 c(T_\theta(z),y) + \nabla f(T_\theta(z))]$$

where $\nabla_\theta T_\theta(z)$ denotes the Jacobian matrix of $T_\theta(z)$ with respect to $\theta$.

The Fréchet derivative $D_2 \hat{\mathcal{A}}(f,\theta,\mu)[h]$ can be computed by

$$D_2 \hat{\mathcal{A}}(f,\theta,\mu)[h] = \frac{\int_\mathcal{X} \omega_y(T_\theta(z))\langle\phi_y(z),h\rangle \mathbf{d}\mu(z)}{\int_\mathcal{X} \omega_y(T_\theta(z))\mathbf{d}\mu(z)}. \tag{139}$$

Recall that $\|f\|_\infty \leqslant M_c$, $\|\nabla f\|_{2,\infty} \leqslant G_f$. Using the above expression we can bound

$$\|(D_2\hat{\mathcal{A}}(f,\theta,\mu) - D_2\hat{\mathcal{A}}(f,\theta,\bar\mu))[h]\|_\infty$$

$$= \left\|\frac{\int_\mathcal{X} \omega_y(T_\theta(z))\langle\phi_y(z),h\rangle \mathbf{d}\mu(z)}{\int_\mathcal{X} \omega_y(T_\theta(z))\mathbf{d}\mu(z)} - \frac{\int_\mathcal{X} \omega_y(T_\theta(z))\langle\phi_y(z),h\rangle \mathbf{d}\bar\mu(x)}{\int_\mathcal{X} \omega_y(T_\theta(z))\mathbf{d}\bar\mu(x)}\right\|_\infty$$

$$\leqslant \left\|\frac{\int_\mathcal{X} \omega_y(T_\theta(z))\langle\phi_y(z),h\rangle \mathbf{d}\mu(z)}{\int_\mathcal{X} \omega_y(T_\theta(z))\mathbf{d}\mu(z)} - \frac{\int_\mathcal{X} \omega_y(T_\theta(z))\langle\phi_y(z),h\rangle \mathbf{d}\bar\mu(x)}{\int_\mathcal{X} \omega_y(T_\theta(z))\mathbf{d}\mu(z)}\right\|_\infty$$

$$+ \left\|\frac{\int_\mathcal{X} \omega_y(T_\theta(z))\langle\phi_y(z),h\rangle \mathbf{d}\bar\mu(x)}{\int_\mathcal{X} \omega_y(T_\theta(z))\mathbf{d}\mu(z)} - \frac{\int_\mathcal{X} \omega_y(T_\theta(z))\langle\phi_y(z),h\rangle \mathbf{d}\bar\mu(x)}{\int_\mathcal{X} \omega_y(T_\theta(z))\mathbf{d}\bar\mu(x)}\right\|_\infty$$

$$= \left\|\frac{\int_\mathcal{X} \omega_y(T_\theta(z))\langle\phi_y(z),h\rangle [\mathbf{d}\mu(z) - \mathbf{d}\bar\mu(x)]}{\int_\mathcal{X} \omega_y(T_\theta(z))\mathbf{d}\mu(z)}\right\|_\infty$$

$$+ \left\|\frac{\int_\mathcal{X} \omega_y(T_\theta(z))\langle\phi_y(z),h\rangle \mathbf{d}\bar\mu(x) \int_\mathcal{X} \omega_y(T_\theta(z))[\mathbf{d}\bar\mu(x) - \mathbf{d}\mu(z)]}{\int_\mathcal{X} \omega_y(T_\theta(z))\mathbf{d}\mu(z) \int_\mathcal{X} \omega_y(T_\theta(z))\mathbf{d}\bar\mu(x)}\right\|_\infty$$

$$\leqslant \exp(2M_c/\gamma) \cdot \|\omega_y(T_\theta(z))\langle\phi_y(z),h\rangle\|_{bl} \cdot d_{bl}(\mu,\bar\mu)$$

$$+ \exp(5M_c/\gamma) \cdot \|\phi_y\|_\infty \cdot \|h\|_\infty \cdot \|\omega_y\|_{bl} \cdot d_{bl}(\mu,\bar\mu).$$

For the first term, note that $\|\omega_y(T_\theta(z))\langle\phi_y(z),h\rangle\|_{bl} \leqslant \|\omega_y\|_{bl} \cdot \|\phi_y\|_{bl} \cdot \|h\|_\infty$ and $\|\omega_y\|_{bl}$ is bounded (see (119)). We just need to bound $\|\phi_y\|_{bl}$. Under Assumption B.5 that $\|\nabla_\theta T_\theta(z)\|_{op} \leqslant G_T$, we clearly have that $\|\phi_y\|_\infty$ is bounded. For $\|\phi_y\|_{lip}$, compute that

$$\nabla_z \phi_y(z) = \nabla_z[\nabla_\theta T_\theta(z)] \times_1 [-\nabla_1 c(T_\theta(z),y) + \nabla f(T_\theta(z))]$$
$$+ \nabla_\theta T_\theta(z)^\top [-\nabla_{11}^2 c(T_\theta(z),y) + \nabla^2 f(T_\theta(z))] \nabla_\theta T_\theta(z).$$

Recall that $\|\nabla^2 f(x)\|_{op}$ is bounded. Consequently, under Assumption C.1, we can see that $\|\nabla_z \phi_y(z)\|$ is bounded. Together, $\|\phi_y\|_{bl}$ is bounded. As a result, we have

$$\|(D_2\hat{\mathcal{A}}(f,\theta,\mu) - D_2\hat{\mathcal{A}}(f,\theta,\bar\mu))[h]\|_\infty = \mathcal{O}(d_{bl}(\mu,\bar\mu) \cdot \|h\|). \tag{140}$$

Based on the above result, we can further bound

$$\|(D_2\mathcal{B}_\mu(f,\theta) - D_2\mathcal{B}_{\bar\mu}(f,\theta))[h]\|_\infty$$

$$= \left\|\left(D_1\mathcal{A}(\hat{\mathcal{A}}(f,\theta,\mu),\beta) \circ D_2\hat{\mathcal{A}}(f,\theta,\mu) - D_1\mathcal{A}(\hat{\mathcal{A}}(f,\theta,\bar\mu),\bar\beta) \circ D_2\hat{\mathcal{A}}(f,\theta,\bar\mu)\right)[h]\right\|_\infty$$

$$\leqslant \|D_1\mathcal{A}(\hat{\mathcal{A}}(f,\theta,\mu),\beta)[(D_2\hat{\mathcal{A}}(f,\theta,\mu) - D_2\hat{\mathcal{A}}(f,\theta,\bar\mu))[h]]\|_\infty \qquad \#\#1$$

$$+ \left\|\left(D_1\mathcal{A}(\hat{\mathcal{A}}(f,\theta,\mu),\beta) - D_1\mathcal{A}(\hat{\mathcal{A}}(f,\theta,\mu),\bar\beta)\right)[D_2\hat{\mathcal{A}}(f,\theta,\bar\mu)[h]]\right\|_\infty \qquad \#\#2$$

$$+ \left\|\left(D_1\mathcal{A}(\hat{\mathcal{A}}(f,\theta,\mu),\bar\beta) - D_1\mathcal{A}(\hat{\mathcal{A}}(f,\theta,\bar\mu),\bar\beta)\right)[D_2\hat{\mathcal{A}}(f,\theta,\bar\mu)[h]]\right\|_\infty. \qquad \#\#3$$

For the first term, use $\|D_1\mathcal{A}(\hat{\mathcal{A}}(f,\theta,\mu),\beta)\|_{op} \leqslant 1$ (75) and (140) to bound

$$\#\#1 \leqslant \|D_2\hat{\mathcal{A}}(f,\theta,\mu)[h] - D_2\hat{\mathcal{A}}(f,\theta,\bar{\mu})[h]\|_\infty = \mathcal{O}(d_{bl}(\mu,\bar{\mu}) \cdot \|h\|).$$

For the second term, recall the expression of $D_2\hat{\mathcal{A}}(f,\theta,\bar{\mu})[h]$ in (139). Under Assumption B.1 and assume that $\|f\|_\infty \leqslant M_c$, one can see that $\|D_2\hat{\mathcal{A}}(f,\theta,\bar{\mu})[h]\|_{bl} = \mathcal{O}(\|h\|)$. Further, use (122) and $d_{bl}(\beta,\bar{\beta}) = \mathcal{O}(d_{bl}(\mu,\bar{\mu}))$ from (101) to bound

$$\#\#2 = \mathcal{O}(\|D_2\hat{\mathcal{A}}(f,\theta,\bar{\mu})[h]\|_{bl} \cdot d_{bl}(\beta,\bar{\beta})) = \mathcal{O}(\|h\| \cdot d_{bl}(\mu,\bar{\mu})).$$

For the third term, use Lemma B.11 to bound

$$\#\#3 = \mathcal{O}(\|D_2\hat{\mathcal{A}}(f,\theta,\bar{\mu})[h]\|_\infty \cdot \|\hat{\mathcal{A}}(f,\theta,\mu) - \hat{\mathcal{A}}(f,\theta,\bar{\mu})\|_\infty) = \mathcal{O}(d_{bl}(\mu,\bar{\mu}) \cdot \|h\|),$$

where we use $\|D_2\hat{\mathcal{A}}(f,\theta,\bar{\mu})[h]\|_\infty = \mathcal{O}(\|h\|)$ and (124). Altogether, we have

$$\|D_2\mathcal{B}_\mu(f,\theta)[h] - D_2\mathcal{B}_{\bar{\mu}}(f,\theta)[h]\|_\infty = \mathcal{O}(d_{bl}(\mu,\bar{\mu}) \cdot \|h\|). \tag{141}$$

$\square$

We are now ready to bound #2.

$$\begin{aligned}\#2 &\leqslant \|D_2\mathcal{B}_\mu\big(\mathcal{B}_\mu^{l-1}(f,\theta),\theta\big)[h] - D_2\mathcal{B}_\mu\big(\mathcal{B}_{\bar{\mu}}^{l-1}(f,\theta),\theta\big)[h]\|_\infty \\ &\quad + \|D_2\mathcal{B}_\mu\big(\mathcal{B}_{\bar{\mu}}^{l-1}(f,\theta),\theta\big)[h] - D_2\mathcal{B}_{\bar{\mu}}\big(\mathcal{B}_{\bar{\mu}}^{l-1}(f,\theta),\theta\big)[h]\|_\infty \\ &= \mathcal{O}(\|\mathcal{B}_\mu^{l-1}(f,\theta) - \mathcal{B}_{\bar{\mu}}^{l-1}(f,\theta)\|_\infty + \|\nabla\mathcal{B}_\mu^{l-1}(f,\theta) - \nabla\mathcal{B}_{\bar{\mu}}^{l-1}(f,\theta)\|_{2,\infty}) \\ &\quad + \mathcal{O}(d_{bl}(\mu,\bar{\mu}) \cdot \|h\|) \\ &= \mathcal{O}(d_{bl}(\mu,\bar{\mu}) \cdot \|h\|),\end{aligned}$$

where we use Lemma B.10 and (138) (124) in the first equality.

**Combining #1 and #2.** Combining the above results, we yield

$$\|D_2\mathcal{B}_\mu^l(f,\theta)[h] - D_2\mathcal{B}_{\bar{\mu}}^l(f,\theta)[h]\|_\infty \leqslant \|D_2\mathcal{B}_\mu^{l-1}(f,\theta)[h] - D_2\mathcal{B}_{\bar{\mu}}^{l-1}(f,\theta)[h]\|_\infty + \mathcal{O}(d_{bl}(\mu,\bar{\mu}) \cdot \|h\|_\infty),$$

which, via recursion, implies that (recall that $D_2\mathcal{E}_\mu(f,\theta)[h] = D_2\mathcal{B}_\mu^l(f,\theta)[h]$)

$$\|D_2\mathcal{E}_\mu(f,\theta)[h] - D_2\mathcal{E}_{\bar{\mu}}(f,\theta)[h]\|_\infty = \mathcal{O}(d_{bl}(\mu,\bar{\mu}) \cdot \|h\|). \tag{142}$$

To bound the second term of (133), compute the expression of $D_2\mathcal{E}_{\bar{\mu}}(f,\theta)[h]$ via the chain rule:

$$D_2\mathcal{E}_{\bar{\mu}}(f,\theta)[h] = D_1\mathcal{B}_{\bar{\mu}}\big(\mathcal{B}_{\bar{\mu}}^{l-1}(f,\theta),\theta\big)\big[D_2\mathcal{B}_{\bar{\mu}}^{l-1}(f,\theta)[h]\big] + D_2\mathcal{B}_{\bar{\mu}}\big(\mathcal{B}_{\bar{\mu}}^{l-1}(f,\theta),\theta\big)[h]. \tag{143}$$

Recall that $\mathcal{E}_{\bar{\mu}}(f,\theta) = \mathcal{B}_{\bar{\mu}}^l(f,\theta)$. We then show in an inductive manner that the second term of (133) is of order $\mathcal{O}(d_{bl}(\mu,\bar{\mu}) \cdot \|h\|)$: For any finite $k \geqslant 1$,

$$\|D_2\mathcal{B}_{\bar{\mu}}^k(f_\theta^\mu,\theta)[h] - D_2\mathcal{B}_{\bar{\mu}}^k(f_\theta^{\bar{\mu}},\theta)[h]\|_\infty = \mathcal{O}(d_{bl}(\mu,\bar{\mu}) \cdot \|h\|). \tag{144}$$

For the base case when $l = 1$, we only have the second term of (143) in $D_2\mathcal{E}_{\bar{\mu}}(f,\theta)[h]$. Consequently, from Lemma B.10, we have

$$\begin{aligned}&\|D_2\mathcal{B}_{\bar{\mu}}\big(\mathcal{B}_{\bar{\mu}}^{l-1}(f_\theta^\mu,\theta),\theta\big) - D_2\mathcal{B}_{\bar{\mu}}\big(\mathcal{B}_{\bar{\mu}}^{l-1}(f_\theta^{\bar{\mu}},\theta),\theta\big)\|_{op} \\ &\quad = \mathcal{O}(\|\mathcal{B}_{\bar{\mu}}^{l-1}(f_\theta^\mu,\theta) - \mathcal{B}_{\bar{\mu}}^{l-1}(f_\theta^{\bar{\mu}},\theta)\|_\infty + \|\nabla\mathcal{B}_{\bar{\mu}}^{l-1}(f_\theta^\mu,\theta) - \nabla\mathcal{B}_{\bar{\mu}}^{l-1}(f_\theta^{\bar{\mu}},\theta)\|_{2,\infty}) = \mathcal{O}(d_{bl}(\mu,\bar{\mu})),\end{aligned} \tag{145}$$

where we use (136) in the second equality.

Now assume that for $l = k$ the statement (144) holds. For any two function $f, f' \in \mathcal{C}(\mathcal{X})$, we bound

$$\|D_2 \mathcal{B}_{\bar{\mu}}^k(f, \theta)[h] - D_2 \mathcal{B}_{\bar{\mu}}^k(f', \theta)[h]\|_\infty$$

$$\leqslant \|D_1 \mathcal{B}_{\bar{\mu}}\big(\mathcal{B}_{\bar{\mu}}^{l-1}(f, \theta), \theta\big)\big[D_2 \mathcal{B}_{\bar{\mu}}^{l-1}(f, \theta)[h] - D_2 \mathcal{B}_{\bar{\mu}}^{l-1}(f', \theta)[h]\big]\|_\infty$$

$$+ \|\Big(D_1 \mathcal{B}_{\bar{\mu}}\big(\mathcal{B}_{\bar{\mu}}^{l-1}(f, \theta), \theta\big) - D_1 \mathcal{B}_{\bar{\mu}}\big(\mathcal{B}_{\bar{\mu}}^{l-1}(f', \theta), \theta\big)\Big)\big[D_2 \mathcal{B}_{\bar{\mu}}^{l-1}(f', \theta)[h]\big]\|_\infty$$

$$+ \|\Big(D_2 \mathcal{B}_{\bar{\mu}}\big(\mathcal{B}_{\bar{\mu}}^{l-1}(f, \theta), \theta\big) - D_2 \mathcal{B}_{\bar{\mu}}\big(\mathcal{B}_{\bar{\mu}}^{l-1}(f', \theta), \theta\big)\Big)[h]\|_\infty.$$

$$\leqslant \|\big(D_2 \mathcal{B}_{\bar{\mu}}^{l-1}(f, \theta) - D_2 \mathcal{B}_{\bar{\mu}}^{l-1}(f', \theta)\big)[h]\|_\infty \qquad\qquad \|D_1 \mathcal{B}_{\bar{\mu}}(f, \theta)\|_{op} \leqslant 1$$

$$+ \mathcal{O}(\|\mathcal{B}_{\bar{\mu}}^{l-1}(f, \theta) - \mathcal{B}_{\bar{\mu}}^{l-1}(f', \theta)\|_\infty \cdot \|D_2 \mathcal{B}_{\bar{\mu}}^{l-1}(f', \theta)[h]\|_\infty) \qquad \text{Lemma B.8}$$

$$+ \mathcal{O}(d_{bl}(\mu, \bar{\mu}) \cdot \|h\|). \qquad\qquad (145)$$

$$= \mathcal{O}((\|f - f'\|_\infty + \|\nabla f - \nabla f'\|_{2,\infty}) \cdot \|h\|) \qquad\qquad \text{Lemma B.5}$$

$$\mathcal{O}((\|f - f'\|_\infty) \cdot \|h\|)$$

$$\mathcal{O}(d_{bl}(\mu, \bar{\mu}) \cdot \|h\|).$$

Plug in $f = f_\theta^\mu$ and $f' = f_\theta^{\bar{\mu}}$ and use Lemmas C.1 and C.2. We prove the statement (144) holds for $l = k + 1$. Consequently, we have that

$$\|D_2 \mathcal{E}_{\bar{\mu}}(f_\theta^\mu, \theta)[h] - D_2 \mathcal{E}_{\bar{\mu}}(f_\theta^{\bar{\mu}}, \theta)[h]\|_\infty = \mathcal{O}(d_{bl}(\mu, \bar{\mu}) \cdot \|h\|). \qquad\qquad (146)$$

In conclusion, we have

$$④ = \mathcal{O}(d_{bl}(\mu, \bar{\mu}) \cdot \|h\|). \qquad\qquad (147)$$

Table 1: Structure of the encoder

| Layer (type) | Output Shape | Param # |
|---|---|---|
| Conv2d-1 | [-1, 64, 32, 32] | 4,800 |
| LeakyReLU-2 | [-1, 64, 32, 32] | 0 |
| Conv2d-3 | [-1, 128, 16, 16] | 204,800 |
| BatchNorm2d-4 | [-1, 128, 16, 16] | 256 |
| LeakyReLU-5 | [-1, 128, 16, 16] | 0 |
| Conv2d-6 | [-1, 256, 8, 8] | 819,200 |
| BatchNorm2d-7 | [-1, 256, 8, 8] | 512 |
| LeakyReLU-8 | [-1, 256, 8, 8] | 0 |
| Conv2d-9 | [-1, 512, 4, 4] | 3,276,800 |
| BatchNorm2d-10 | [-1, 512, 4, 4] | 1,024 |
| LeakyReLU-11 | [-1, 512, 4, 4] | 0 |

Table 2: Structure of the generator

| Layer (type) | Output Shape | Param # |
|---|---|---|
| ConvTranspose2d-1 | [-1, 256, 4, 4] | 262,144 |
| BatchNorm2d-2 | [-1, 256, 4, 4] | 512 |
| ReLU-3 | [-1, 256, 4, 4] | 0 |
| ConvTranspose2d-4 | [-1, 128, 8, 8] | 524,288 |
| BatchNorm2d-5 | [-1, 128, 8, 8] | 256 |
| ReLU-6 | [-1, 128, 8, 8] | 0 |
| ConvTranspose2d-7 | [-1, 64, 16, 16] | 131,072 |
| BatchNorm2d-8 | [-1, 64, 16, 16] | 128 |
| ReLU-9 | [-1, 64, 16, 16] | 0 |
| ConvTranspose2d-10 | [-1, 3, 32, 32] | 3,072 |
| Tanh-11 | [-1, 3, 32, 32] | 0 |

# D Experiment Details

We use the generator from DC-GAN Radford et al. [2015]. And the adversarial ground cost $c_\xi$ in the form of

$$c_\xi(x, y) = \|\phi_\xi(x) - \phi_\xi(y)\|_2^2, \tag{148}$$

where $\phi_\xi : \mathbb{R}^q \to \mathbb{R}^{\hat{q}}$ is an encoder that maps the original data point (and the generated image) to a higher dimensional space ($\hat{q} > q$). We pick $\phi_\xi$ to be an CNN with a similar structure as the discriminator of DC-GAN except that we discard the last layer which was used for classification. Specifically, the networks used are given in Table 1 and 2.

We set the step size $\beta$ of SiNG to be 30 and set the maximum allow Sinkhorn divergence in each iteration to be 0.1. Note that the step size is set after the normalization in (11). For Adam, RMSprop, and AMSgrad, we set all of their initial step sizes to be $1.0 \times e^{-3}$, which is in general recommended by the GAN literature. The minibatch sizes of both the real images and the generated images for each iteration are set to 3000. We uniformly set the $\gamma$ parameter in the objective (recall that $\mathcal{F}(\alpha_\theta) = \mathcal{S}_{c_\xi}(\alpha_\theta, \beta)$) and the constraint to 100.

The code is in `https://github.com/shenzebang/Sinkhorn_Natural_Gradient`.

# E  PyTorch Implementation

In this section, we focus on the empirical version of SiNG, where we approximate the gradient of the function $F$ by a minibatch stochastic gradient and approximate SIM by eSIM. In this case, all components involved in the optimization procedure can be represented by finite dimensional vectors.

It is known that the stochastic gradient admits an easy implementation in PyTorch. However, at the first sight, the computation of eSIM is quite complicated as it requires to construct two sequences $f^t$ and $g^t$ to estimate the Sinkhorn potential and the Fréchet derivative. As we discussed earlier, it is well known that we can solve the inversion of a p.s.d. matrix via the Conjugate Gradient (CG) method with only matrix-vector-product operations. In particular, in this case, we no longer need to explicitly form eSIM in the computer memory. Consequently, to implement the empirical version of SiNG using CG and eSIM, one can resort to the auto-differential mechanism provided by PyTorch: First, we use existing PyTorch package like geomloss[3] to compute the tensor $\mathbf{f}$ representing the Sinkhorn potential $f_\theta^\epsilon$. Note the the sequence $f^t$ is constructed implicitly by calling geomloss. We then use the ".detach()" function in PyTorch to maintain only the value of the $\mathbf{f}$ while discarding all of its "grad_fn" entries. We then enable the "autograd" mechanism is PyTorch and run several loops of Sinkhorn mapping $\mathcal{A}(f, \alpha_\theta)$ ($\mathcal{A}(f, \alpha_{\theta^t})$) so that the output tensor now records all the dependence on the parameter $\theta$ via the implicitly constructed computational graph. We can then easily compute the matrix-vector-product use the Pearlmutter's algorithm (Pearlmutter, 1994).

## Footnotes

[1]Recall that $T(\mathbb{R}^d, \mathcal{C}(\mathcal{X}))$ is the family of bounded linear operators from $\mathbb{R}^d$ to $\mathcal{C}(\mathcal{X})$

[2] Recall that $T(U, W)$ is the family of bounded linear operators from $U$ to $W$.

[3]https://www.kernel-operations.io/geomloss/