[Reviews · NeurIPS 2020]

Review 1

Summary and Contributions: Authors propose a new algorithm, sinkhorn natural gradient, for learning paramters in a generative model. A fairly thorough theoretical analysis is provided. Some experimental validation is presented as well. I appreciate author's response. I have increased my previous judgement.

Strengths: I think it is a very good paper. It develops a new algorithm akin to natural gradient, and also develop computational theory of the Sinkhorn information matrix, a computationally tractable alternative to wasserstein information matrix. The approximation theory for the gradient and hessians is very much welcome. It involves some tedious calculations but to my understanding they are novel and are a nice contribution to the community. I foresee this paper will generate impact.

Weaknesses: The validation section is a bit weak, and may be improved. I am not surprised of computational improvements because of the use of extra hessian information. However, the computational cost of such operation can be large. Authors should either comment on that or quantify the gains. The paper would be much stronger if authors were able to argue this extra computational expense is justified. Does the quality of generated images improve? authors may show examples of generated images. How these results compare to the ones of Genevay et al, 2018? It is too bad the cost function and domain have to be bounded. How the authors may deal with the unbounded case? perhaps they may appeal to results in https://arxiv.org/abs/1905.11882

Correctness: Arguments seem correct to me

Clarity: It is well written. Some notation seems unnecessary (e.g. defining a discrete operator \mathcal{A})

Relation to Prior Work: Prior work is correctly acknowledged. Perhaps the author may need to cite some other work that also deals with the hessian of the sinkhorn divergence (e.g. https://arxiv.org/pdf/1812.09150.pdf)

Reproducibility: Yes

Additional Feedback: It is too bad the cost function has to be bounded. How the authors may deal with the unbounded case? perhaps they may appeal to results in https://arxiv.org/abs/1905.11882. Additionally, recently the Sinkhorn EM algorithm was proposed, which involves similar hessian computations. Authors may comment on the relationship between both methods (if any) https://arxiv.org/abs/2006.16548 There is an unsatisfying issue, which is not unique to this paper: authors develop the natural gradient method using the Sinkhorn distance (i.e. the debiased entropic OT distance) but then define the Sinkhorn information matrix using the entropic OT distance without Debiasing. This is confusing as it does not lead insights as to which one is the one that really matters. Some more clarification on this would be appreciated.


Review 2

Summary and Contributions: The authors aim at deriving a formula for the natural gradient with Sinkhorn proximity constraints in the case the probability distributions are given by the pushforward of a ground distribution through a parametrized map, the standard setting for GANs. Edit after rebuttal: I have read the authors' response and the other reviews. I have updated my score.

Strengths: The derived methods seem well-grounded in entropic regularized optimal transport and the derived formulas easy to implement.

Weaknesses: The experiments could have been more extensive. A thorough description is missing. Several questions are left unanswered: How stable is the training with the Sinkhorn natural gradient? How good are the results? How good would the results be, if the ground cost would be fixed (non-adversarial)? etc.

Correctness: The methods seem correct.

Clarity: The structure, notations and clarity of the paper can be improved. For instance, the part 'Training adversarial generative models' is not well written.

Relation to Prior Work: Some difference are highlighted, like the relation to Wasserstein natural gradients, but not much is discussed.

Reproducibility: Yes

Additional Feedback:


Review 3

Summary and Contributions: The paper studies the minimization problem in generative models. The authors consider using a gradient operator generated by Sinkorn divergence, a perturbed Wasserstein divergence. They design fast algorithm based on Sinkorn natural gradient. In experiments, they compare Sinkorn natural gradient with other methods and demonstrate its numerical benefits.

Strengths: The paper studies cruical questions in AI optimization problems. This is to consider parameterization invariant gradient operator in parameter space. The current natural gradient in Fisher-Rao metric can be not well defined in generative models. The Wasserstein natural metric and its generalizations are very useful in this direction.

Weaknesses: I have several concerns in this work. 1. For Sinkhorn divergence, can you provide us several analytical examples, in which SIM has closed form solutions, such as generative model one dimensional space and Gaussian family? 2. If the ground cost is not quadratic, does the Sinkorn divergence still provide us metric structure? For example, suppose the ground cost is homogenous degree one, see c(x,y)=|x-y|_1, can you provide us a Sinkhorn matrix? Any simple example?

Correctness: The method is correct.

Clarity: The paper is well written. Some clarification is needed when the ground cost is not homogenous of degree two.

Relation to Prior Work: For the discussion of Sinkhorn divergence, the connection with Schrodinger bridge problem and Fisher information regularization is needed. See attached references. F. Leger, W.C. Li. Hopf-Cole transformation via generalized Schrodinger bridge problem. 2. S.N.Chow, W.C. Li, C.C. Mou, H.M. Zhou. A discrete Schrodinger bridge problem via optimal transport on graphs. For the discussion of Wasserstein information matrix, the following important papers are also needed: 1. W.C. Li, J.X. Zhao. Wasserstein information matrix. 2. W.C. Li, G. Montufar. Ricci curvature for parameter statistics via optimal transport, Information geometry.

Reproducibility: Yes

Additional Feedback: The authors address my questions. In particular, they point out an interesting analytical example. A good paper. It may be also good to include this example in the revision.


Review 4

Summary and Contributions: This paper proposes a natural gradient algorithm for training implicit generative models. They propose to use the Sinkhorn divergence as the "natural" metric. They then derive an update that requires the computation of the Sinkhorn Information Matrix (SIM), and show that propose a way to approximate it efficiently. They also provide an empirical version of SIM. Finally they show experimentally that the proposed method converge faster than standard SGD type algorithm for training a generative model.

Strengths: I believe this work is an important contribution to the field of natural gradient algorithm. The work is well motivated by theory and the derivation of the algorithm is quite rigorous, the paper tries to justify each choice as much as possible through theory.

Weaknesses: I think the experimental section of the paper is bit weak and I'm not convinced the proposed algorithm can scale very well (see below for more details)

Correctness: I haven't checked the proofs but from a quick look the claims seem reasonable and well supported. The empirical methodology is a bit weak: - The author compare the convergence of the different methods in terms of number of epochs, however I believe it's a bit unfair to other methods which are much simpler, I think wall clock time would be more appropriate, as the cost per iteration of the proposed method could be several times higher than SGD. - It would have been nice to have some confidence intervals for the plots. - A lot of questions remain unanswered: what is the effect of the number of iteration of conjugate gradient to compute the inverse on the convergence ? same question but for computing the SIM ? Furthermore what is the influence of the batch size ? I suspect that the proposed algorithm requires larger batch-size in order to work. - I find the setting for comparing the different methods quite limited. It would have been more interesting to compare the performance of the proposed algorithm in a standard GAN setup.

Clarity: The paper is overall well written and quite clear. However the experimental section lack some details, such as the number of iteration of conjugate gradient used to compute the inverse and the number of iterations (or accuracy) used for computing the SIM.

Relation to Prior Work: The related work is well discussed.

Reproducibility: No

Additional Feedback: === Edit after rebuttal === I'm increasing my score as the authors addressed most of my comments in the rebuttal.

[Author Response · NeurIPS 2020]

We thank the reviewers for their careful consideration and constructive feedback. Below, please find our responses.

**To Reviewer #1.** Q1. Computational cost of using Hessian. A1. The reviewer is right: The per-iteration cost of SiNG is larger than SGD-based methods since we use the conjugate gradient method to compute the update direction. However, this additional cost is well justified as the the output of SiNG has a significantly smaller objective value and hence higher solution quality compared to SGD-based methods, when using *the same* wall-clock time (e.g. see figure (iv)). We will add the discussion in our revision. Q2. Image quality compared to [Genevay et al, 2018]. A2. Please see Figure (i) ([Genevay et al, 2018]) and Figure (ii) (our paper). The entropy regularization of the Sinkhorn divergence is set to $\gamma = 100$ as suggested in Table 2 of [Genevay et al, 2018]. The regularization for the constraint is set to $\gamma = 1$ in SiNG. We used ADAM as the optimizer for the discriminators (with step size $10^{-3}$ and batch size 4000). We can see that the images generated using SiNG are much more vivid than the ones obtained using SGD-based optimizers. We remark that our main goal has been to showcase that SiNG is more efficient in reducing the objective value compared to SGD-based solvers, and hence, we have used a relatively simpler DC-GAN type generator and discriminator (details given in the supplementary materials). If more sophisticated ResNet type generators and discriminators are used, the image quality can be further improved.

(i) Adam-GAN      (ii) SiNG-GAN      (iii) stability of SiNG      (iv) G-loss vs time

Q3. Unbounded cost functions. A3. Thanks for pointing out this interesting work. We will definitely look into it. Q4. Relationship to Sinkhorn EM (sEM). A4. We thank the reviewer for pointing out this interesting work. SiNG and sEM have different problem formulations: In SiNG, the objective is to find a parameterized measure $\alpha_\theta$ such that the functional $\mathcal{F}$ (see eq. (1) in our paper) is minimized; On the other hand, sEM considers an entropy regularized optimal transport distance with a parameterized ground cost $g_\theta$ (see eq.(3) in their paper). Hence, the Hessian computations involved in SiNG and sEM are quite different. We will discuss this in our revision. Q5. Sinkhorn information matrix (SIM) is defined without debiasing. A5. SIM is defined to be the Hessian of the Sinkhorn divergence in eq. (10). There are two terms in SIM: $\nabla_\theta^2 \mathcal{S}(\alpha_\theta, \beta) = \nabla_\theta^2 \mathrm{OT}_\gamma(\alpha_\theta, \beta) + \nabla_\theta^2 \mathrm{OT}_\gamma(\alpha_\theta, \alpha_\theta)$. We only derived the explicit expression of the first term due to space limitation. The second "debiasing" term is obtained similarly. Both terms do matter and are involved in our computations. We will elaborate on this in our revision. We apologize for the confusion.

**To Reviewer #2.** Q1. The stability of SiNG. A1. SiNG is stable in both the fixed ground cost case (see figure (iv)) and the adversarial case (see figure (iii)). Q2. How good are the results? A2. Please see our response A2 to reviewer #1 and figures (i) and (ii). Q3. Results for fixed ground cost. A3. We present the comparison of the generator loss (the objective value) vs time plot in figure (iv), where we have used the DC-GAN generator and the squared $\ell_2$ distance as ground metric (no encoder). The entropy regularization parameter $\gamma$ is set to 0.01 for both the objective and the constraint. We can see that SiNG is much more efficient at reducing the objective value than ADAM given the same amount of time. Due to the limitation of space, we do not provide the images here, but we observe that SiNG is producing more diversified images than ADAM (possibly due to mode collapse of ADAM). We will discuss these in our revision.

**To Reviewer #3.** Q1. SIM for 1-d Gaussian. A1. Please see Theorem 1 of https://arxiv.org/abs/2006.02572 for the closed form expression of entropy regularized optimal transport between to Gaussians. Using their result, in the simplest case where only the mean of $\alpha$ is parameterized, i.e. $\alpha = \mathcal{N}(\mu(\theta), \sigma^2)$, SIM admits the closed form $\nabla_\theta^2 \mathcal{S}(\alpha_\theta, \beta) = 2\nabla_\theta^2 \mu(\theta)$. Q2. SIM under $\ell_1$ ground cost. A2. This is an interesting question. Right now we are not able to provide SIM in this case: Our results rely on the smoothness of the Sinkhorn potential and hence the smoothness of the ground cost function which does not apply to the $\ell_1$ case. Q3. Relations to previous work. A3. We will definitely discuss these important and interesting related works in our revision.

**To Reviewer #5.** Q1. Wall clock time comparison. A1. The reviewer is correct about this. SiNG has higher per-iteration computational complexity compared to the simple SGD based methods. However, such cost is justified by the efficiency of SiNG in reducing objective value. Please see A1 to Reviewer #1 and figure (iv). Q2. Add confidence interval in the plots A2. We will add confidence in our revision. Q3. Comparison in standard GAN setting. A3. In the standard GAN setting, by discarding the last layer of the discriminator, we can have an encoder of the input image. This encoder will be acting like the adversarial ground cost we trained in eq. (7). We will add comparisons between SiNG and SGD based solvers in standard GAN setting. Thanks. Q4. Details of implementation. A3. We take 20 CG steps to compute the inverse of SIM without explicitly formulating it (only matrix vector product). In our experiments, batch size of 4000 is already sufficient. We will elaborate more on the details of the experiments in our revision.

[Meta-Review · NeurIPS 2020]

All reviewers agree that this is a good paper and the authors have adequately answered their queries in the rebuttal (in particular by providing an interesting and non-trivial example). There only some minor concerns about the scalability of the numerical method and the lack of comparison with standard GAN setups. I thus recommend to accept the paper.